# VAE-Var: Variational Autoencoder-Enhanced Variational Methods for Data Assimilation in Meteorology

**Yi Xiao**[1,2]   **Qilong Jia**[1]   **Kun Chen**[2]   **Lei Bai**[2✉]   **Wei Xue**[1✉]

[1]Department of Computer Science and Technology, Tsinghua University
[2]Shanghai Artificial Intelligence Laboratory
{y-xiao22,jql23}@mails.tsinghua.edu.cn    kunc3301@163.com
baisanshi@gmail.com    xuewei@tsinghua.edu.cn

## Abstract

Data assimilation (DA) is an essential statistical technique for generating accurate estimates of a physical system's states by combining prior model predictions with observational data, especially in the realm of weather forecasting. Effectively modeling the prior distribution while adapting to diverse observational sources presents significant challenges for both traditional and neural network-based DA algorithms. This paper introduces VAE-Var, a novel neural network-based data assimilation algorithm aimed at 1) enhancing accuracy by capturing the non-Gaussian characteristics of the conditional background distribution $p(\mathbf{x}|\mathbf{x}_b)$, and 2) efficienctly assimilating real-world observational data. VAE-Var utilizes a variational autoencoder to learn the background error distribution, with its decoder creating a variational cost function to optimize the analysis states. The advantages of VAE-Var include: 1) it maintains the framework of traditional variational assimilation, enabling it to accommodate various observation operators, particularly irregular observations; 2) it lessens the dependence on expert knowledge for constructing the background distribution, allowing for improved modeling of non-Gaussian structures; and 3) experimental results indicate that, when applied to the FengWu weather forecasting model, VAE-Var outperforms DiffDA and two traditional algorithms (interpolation and 3DVar) in terms of assimilation accuracy in sparse observational contexts, and is capable of assimilating real-world GDAS prepbufr observations over a year.

## 1 Introduction

Data assimilation (DA) is a statistical approach aimed at modeling the posterior likelihood distribution of a physical system's states, represented as $p(\mathbf{x}|\mathbf{x}_b, \mathbf{y})$. This distribution is conditioned on prior predictions, known as background states $\mathbf{x}_b$, and observational data $\mathbf{y}$, ultimately yielding an accurate estimate, referred to as analysis states $\mathbf{x}_a$, through sampling from $p(\mathbf{x}|\mathbf{x}_b, \mathbf{y})$ or by maximizing the likelihood. Data assimilation is critical for deriving initial states in various fields, particularly in numerical weather forecasting (Bauer et al., 2015; Kalnay, 2003; Carrassi et al., 2018).

In data assimilation for weather forecasting, background states $\mathbf{x}_b$ are essentially predictions produced by an imprecise forecasting model, with the conditional likelihood $p(\mathbf{x}|\mathbf{x}_b)$ often remaining unknown. Due to the high dimensionality of weather systems, modeling $p(\mathbf{x}|\mathbf{x}_b)$ presents significant challenges (Strogatz, 2018). Since meteorological observations originate from different sources, the types of $\mathbf{y}$ are highly diverse. Typically, the observation operator $\mathcal{H}$ is used to represent the mapping from physical space to observational space. An effective data assimilation algorithm must not only deliver accurate estimates of the conditional background distribution $p(\mathbf{x}|\mathbf{x}_b)$ but also proficiently manage various observation operators, preferably in a zero-shot manner.

Traditional algorithms and neural network-based approaches offer distinct advantages in addressing the data assimilation problem. Traditional data assimilation methods, particularly variational assimilation algorithms, operate under the assumption that the background error, $\mathbf{x} - \mathbf{x}_b$, follows

a Gaussian distribution that is independent of $\mathbf{x}_b$ (Barker et al., 2004; Bannister, 2008; Descombes et al., 2015; Trémolet, 2006). Under this assumption, the posterior likelihood $p(\mathbf{x}|\mathbf{x}_b, \mathbf{y})$ can be calculated explicitly and estimation can be achieved by maximizing the likelihood function. The advantage of variational methods lies in its robustness to various observation operators; regardless of how $\mathcal{H}$ is formulated, maximization can always be performed using techniques such as gradient descent. This approach has proven successful for over two decades (Rabier et al., 1998; 2000). However, it relies heavily on expert knowledge for constructing the correlations of background error structures (Huang et al., 2009); additionally, the assumption of Gaussian error distribution may constrain the potential accuracy of this approach.

In recent years, generative neural networks, particularly diffusion models, have been increasingly utilized in data assimilation to model the background conditional distribution. Two notable examples are SDA (Rozet & Louppe, 2023) and DiffDA (Huang et al., 2024). In both studies, a diffusion model is first employed to represent the background distribution $p(\mathbf{x}|\mathbf{x}_b)$. Subsequently, SDA leverages diffusion posterior sampling techniques (Chung et al., 2022) to incorporate observational information $\mathbf{y}$, while DiffDA utilizes the repaint technique (Lugmayr et al., 2022) to integrate observations and sample from $p(\mathbf{x}|\mathbf{x}_b, \mathbf{y})$ during inference. The advantage of these approaches over traditional algorithms lies in their ability to model more complex structures for $p(\mathbf{x}|\mathbf{x}_b)$, potentially enhancing the accuracy of the analysis states $\mathbf{x}_a$. However, these methods frequently show limitations in dealing with different observation operators. For example, DiffDA's repaint technique assumes that observations are aligned with a grid, whereas in real-world scenarios, observations are typically very sparse and likely fall outside the grid. Similarly, SDA relies on certain assumptions to apply diffusion posterior sampling, and experimental results indicate that these assumptions can undermine assimilation accuracy, particularly in cases where observations are sparse or noisy.

In this paper, we aim to develop a neural network-based data assimilation algorithm that not only captures the non-Gaussian characteristics of the conditional background distribution for enhanced accuracy but also effectively assimilates data under real-world observations (sparse and outside of the grid). We introduce VAE-Var, a novel data assimilation algorithm in which a variational autoencoder is first employed to learn the conditional background distribution and then the decoder component is utilized to construct a variational cost function, which, when optimized, yields the analysis states. The advantages of our approach are outlined as follows:

Table 1: Comparison among traditional DA method (Trad-DA), diffusion-based DA (NN-DA) and VAE-Var

|  | Structure of $p(\mathbf{x}|\mathbf{x}_b)$ | Observation Operator |
| --- | --- | --- |
| Trad-DA | Gaussian | **arbitrary** |
| NN-DA | **non-Gaussian** | limited |
| VAE-Var | **non-Gaussian** | **arbitrary** |

- This algorithm inherits the framework of traditional variational assimilation by explicitly modeling the posterior probability function $p(\mathbf{x}|\mathbf{x}_b, \mathbf{y})$ and maximizing it to derive the analysis states. As a result, compared to other neural network data assimilation methods such as SDA and DiffDA, VAE-Var can better handle different types of observation operators, particularly irregular observations that do not fall on the grid points of the physical field.

- Unlike traditional variational assimilation algorithms, VAE-Var alleviates the dependence on expert knowledge for constructing the conditional background distribution, enabling the model to effectively capture non-Gaussian structures.

- Experimental results on the FengWu weather forecasting system of $0.25°$ resolution demonstrate that VAE-Var achieves higher assimilation accuracy than both DiffDA and two traditional algorithms for most variables in the sparse observational settings. Furthermore, when integrated with FengWu, it can reliably assimilate real-world GDAS prepbufr observations over a one-year period.

## 2 PRELIMINARIES

**Variational Assimilation** Variational assimilation seeks to determine the probability density function $p(\mathbf{x}|\mathbf{x}_b, \mathbf{y})$ of the physical state's distribution at a specific time, given known observational conditions $\mathbf{y}$ and the background state $\mathbf{x}_b$, and calculate the analysis state $\mathbf{x}_a$ by maximizing this probability density function, that is, $\mathbf{x}_a = \arg\max_{\mathbf{x}} p(\mathbf{x}|\mathbf{x}_b, \mathbf{y})$.

Since $\mathbf{x}_b$ is derived from a weather forecasting model and $\mathbf{y}$ from observatories and satellites, it is reasonable to assume that $\mathbf{x}_b$ and $\mathbf{y}$ are independent. Under this assumption, the following equation holds, with the derivation provided in the Appendix:

$$\arg\max_{\mathbf{x}} p(\mathbf{x}|\mathbf{x}_b, \mathbf{y}) = \arg\max_{\mathbf{x}} p(\mathbf{y}|\mathbf{x})p(\mathbf{x}|\mathbf{x}_b). \tag{1}$$

We define the variational cost as:

$$\mathcal{L}(\mathbf{x}) = -\log p(\mathbf{y}|\mathbf{x})p(\mathbf{x}|\mathbf{x}_b) = -\log p(\mathbf{y}|\mathbf{x}) - \log p(\mathbf{x}|\mathbf{x}_b). \tag{2}$$

Maximizing the probability density function is equivalent to minimizing the variational cost. This cost comprises two terms: an observation term, $\mathcal{L}_o(\mathbf{x}, \mathbf{y}) = -\log p(\mathbf{y}|\mathbf{x})$ and a background term, $\mathcal{L}_b(\mathbf{x}, \mathbf{x}_b) = -\log p(\mathbf{x}|\mathbf{x}_b)$.

**Observation Term**    In this paper, we only consider observations where the observed variables are consistent with the physical field variables, and exclude more complex observations like brightness temperature. In this case, the error of the observation operator can be simplified to a Gaussian distribution with a diagonal covariance matrix (Kalnay, 2003), that is, $\mathbf{y}|\mathbf{x} \sim \mathcal{N}(\mathcal{H}(\mathbf{x}), \mathbf{R})$, where $\mathbf{R}$ corresponds to the variance matrix. The observation term can then be formulated as follows:

$$\mathcal{L}_o(\mathbf{x}, \mathbf{y}) = \frac{1}{2}(\mathbf{y} - \mathcal{H}(\mathbf{x}))^{\mathrm{T}}\mathbf{R}^{-1}(\mathbf{y} - \mathcal{H}(\mathbf{x})). \tag{3}$$

**Background Term**    As previously mentioned, the background state $\mathbf{x}_b$ is derived by integrating a forecasting model from a previous state. In most cases, the forecasting model cannot perfectly capture all dynamical information and is subject to inherent errors. Traditional variational assimilation algorithms assume that these errors are independent of $\mathbf{x}_b$ and they follow a Gaussian distribution, that is, $\mathbf{x} - \mathbf{x}_b \sim \mathcal{N}(\mathbf{0}, \mathbf{B})$, where $\mathbf{B}$ is the background error covariance matrix. The background term can then be expressed as:

$$\mathcal{L}_b(\mathbf{x}, \mathbf{x}_b) = \frac{1}{2}(\mathbf{x} - \mathbf{x}_b)^{\mathrm{T}}\mathbf{B}^{-1}(\mathbf{x} - \mathbf{x}_b). \tag{4}$$

In global weather systems, the dimensionality of physical states can be extremely large. For instance, in the FengWu forecasting model with 37 vertical layers (Chen et al., 2023), the dimension of physical fields is $189 \times 721 \times 1440 \approx 2 \times 10^8$, making the dimension of $\mathbf{B}$ approximately $4 \times 10^{16}$. Explicitly constructing such a large matrix is generally impractical. To address this, a common approach is to decompose $\mathbf{B}$ as $\mathbf{B} = \mathbf{U}\mathbf{U}^{\mathrm{T}}$ and apply a variable transformation, $\mathbf{x} = \mathbf{U}\mathbf{z} + \mathbf{x}_b$. With this transformation, the background term can be rewritten as: $\tilde{\mathcal{L}}_b(\mathbf{z}) = \mathcal{L}_b(\mathbf{U}\mathbf{z} + \mathbf{x}_b, \mathbf{x}_b) = \frac{1}{2}\mathbf{z}^{\mathrm{T}}\mathbf{z}$, and the observation term becomes: $\tilde{\mathcal{L}}_o(\mathbf{z}) = \mathcal{L}_o(\mathbf{U}\mathbf{z} + \mathbf{x}_b, \mathbf{y})$. By minimizing $\tilde{\mathcal{L}}(\mathbf{z}) = \tilde{\mathcal{L}}_b(\mathbf{z}) + \tilde{\mathcal{L}}_o(\mathbf{z}) = \frac{1}{2}\mathbf{z}^{\mathrm{T}}\mathbf{z} + \mathcal{L}_o(\mathbf{U}\mathbf{z} + \mathbf{x}_b, \mathbf{y})$, it eliminates the need to explicitly calculate the inverse of $\mathbf{B}$. In traditional 3DVar, the sparse matrix $\mathbf{U}$ is derived by utilizing expert knowledge to statistically analyze background error samples, which are constructed using the NMC method (Descombes et al., 2015).

## 3    VAE-VAR

### 3.1    BACKGROUND ERROR ESTIMATION BASED ON VAE

Reviewing the process of traditional variational algorithms, we can identify two key factors for their successful implementation: First, it assumes that the error of the background states $\mathbf{x} - \mathbf{x}_b$ is independent of the background states $\mathbf{x}_b$, which enables the explicit construction of the expression $p(\mathbf{x}|\mathbf{x}_b)$. Second, it assumes that the background states' error follows a Gaussian distribution and constructs a linear mapping $\mathbf{U}$ through expert knowledge, which can map a variable $\mathbf{z}$ from the standard normal distribution to the background states' error distribution ($\mathbf{U}^{-1}(\mathbf{x} - \mathbf{x}_b) \sim \mathcal{N}(\mathbf{0}, \mathbf{I})$).

In our study, we choose to maintain the first assumption. This decision is based on the fact that, in real-world scenarios, the dimensionality is often very high while the available training dataset is relatively small. If we were to eliminate this assumption, there is a significant risk that the network would learn specious correlations between $\mathbf{x}$ and $\mathbf{x}_b$, a common issue in data assimilation. Conversely, we opt to discard the second assumption, as the high non-linearity of the forecasting

model is likely to result in a non-Gaussian error distribution. An illustrative example is provided in the Appendix for further clarification. If the error distribution is non-Gaussian, then a non-linear mapping (denoted as $\mathcal{F}$) is needed to map the standard normal distribution to the background states' error distribution ($\mathcal{F}^{-1}(\mathbf{x} - \mathbf{x}_b) \sim \mathcal{N}(\mathbf{0}, \mathbf{I})$).

There are three primary types of candidate network structures suitable for this task. The first option is the normalizing flow (Rezende & Mohamed, 2015; Kingma & Dhariwal, 2018), which establishes a deterministic mapping between the standard normal distribution and a target distribution. However, its high computational cost and large parameter size make it impractical for learning high-dimensional global weather fields. The second option is the diffusion model, which has strong fitting capabilities, but introduces multiple rounds of sampling during inference, and the transformation between the Gaussian and target space is not straightforward to track. The third and preferred choice is the variational autoencoder (VAE) (Kingma & Welling, 2013). VAE exhibits strong fitting abilities and has been successful on high-dimensional data (Doersch, 2016). While VAE creates a stochastic mapping from a latent Gaussian variable to the target space, the core of the mapping (i.e., the decoder) is deterministic, making it easier to formulate the variational cost. Thus, we adopt the VAE to perform the transformation.

A VAE typically comprises two components: an encoder $\mathcal{E}$ and a decoder $\mathcal{D}$. The encoder maps the input data into a mean and a variance, from which we sample a Gaussian distribution. This sampled output is then fed into the decoder. The goal of the VAE is to learn the structure of the data space. Once the VAE is well-trained, we can simply sample from a standard normal distribution, pass it through the decoder, and the output will resemble a sample from the original dataset. Essentially, the decoder in the VAE is a good choice to serve as the non-linear mapping $\mathcal{F}$ for transforming the latent space to the data space.

## 3.2 GENERAL FORMULATION OF VAE-VAR

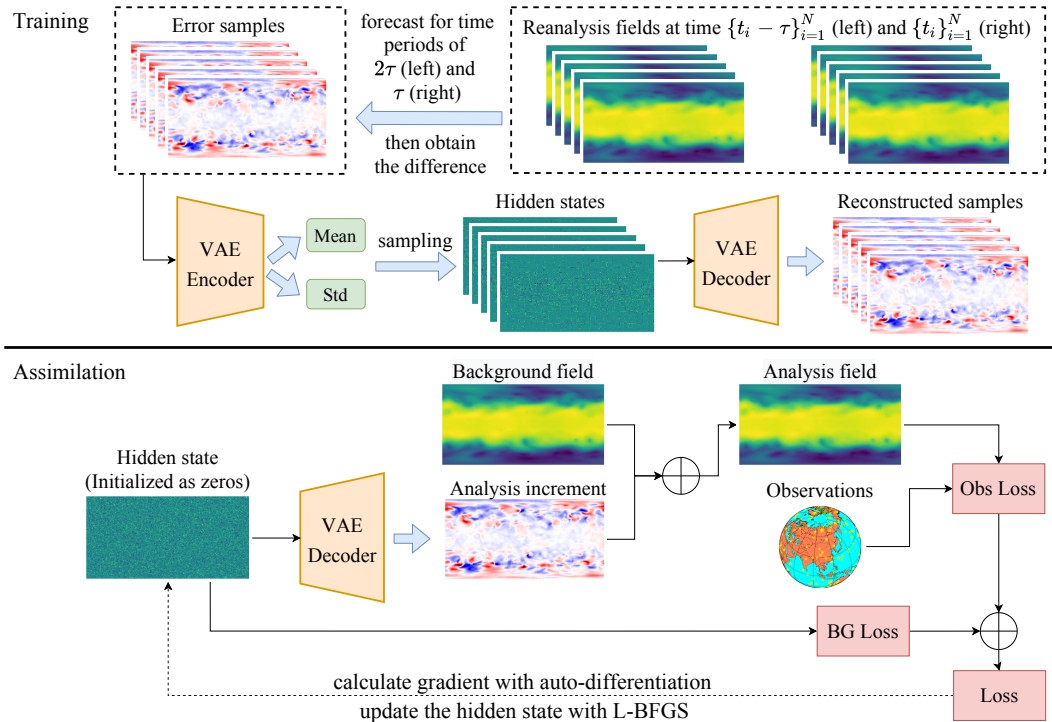

Figure 1: **VAE-Var Framework**. The upper half panel demonstrates the training phase; the lower half panel demonstrates the assimilation phase. Please refer to the main text for detailed explanation.

Here, we propose our VAE-Var framework for data assimilation. As shown in Figure 1, it comprises two phases: training and assimilation.

**Training Set Construction**   During the training phase, we adopt the traditional NMC (National Meteorological Center) method to generate historical error samples, as outlined in Algorithm 1. The reanalysis data are treated as the ground truth of states. We select historical reanalysis states at intervals of $\tau$, then integrate them for a time period of $\tau$ and $2\tau$ respectively, ensuring that both are predictions for the same timestamp. By subtracting these two states, we generate an error sample. Repeating this process across a long period of reanalysis data allows us to construct a training set.

---

**Algorithm 1** Training Set Construction

---

**Require:** : Forecasting model $\mathcal{M}$, sample number $N$, time gap $\tau$
  **for** $i$ from 1 to $N$ **do**
    Randomly pick reanalysis data with an interval of $\tau$: $\mathbf{x}_0, \mathbf{x}_\tau$
    $\hat{\mathbf{x}}_1 \leftarrow \mathcal{M}_{\tau \to 2\tau}(\mathbf{x}_\tau)$
    $\hat{\mathbf{x}}_2 \leftarrow \mathcal{M}_{\tau \to 2\tau} \circ \mathcal{M}_{0 \to \tau}(\mathbf{x}_0)$
    Add $\hat{\mathbf{x}}_1 - \hat{\mathbf{x}}_2$ to the training set
  **end for**

---

**Algorithm 2** VAE-Var Assimilation

---

**Require:** : Trained decoder $\mathcal{D}$, background state $\mathbf{x}_b$, observation $\mathbf{y}$
  $\mathbf{z} \leftarrow \mathbf{0}$
  $\tilde{\mathcal{L}}_b(\mathbf{z}) \leftarrow \frac{1}{2}\lambda \mathbf{z}^{\mathrm{T}}\mathbf{z}$
  $\tilde{\mathcal{L}}_o(\mathbf{z}) \leftarrow \mathcal{L}_o(\mathcal{D}(\mathbf{z}) + \mathbf{x}_b, \mathbf{y})$
  Minimize $\tilde{\mathcal{L}}(\mathbf{z}) = \tilde{\mathcal{L}}_b(\mathbf{z}) + \tilde{\mathcal{L}}_o(\mathbf{z})$ with L-BFGS (Liu & Nocedal, 1989; Paszke et al., 2017) and get the minimum point $\mathbf{z}^\star$
  $\mathbf{x}_a = \mathcal{D}(\mathbf{z}^\star) + \mathbf{x}_b$

---

**Loss Function**   We employ the standard approach for training the VAE, where the loss function consists of two components: the reconstruction loss ($Loss_{rec}$) and the Kullback-Leibler (KL) divergence ($Loss_{KL}$). The reconstruction loss quantifies how accurately the decoder can regenerate the original input from the latent space, which, in our case, is defined as the mean squared error between the input and its reconstruction. The KL divergence ensures that the learned latent distribution aligns closely with a standard normal distribution. These two terms together guide the optimization process, enabling the VAE to capture the underlying data distribution effectively. Additionally, a hyperparameter $\sigma$ is introduced to balance the reconstruction loss and the KL divergence, and the total loss is expressed as $Loss = \frac{1}{\sigma^2}Loss_{rec} + Loss_{KL}$.

**Assimilation**   Once VAE is trained, we use its decoder to implement VAE-Var assimilation, as described in Algorithm 2. Similar to traditional algorithms, VAE-Var aims to optimize variables in the latent space, denoted by $\mathbf{z}$. The key difference is that the transformation from the latent space to the physical space is non-linear, i.e., $\mathbf{x} = \mathcal{D}(\mathbf{z}) + \mathbf{x}_b$, where $\mathcal{D}(\mathbf{z})$ represents the decoder output. As a result, the observation term is modified accordingly, expressed as $\tilde{\mathcal{L}}_o(\mathbf{z}) = \mathcal{L}_o(\mathcal{D}(\mathbf{z}) + \mathbf{x}_b, \mathbf{y})$. It is important to note that, from a rigorous mathematical perspective, when the mapping between the latent space and the physical space is nonlinear, the transformation of the probability density function introduces an additional non-constant determinant term in the background field, which is almost computationally intractable. To account for this, we empirically scale the original background term $\frac{1}{2}\mathbf{z}\mathbf{z}^{\mathrm{T}}$ by a positive parameter $\lambda$ for proper compensation. We provide a further explanation on the formulation of the background term in the Appendix. For any given observation $\mathbf{y}$, as long as the observation operator $\mathcal{H}$ is constructed in an auto-differentiable way, which will be explained in Section 4, then all components of $\tilde{\mathcal{L}}(\mathbf{z})$ are differentiable, and we can leverage auto-differentiation to perform back-propagation and compute the gradient $\frac{\partial \tilde{\mathcal{L}}(\mathbf{z})}{\partial \mathbf{z}}$, with the parameters of $\mathcal{D}$ fixed. Following this, the L-BFGS algorithm can be used to find the minimum of $\tilde{\mathcal{L}}(\mathbf{z})$.

---

**Algorithm 3** Cyclic Forecasting and Assimilation with VAE-Var

---

**Require:** : Initial state $\mathbf{x}_b^{(0)}$, forecasting model $\mathcal{M}$, observations, steps $L$, assimilation cycle $T$.
1:  $t \leftarrow 0$                                                  ▷ Initialize the time stamp
2:  **for** $step$ from 0 to $L$ **do**
3:     $\mathbf{z}_a^{(t)} \leftarrow \arg\min_{\mathbf{z}} \tilde{\mathcal{L}}\left(\mathbf{z} \middle| \mathbf{x}_b^{(t)}, \mathbf{y}^{(t)}\right)$           ▷ Obtain the analysis control variable
4:     $\mathbf{x}_a^{(t)} = \mathcal{D}(\mathbf{z}_a^{(t)}) + \mathbf{x}_b^{(t)}$                 ▷ Recover the analysis states.
5:     $\mathbf{x}_b^{(t+T)} \leftarrow \mathcal{M}_{t \to t+T}(\mathbf{x}_a^{(t)})$          ▷ Obtain the background at the next time step.
6:     $t \leftarrow t + T$
7:  **end for**

---

**Cyclic Forecasting and Assimilation**    VAE-Var and the weather forecasting model can be coupled to realize the cyclic forecasting and assimilation, similar to FengWu-4DVar (Xiao et al., 2024). Denoting $\tilde{\mathcal{L}}\left(\cdot | \mathbf{x}_b, \mathbf{y}\right)$ the objective function with respect to observations $\mathbf{y}$ and the background states $\mathbf{x}_b$, and using the superscript to represent time steps, the coupling of VAE-Var with model forecasts is implemented as shown in Algorithm 3.

## 4 RESULTS

### 4.1 EXPERIMENTAL SETUP

**Forecasting Model**    Our experiments utilize FengWu (Chen et al., 2023), a prominent data-driven global medium-range weather forecasting model. FengWu simulates five atmospheric variables (each across 13 pressure levels in the version we use) and four surface variables, leading to a total of 69 predictands. In this study, the atmospheric variables are geopotential (z), specific humidity (q), zonal wind component (u), meridional wind component (v), and air temperature (t); the 13 sub-variables across various vertical levels are denoted using abbreviations of their short names and pressure levels (e.g., z500 represents geopotential height at 500 hPa). The four surface variables are 2-meter temperature (t2m), 10-meter zonal wind component (u10), 10-meter meridional wind component (v10), and mean sea level pressure (mslp). We test a spatial resolution of $721 \times 1440$. Following the convention of the original paper (Chen et al., 2023), the six-hour forecasting model is trained using the ERA5 dataset from 1979 to 2015.

**VAE Structure and Training**    Our approach allows considerable flexibility in selecting the neural network structure for VAE. Rather than designing a new architecture from the ground up, we opted to adapt the well-established FengWu structure, which primarily consists of multiple "Swin Transformer v2" (Liu et al., 2022) networks and is proved to be capable of capturing complex spatial and temporal relationships. In our experiments, we employ two FengWu networks to construct the encoder and decoder, as shown in Figure 2. As for the encoder, the channel number of "Mean_i" and "Std_i" is set to 6 (the original channel number is 13) for $1 \le i \le 5$ and the channel number of "Mean_0" and "Std_0" is set to 2 (the original channel number is 4); this guarantees that the latent space has a smaller dimensionality than the original space. The decoder is almost symmetric to the encoder, with the key difference being that the input layer channel number is half of the encoder's output layer channel number. We use ERA5 reanalysis data from 1979 to 2015 to train the VAE model, and the loss weight $\sigma$ is set to 2.0.

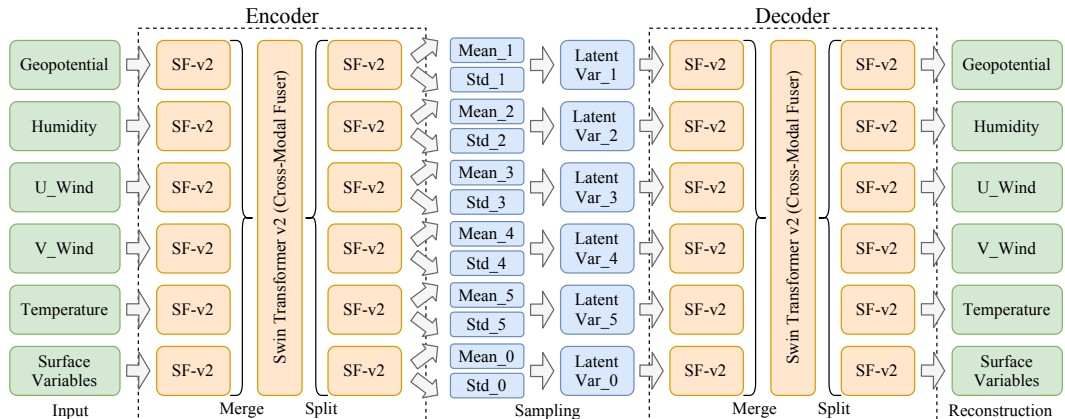

Figure 2: **VAE Structure for the FengWu forecasting system.** "SF-v2" is abbreviated for "Swin Transformer v2". Both the encoder and the decoder adopt the FengWu backbone structure.

**Observation Setup**    Two types of observations are considered in our experiments. The first type involves simulated observations from the ERA5 (Hersbach et al., 2020) dataset. Following Huang et al. (2024), we apply sparse grid column masks to mimic the sparse distribution of real-world observations. For instance, if we have 1000 observation columns, only 1000 points from the $721 \times$

1440 grid are observed, and when a grid point is observed, we have data for all variables at all height levels. The second type uses observations from the Global Data Assimilation System (GDAS) prepbufr dataset (Rodell et al., 2004), which provides data from various observing systems and instruments. In the second experiment, the observations may fall outside the grid points and are typically sparse across height levels, which closely resembles an operational scenario.

**Cyclic Forecasting and Assimilation Setup**   The initial state for our cyclic forecasting is derived from the ERA5 dataset. In our experiments, the forecasting system begins at 00:00 on January 1. To generate the background states $\mathbf{x}_0^b$, we begin with the ERA5 reanalysis field at 00:00 on December 30 of the previous year, run the 6-hour forecasting model for eight steps, and use the resulting fields of variables as the starting point for the cyclic forecasting process. The assimilation cycle $T$ equals six hours, consistent with the pre-trained six-hour Fengwu forecasting model.

**Baselines**   VAE-Var is compared against three baselines: DiffDA, interpolation and a traditional variational method (3DVar). For DiffDA, we draw from the results presented in the original paper (Huang et al., 2024); the interpolation method employs bilinear interpolation for station-based data; the 3DVar method is implemented following the approach outlined in Descombes et al. (2015).

## 4.2   ERA5-SIMULATED OBSERVATIONS

In the ERA5-simulated observational setting, all observations are positioned on the grid. Thus, the observation operator $\mathcal{H}$ is defined as a mask operator, where the values at the observation locations are set to one, and the values at locations without observations are set to zero. We implement $\mathcal{H}$ with element-wise matrix multiplication in PyTorch, and it is clearly differentiable. The observation covariance matrix $\mathbf{R}$ is assumed to be diagonal, with the square root of each entry set to 0.1 times the standard deviation of the respective variable. The parameter $\lambda$ is set to 4.0.

**Fixed Observations**   First, we examine the scenario where the observation positions remain constant over time. To align with DiffDA, we initiate our forecasting system on January 1, 2022, simulate it for 15 days, and conduct experiments with four different observation amounts: 1000, 2000, 4000, and 8000 columns. We calculate the root mean square errors (RMSE) of the analysis states at various time steps to assess assimilation accuracy. The results are illustrated in Figure 3, which indicate that VAE-Var consistently outperforms all three baseline methods, across all three demonstrated variables and the four observation amounts. Results for other important variables are left in the Appendix. Notably, when the observation amount is relatively low (1000 columns), VAE-Var demonstrates a marked advantage over the other two algorithms, highlighting its strong capability in leveraging background information. As the observation amount increases, VAE-Var maintains superior performance, albeit with a reduced margin, since the significance of effectively utilizing background information diminishes with more observations.

**Unfixed Observations**   Next, we conduct experiments where the observation positions vary over time, while keeping the observation amount constant. The results, presented in Figure 4, are consistent with the "fixed observation" case. VAE-Var continues to outperform DiffDA, the interpolation method and 3DVar across all demonstrated variables and all observational settings.

## 4.3   GDAS PREPBUFR OBSERVATIONS

In this experiment, we aim to assimilate real-world observations. The observation dataset we choose is the GDAS prepbufr dataset. This dataset consists of a global collection of surface and upper air reports gathered by the National Centers for Environmental Prediction (NCEP). It includes observations from land and marine surfaces, radiosondes, pibals, aircraft, and Global Telecommunications System (GTS) reports. It also incorporates data from wind profilers, U.S. radar-derived winds, SSM/I oceanic winds and TCW retrievals, as well as satellite wind data provided by the National Environmental Satellite Data and Information Service (NESDIS). We utilize this observational dataset because it is comprehensive and directly applied in today's operational weather forecasting systems.

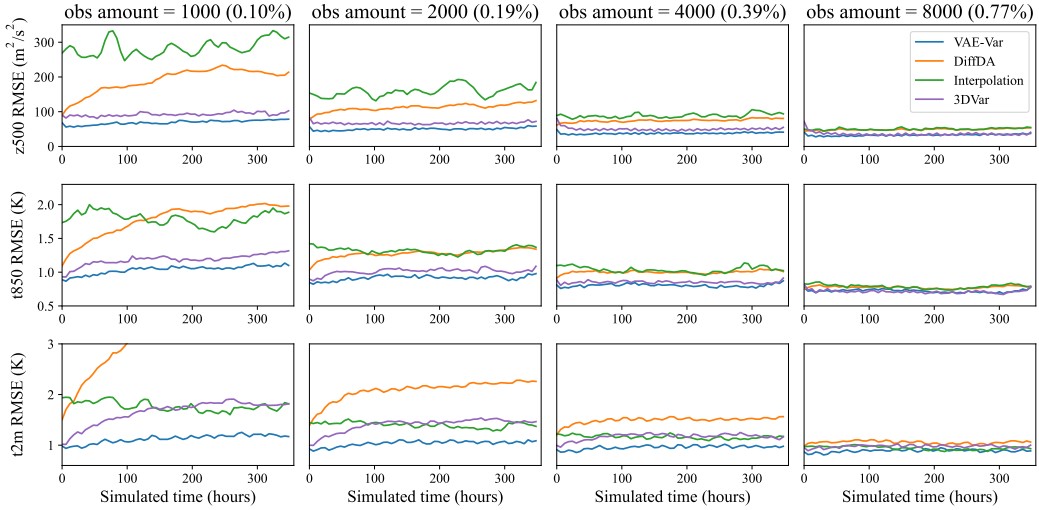

Figure 3: **Results for fixed observation positions (15 days).** The system is simulated in an auto-regressive manner for 15 days, starting from January 1, 2022. Following DiffDA, we demonstrate the RMSEs of the analysis field for three variables (z500, t850, t2m) in three different rows.

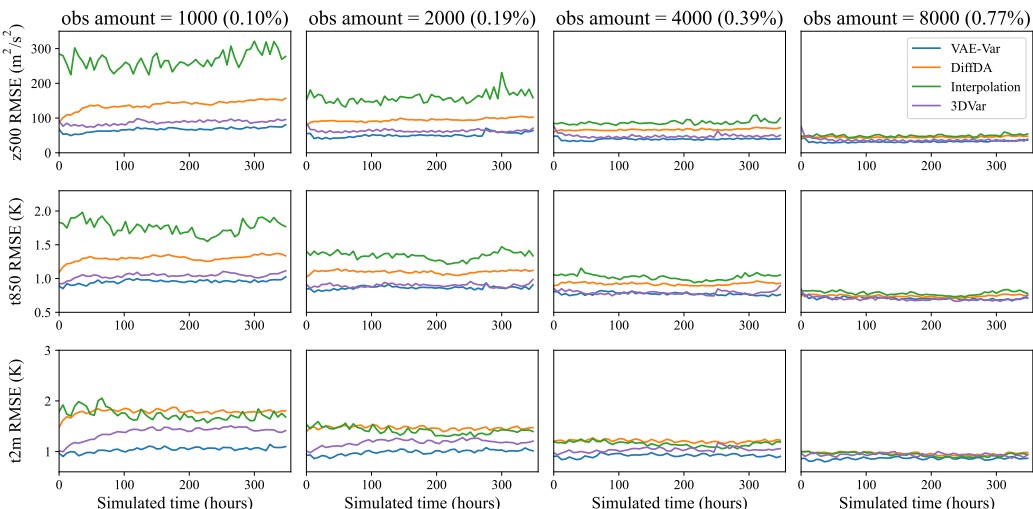

Figure 4: **Results for unfixed observation positions (15 days).** The system is simulated in an auto-regressive manner for 15 days, starting from January 1, 2022. Following DiffDA, we demonstrate the RMSEs of the analysis field for three variables (z500, t850, t2m) in three different rows.

**Data Processing**   In the GDAS prepbufr dataset, each message represents a single measurement taken by an observational instrument at a specific time and location. Since our assimilation algorithm focuses on a specific time point, we only select observations within half an hour of the assimilation time for processing. For instance, if the current time is 3 PM, we choose observations taken between 2:30 PM and 3:30 PM. Moreover, the spatial locations of observations are often not located on the grid of FengWu. To address this, we construct a new, finer grid for the observations and map the observations to the nearest grid points. Theoretically, this grid can be fine enough to minimize the error introduced by nearest-neighbor approximation. As a trade-off between accuracy and computational cost, the grid is designed to have the same latitude and longitude resolution as the FengWu physical grid, but with 40 vertical layers, which is much denser than FengWu. Also, we applied quality control after gridding the data, filtering out observations that deviate substantially from the corresponding ERA5 values.

**Observation Operator**   After data processing, all observations are placed on a grid with a finer resolution than FengWu. The observation operator $\mathcal{H}$ is then defined as the interpolation from the FengWu grid to the observation grid, which can be implemented using a differentiable 'Linear' layer in PyTorch. Details of this implementation are provided in the Appendix.

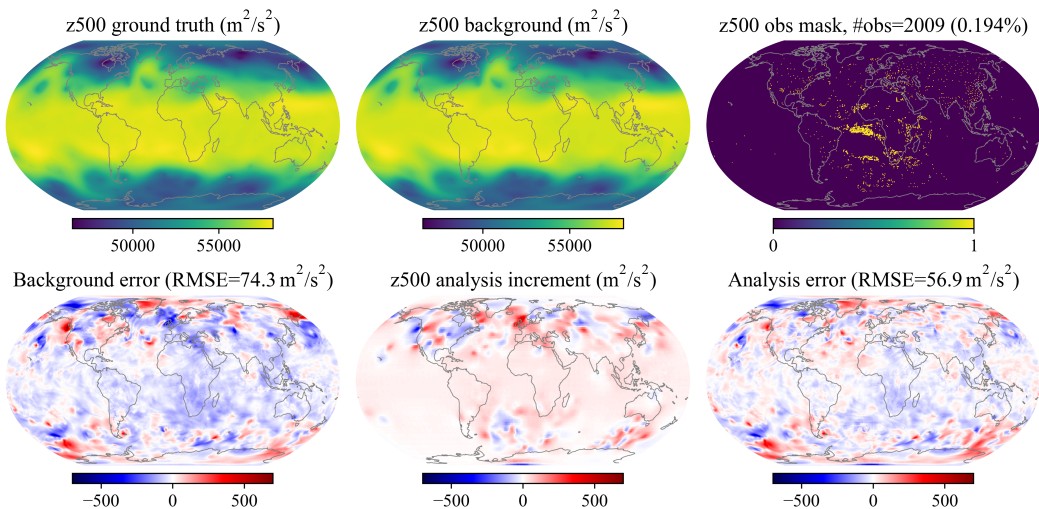

Figure 5: **Visualization of z500 assimilation results at time 2017-01-01 00:00:00.** ERA5 ground truth, the background field and the observation mask are demonstrated in the first row; the error of the background field (background field minus ground truth), the analysis increment (analysis field minus background field) and the error of the analysis field (analysis field minus ground truth) are shown in the second row. Strictly, the figure titled "z500 obs mask" correspond to observations at 501hPa because the observations use a different height axis. Please refer to the Appendix for details.

**Analysis Field Visualization**   We select the year 2017 for conducting the assimilation experiment because the completeness of the observational data is highest for that year. In Figure 5, we visualize the assimilation results at time 2017-01-01 00:00. At this time, there are a total of 375,371 observations, which are reduced to 142,647 (approximately 0.2% of the total grid points) after quality control. From the visualization of the observation mask, it is evident that the observations are primarily concentrated over Eurasia, North America, and the South Atlantic Ocean. The observations over the first two regions are mainly derived from radiosondes launched from the ground, while those over the South Atlantic come from retrievals based on geostationary satellite data. The analysis increment closely aligns with the observations, and in areas where the observations are more concentrated, the analysis increment is more pronounced. By comparing the errors between the analysis field and the background field, we observe a reduction in error after assimilation, which highlights the effectiveness of our method.

**Cyclic Forecasting and Assimilation Results**   Figure 6 presents the RMSE and Bias of the analysis fields at various time steps when VAE-Var is coupled with FengWu for a year-long simulation. In the GDAS prepbufr observational settings, it is common to encounter positions where only a subset of variables is observed. Since the DiffDA is currently unable to address such cases, we compare VAE-Var with the interpolation and 3DVar baselines in this section. The results show that for all eight reported variables, VAE-Var outperforms both methods in RMSE and Bias, demonstrating that our approach effectively reduces both error and systematic bias in the cyclic forecasting system.

### 4.4   COMPUTATIONAL COST

In VAE-Var, the linear transformation $\mathbf{U}$ in 3DVar is replaced with a nonlinear $\mathcal{D}$. The number of floating-point operations in $\mathcal{D}$ is no samller than in $\mathbf{U}$, so from a floating-point computation perspective, the computational complexity of VAE-Var is not lower than that of 3DVar. However, VAE-Var's main advantage is its easy implementation on GPUs, which significantly improves overall assimilation efficiency. For example, on a single A100 GPU, one cycle of assimilation takes approximately

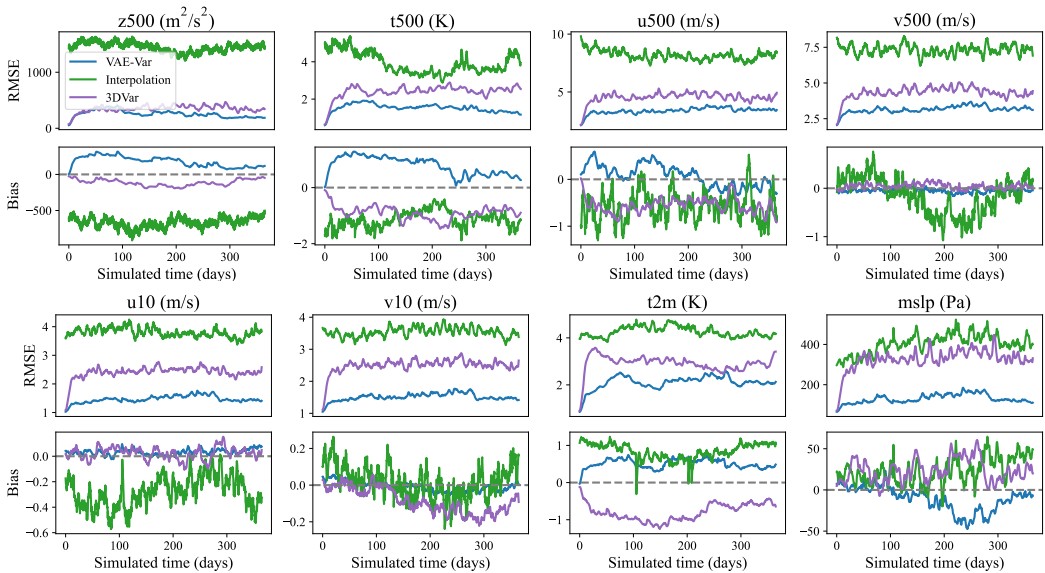

Figure 6: **Results for GDAS prepbufr observations (one year).** The system is simulated in an auto-regressive manner for one year, starting from January 1, 2017. The y-axis represents RMSE and Bias with respect to the ERA5 ground truth. Due to the varying amounts of observations over time, the RMSE for the interpolation method experiences significant fluctuations. The curves have been smoothed to provide better visualization.

18 seconds. In contrast, 3DVar is typically implemented on CPUs, where one cycle of assimilation takes several minutes (Smith et al., 2014). Therefore, in practice, VAE-Var can easily leverage GPU capabilities to achieve assimilation one order of magnitude faster than traditional 3DVar.

## 5 RELATED WORK

The concept behind our work is similar to "latent space data assimilation". For instance, in Amendola et al. (2020), an LSTM is used to map physical states into latent space, where observed data are assimilated using a Kalman Filter. Similarly, Peyron et al. (2021) and Fan et al. (2025) employ autoencoders to construct the latent space and have achieved promising results. In Melinc & Zaplotnik (2023), a VAE is trained for learning the map, with experiments conducted on the ERA5 dataset. However, the observational settings and forecasting models in these works are overly simplified and the effectiveness on more complicated systems remains to be evaluated.

While these approaches focus on finding the latent space of the physical fields, the critical aspect in data assimilation is not the physical fields themselves but the uncertainty (or error) of these fields. In contrast, the VAE-Var algorithm we propose directly learns the latent space of the error fields, offering a straightforward yet effective method for data assimilation. Moreover, our algorithm is proved effective on a global-scale AI forecasting model with real-world observations.

## 6 CONCLUSION

In this paper, we introduce a novel data assimilation algorithm, VAE-Var, which leverages a variational autoencoder to model the non-Gaussian structures of the background error distribution while utilizing the traditional variational methods for effective assimilation of real-world irregularly sampled observations. Experimental results on ERA5-simulated observations show that our method surpasses both an AI data assimilation algorithm, DiffDA, and two traditional methods (interpolation and 3DVar). Additionally, we conducted experiments using real-world observations, demonstrating that VAE-Var is capable of performing cyclic forecasting and assimilation over the span of one year. VAE-Var is also shown to have high computational efficiency on GPU devices.

ACKNOWLEDGEMENTS

This work is supported by the National Natural Science Foundation of China (NO.U2242210). This work is also supported by the Shanghai Science and Technology Commission Project (23DZ1204704) and Shanghai Artificial Intelligence Laboratory.

We sincerely acknowledge the European Centre for Medium-Range Weather Forecasts (ECMWF) for providing the ERA5 dataset and the National Center for Environmental Prediction (NCEP) for providing the GDAS dataset, which are instrumental in this study. Their efforts in data collection, archiving, and dissemination are greatly appreciated.

We are also grateful to the Research Support, IT, and Infrastructure team at the Shanghai AI Laboratory for their provision of computational resources and network support.

Additionally, we extend our heartfelt thanks to Dr. Ben Fei from the Chinese University of Hong Kong and Hang Fan from Shanghai AI Laboratory for their valuable assistance and insightful discussions throughout this research. Their contributions have greatly enriched this work.

REPRODUCIBILITY STATEMENT

All experiments in this paper are fully reproducible. The code of VAE-Var is available at https://github.com/xiaoyi018/VAE-Var. For the baseline algorithms, three baselines are considered: DiffDA, 3DVar, and interpolation. Regarding DiffDA, we directly adopt the results from the original paper. For 3DVar, the construction details of the background error covariance matrix are provided in the Appendix. As for the interpolation method, we clarify that it is based on the bilinear interpolation algorithm. Additionally, all the datasets we use, including ERA5 and GDAS prepbufr, are available online.

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

# A APPENDIX

## A.1 DERIVATION OF EQUATION 1

We aim to maximize $p(\mathbf{x}|\mathbf{x}_b, \mathbf{y})$, which can be rewritten as:

$$\arg\max_{\mathbf{x}} p(\mathbf{x}|\mathbf{x}_b, \mathbf{y}) = \arg\max_{\mathbf{x}} \frac{p(\mathbf{x}_b, \mathbf{y}|\mathbf{x})p(\mathbf{x})}{p(\mathbf{x}_b, \mathbf{y})} = \arg\max_{\mathbf{x}} p(\mathbf{x}_b, \mathbf{y}|\mathbf{x})p(\mathbf{x}).$$

Here, $p(\mathbf{x}_b, \mathbf{y})$ is independent of $\mathbf{x}$ and thus can be omitted from the optimization.

Now, assuming that $\mathbf{x}_b$ and $\mathbf{y}$ are independent conditional on $\mathbf{x}$, we can decompose $p(\mathbf{x}_b, \mathbf{y}|\mathbf{x})$ as $p(\mathbf{x}_b|\mathbf{x})p(\mathbf{y}|\mathbf{x})$. Substituting this into the equation gives:

$$\arg\max_{\mathbf{x}} p(\mathbf{x}|\mathbf{x}_b, \mathbf{y}) = \arg\max_{\mathbf{x}} p(\mathbf{x}_b|\mathbf{x})p(\mathbf{y}|\mathbf{x})p(\mathbf{x}).$$

Next, applying Bayes' rule to $p(\mathbf{x}_b|\mathbf{x})$, we have $p(\mathbf{x}_b|\mathbf{x}) = p(\mathbf{x}|\mathbf{x}_b)p(\mathbf{x}_b)/p(\mathbf{x})$. Substituting this back, and noting that $p(\mathbf{x}_b)$ is constant with respect to $\mathbf{x}$, we obtain:

$$\arg\max_{\mathbf{x}} p(\mathbf{x}|\mathbf{x}_b, \mathbf{y}) = \arg\max_{\mathbf{x}} p(\mathbf{x}|\mathbf{x}_b)p(\mathbf{y}|\mathbf{x}).$$

This is the right-hand side of Equation 1, which follows logically under the given assumptions.

## A.2 NON-GAUSSIAN FEATURES OF THE BACKGROUND ERROR

We use the Lorenz 63 (Lorenz, 1963) system to provide an illustrative demonstration of the background error's non-Gaussian features. This system involves three parameters: $\sigma$, $\rho$, $\beta$, as shown in Equation 5. We create dynamical models using two different sets of parameter values: $\sigma = 10, \rho = 28, \beta = \frac{8}{3}$ (for ground truth model) and $\sigma = 10, \rho = 29, \beta = \frac{8}{3}$ (for prediction model). By numerically integrating these models from randomly-chosen identical initial states and calculating the difference between their outputs, we generate a set of error samples, as depicted with the blue dots in Figure 7. The error distribution is observed to be distinctly non-Gaussian and exhibits a non-convex structure.

$$\begin{cases} \dfrac{\mathrm{d}X}{\mathrm{d}t} = \sigma(Y - X) \\ \dfrac{\mathrm{d}Y}{\mathrm{d}t} = X(\rho - Z) - Y \\ \dfrac{\mathrm{d}Z}{\mathrm{d}t} = XY - \beta Z \end{cases} \quad (5)$$

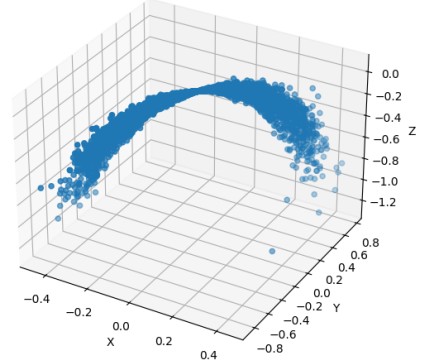

Figure 7: Distribution of the background error for the Lorenz 63 system.

## A.3 MATHEMATICAL EXPLANATION OF VARIATIONAL COST

Although our method is generally heuristic, a mathematical explanation can be provided under some assumptions. Suppose that the VAE is well learnt and the decoder is a bijection map from the low-dimensional latent space to a manifold of the high-dimensional physical space. In this case, according to the change of variables formula in Theorem (2.80) of Giaquinta & Modica (2010)

(Chapter 2.5.2), supposing that the physical space has a dimensionality of $d_x$, and the latent space has a dimensionality of $d_z$, then for any measurable set $A \subseteq \mathbb{R}^{d_z}$, the following equation holds:

$$\int_A p_\mathbf{x}(\mathcal{D}(\mathbf{z})) J\left(\mathbf{D}\mathcal{D}\right)(\mathbf{z})\, \mathrm{d}\mathbf{z} = \int_{\mathbb{R}^{d_x}} p_\mathbf{x}(\mathbf{x}) \mathcal{H}^0\left(A \cap \mathcal{D}^{-1}(\{\mathbf{x}\})\right) \mathrm{d}\mathcal{H}^{d_z}(\mathbf{x}), \quad (6)$$

where $p_\mathbf{x}$ correspond to the probability density function of $\mathbf{x}$, $\mathcal{D}^{-1}(\{\mathbf{x}\})$ represents the preimage of the set under the mapping $\mathcal{D}$, $\mathcal{H}^s$ corresponds to the Hausdorff $s$-dimensional measure, and $J(\mathbf{D}f)(\mathbf{x})$ is defined as:

$$J(\mathbf{D}f)(\mathbf{x}) = \sqrt{\det\left(\frac{\partial f(\mathbf{x})}{\partial \mathbf{x}}\right)^{\mathrm{T}} \left(\frac{\partial f(\mathbf{x})}{\partial \mathbf{x}}\right)}. \quad (7)$$

Intuitively, the same event should have the same probability measured in two different space. Denoting $p_\mathbf{z}$ the probability density function of $\mathbf{z}$, the following "equation" holds:

$$p_\mathbf{x}(\mathbf{x})\, \mathrm{d}\mathcal{H}^{d_z}(\mathbf{x}) = p_\mathbf{z}(\mathbf{z})\, \mathrm{d}\mathbf{z}. \quad (8)$$

Combining these two equations above, we have:

$$\int_A p_\mathbf{x}(\mathcal{D}(\mathbf{z})) J\left(\mathbf{D}\mathcal{D}\right)(\mathbf{z})\, \mathrm{d}\mathbf{z} = \int_A p_\mathbf{z}(\mathbf{z}) \mathcal{H}^0\left(A \cap \{\mathbf{z}\}\right) \mathrm{d}\mathbf{z}. \quad (9)$$

It is apparent that for $\mathbf{z} \in A$, $\mathcal{H}^0\left(A \cap \{\mathbf{z}\}\right) = 1$ holds; thus,

$$\int_A p_\mathbf{x}(\mathcal{D}(\mathbf{z})) J\left(\mathbf{D}\mathcal{D}\right)(\mathbf{z})\, \mathrm{d}\mathbf{z} = \int_A p_\mathbf{z}(\mathbf{z})\, \mathrm{d}\mathbf{z}. \quad (10)$$

Since this equation holds for any measurable set $A$, we have

$$p_\mathbf{x}(\mathcal{D}(\mathbf{z})) J\left(\mathbf{D}\mathcal{D}\right)(\mathbf{z}) = p_\mathbf{z}(\mathbf{z}). \quad (11)$$

Denoting $\delta = \mathbf{x} - \mathbf{x}_b$, $p_\delta$ the probability density function of $\delta$; then

$$p_\delta(\delta) = p_\mathbf{x}(\mathbf{x}) \left|\det \frac{\partial \mathbf{x}}{\partial \delta}\right| = p_\mathbf{x}(\mathbf{x}). \quad (12)$$

Further, we have:

$$\begin{aligned} p_\delta(\delta) &= p_\mathbf{z}(\mathbf{z}) \left[J\left(\mathbf{D}\mathcal{D}\right)(\mathbf{z})\right]^{-1} \\ &= p_\mathbf{z}(\mathbf{z}) \left(\det\left(\frac{\partial \mathcal{D}(\mathbf{z})}{\partial \mathbf{z}}\right)^{\mathrm{T}} \left(\frac{\partial \mathcal{D}(\mathbf{z})}{\partial \mathbf{z}}\right)\right)^{-1/2}. \end{aligned} \quad (13)$$

Since, the latent space of VAE is assumed to follow the standard Gaussian distribution, therefore, $p(\mathbf{z}) \sim \mathcal{N}(\mathbf{0}, \mathbf{I})$. Then,

$$\begin{aligned} \tilde{\mathcal{L}}_b(\mathbf{z}) = -\log p(\mathbf{x}|\mathbf{x}_b) = -\log p_\delta(\delta) &= -\log p_\mathbf{z}(\mathbf{z}) + \frac{1}{2}\log\det\left(\frac{\partial \mathcal{D}(\mathbf{z})}{\partial \mathbf{z}}\right)^{\mathrm{T}} \left(\frac{\partial \mathcal{D}(\mathbf{z})}{\partial \mathbf{z}}\right) \\ &= \frac{1}{2}\mathbf{z}^{\mathrm{T}}\mathbf{z} + \frac{1}{2}\log\det\left(\frac{\partial \mathcal{D}(\mathbf{z})}{\partial \mathbf{z}}\right)^{\mathrm{T}} \left(\frac{\partial \mathcal{D}(\mathbf{z})}{\partial \mathbf{z}}\right). \end{aligned} \quad (14)$$

Compared to the formula used in the paper, the strict expression includes an additional determinant term, which represents the scaling factor of the volume element when mapping from the latent space to the physical space.

**Simplification** Since Transformer-based neural networks are highly complex, nonlinear, and high-dimensional systems, this scaling factor can fluctuate unpredictably over time. Including this term would make the assimilation objective function extremely difficult to optimize. For this reason, we empirically omitted the determinant term from our formula. As a compensatory measure, we introduced an adjustable parameter $\lambda$ in the background term of the assimilation objective function, that is, $\mathcal{L}_b(\mathbf{z}) = \frac{1}{2}\lambda\mathbf{z}^{\mathrm{T}}\mathbf{z}$. Since the observation deviation term $\mathcal{L}_o\left(\mathcal{D}(\mathbf{z}) + \mathbf{x}_b, \mathbf{y}\right)$ is calculated in

the physical space, and the background error term $\mathcal{L}_b(\mathbf{z}) = \frac{1}{2}\mathbf{z}^T\mathbf{z}$ is calculated in the latent space, the former would intuitively be $\left(\frac{d_x}{d_z}\right)^2$ times larger than the latter. To balance this dimensionality difference, the parameter $\lambda$ should be approximately equal to $\left(\frac{d_x}{d_z}\right)^2$. In our experimental setup, the physical space has dimensions $d_x = 69 \times 721 \times 1440$ and the latent space has dimensions $d_z = 32 \times 721 \times 1440$. This gives $\lambda \approx 4.64$, and we round this value to 4 for simplicity.

We admit that such a simplification is not mathematically rigorous, but it greatly reduces the optimization difficulty and is experimentally proven to be effective.

**Rigorous Consideration of Jacobian Determinant**  In Bayesian modeling, one class of approaches rigorously addressing the Jacobian determinant of the mapping $\mathcal{D}$ from a Gaussian prior to the target dataset distribution is normalizing flow (Rezende & Mohamed, 2015). These methods constrain the structure of $\mathcal{D}$ to ensure the Jacobian determinant is computationally tractable. Typical constraints include the use of $1 \times 1$ convolution layers, single-variable activation functions, or invertible transformations with simple Jacobian structures. For instance, Glow (Kingma & Dhariwal, 2018) is a well-known example in this family of methods.

While normalizing flow approaches retain the universal approximation property, they face significant challenges when applied to high-dimensional systems, such as weather modeling. Specifically, the structural constraints lead to substantial computational complexity, making training time-intensive and resource-demanding. To our knowledge, few success has been achieved in modeling high-dimensional systems using normalizing flow.

In future work, we will explore incorporating the advantages of normalizing flows into VAE-Var, with a focus on addressing the scalability and efficiency challenges in high-dimensional meteorological systems.

### A.4  GDAS PREBUFR OBSERVATION HEIGHTS

The potential height of the observation coordinate axis is evenly distributed logarithmically from 50 hPa to 1000 hPa. Specifically, the coordinate axis includes the following 40 potential height levels:

50 hPa, 54 hPa, 58 hPa, 63 hPa, 68 hPa, 73 hPa, 79 hPa, 86 hPa, 92 hPa, 100 hPa, 108 hPa, 116 hPa, 126 hPa, 136 hPa, 147 hPa, 158 hPa, 171 hPa, 185 hPa, 199 hPa, 215 hPa, 232 hPa, 251 hPa, 271 hPa, 293 hPa, 316 hPa, 341 hPa, 368 hPa, 398 hPa, 430 hPa, 464 hPa, 501 hPa, 541 hPa, 584 hPa, 631 hPa, 681 hPa, 735 hPa, 794 hPa, 858 hPa, 926 hPa, 1000 hPa

In contrast, the potential height axis of the physical space coordinate axis of FengWu includes the following 13 levels:

50 hPa, 100 hPa, 150 hPa, 200 hPa, 250 hPa, 300 hPa, 400 hPa, 500 hPa, 600 hPa, 700 hPa, 850 hPa, 925 hPa, 1000 hPa

### A.5  IMPLEMENTATION OF THE 3DVAR BASELINE

The implementation of the 3DVar baseline follows the GEN_BE method from Descombes et al. (2015) and we reproduce it with Python. Generally speaking, GEN_BE consists of three steps. First, 1000 ERA5 fields are sampled between 1979 and 2015; then the NMC method is applied to calculate the resulting background error samples; last, calculate the statistics of $\mathbf{U}$. In GEN_BE, $\mathbf{U}$ is assumed to be decomposed into four operators, $\mathbf{U} = \mathbf{U}_p\mathbf{S}\mathbf{U}_v\mathbf{U}_h$, where $\mathbf{U}_p$ handles the transformation of physical variables, $\mathbf{S}$ represents the standard deviation, $\mathbf{U}_v$ corresponds to vertical transformation, and $\mathbf{U}_h$ handles horizontal transformation.

The details for constructing the four operators are as follows:

**Implementation of $\mathbf{U}_p$**  $\mathbf{U}_p$ is the only transformation that does not require parameters derived from the NMC samples. It is used to convert the stream function and potential function ($\phi$, $\psi$) into wind components ($u$, $v$) in the latitude-longitude directions, while keeping other variables un-

changed. Specifically, it is computed using the following equations:

$$u = \frac{\partial \psi}{\partial y} + \frac{\partial \phi}{\partial x}, \ v = -\frac{\partial \psi}{\partial x} + \frac{\partial \phi}{\partial y}$$

**Implementation of S**  $\ \mathbf{S}$ represents the error variance for each variable layer. To calculate it, we first apply the inverse transformation of $\mathbf{U}_p$ to all error samples (i.e., converting winds from the latitude-longitude direction back to stream functions and potential functions). This is done using spectral methods, which are supported by the 'torch-harmonic' library in PyTorch for solving the differential equations. After transforming the fields, we compute the sample variance at each pixel. The resulting variances are averaged in the latitude-longitude direction, yielding 69 values corresponding to the error variance at each vertical layer.

**Implementation of $\mathbf{U}_v$**  $\ \mathbf{U}_v$ is computed using eigenvalue decomposition. Specifically, for each upper-air variable, we first average the variable in the latitude-longitude direction to obtain 1000 69-length vectors. The covariance matrix $\mathbf{A}$ of these 1000 vectors is then computed. We perform eigen-decomposition on $\mathbf{A}$ such that $\mathbf{A} = \mathbf{P}\mathbf{\Lambda}\mathbf{P}^{-1}$, and $\mathbf{U}_v$ is then defined as $\mathbf{U}_v = \mathbf{P}\mathbf{\Lambda}^{1/2}$, where $\mathbf{P}$ is the eigenvector matrix and $\mathbf{\Lambda}$ is the diagonal matrix of eigenvalues.

**Implementation of $\mathbf{U}_h$**  $\ \mathbf{U}_h$ is essentially a horizontal recursive filter applied to each of the 69 layers. The length scale $L$ for the filter is given by:

$$L = \left( \frac{8 \cdot \text{Variance}(\phi)}{\text{Variance}(\nabla^2 \phi)} \right)^{1/4}$$

This filter is applied to each layer individually to perform horizontal smoothing.

To summarize, $\mathbf{U}_p$ is a fixed, parameter-free variable transformation operator that converts stream and potential functions into wind components; $\mathbf{S}$ represents error variance with 69 parameters corresponding to each vertical layer; $\mathbf{U}_v$ corresponds to eigenvalue decomposition applied to the 1000 samples, yielding 69*69 + 69 parameters; $\mathbf{U}_h$ is a horizontal recursive filter applied to each of the 69 layers, with 69 parameters.

## A.6 IMPLEMENTATION OF THE OBSERVATION OPERATOR

As for the real-world observations, the observation operator aims to do vertical interpolation from the physical grids to the observational grids. Since the geopotential height changes approximately exponentially with physical height, logarithmic linear interpolation is required when performing vertical layer interpolation. The computation of the interpolation matrix $P \in \mathbb{R}^{40 \times 13}$ is shown in Algorithm 4, where $X$ represents the scale of the geopotential height at the physical grid points and $Y$ represents that at the observational grid points. After calculating the interpolation matrix, the PyTorch construction of the observation operator is shown in Algorithm 5.

---

**Algorithm 4** Logarithmic Interpolation Matrix Construction

---

**Require:** : Vectors $X$ (scale of the vertical axis of the physical space) and $Y$ (scale of the vertical axis of the observational space)

    $P \leftarrow \text{zeros}(\text{len}(Y), \text{len}(X))$
    **for** $i = 0$ to $\text{len}(Y) - 1$ **do**
        **for** $j = 0$ to $\text{len}(X) - 1$ **do**
            **if** $Y[i] = X[j]$ **then**
                $P[i, j] \leftarrow 1$
            **else if** $Y[i] > X[j]$ and $Y[i] < X[j + 1]$ **then**
                $P[i, j] \leftarrow \frac{\log(X[j+1]) - \log(Y[i])}{\log(X[j+1]) - \log(X[j])}$
                $P[i, j + 1] \leftarrow \frac{\log(Y[i]) - \log(X[j])}{\log(X[j+1]) - \log(X[j])}$
            **end if**
        **end for**
    **end for**
    **return** $P$

---

---

**Algorithm 5** Observation Operator Implementation in PyTorch

---

**Require:** : Physical layer $x \in \mathbb{R}^{69 \times 721 \times 1440}$, interpolation matrix $P$
  $x_0 \leftarrow x.\text{unsqueeze}(0)$
  $y \leftarrow []$
  $y.\text{append}(x_0[:, :4])$
  **for** $i = 0$ to $5$ **do**
    $mat \leftarrow x_0[:, 4 + i \times 13 : 4 + (i+1) \times 13]$
    $mat \leftarrow \text{torch.nn.functional.linear}(mat.transpose(1,3), P).transpose(1,3)$
    $y.\text{append}(mat)$
  **end for**
  $y \leftarrow \text{torch.cat}(y, 1)$
  $y \leftarrow y.\text{squeeze}(0)$
  **return** $y$

---

### A.7 RELATIONSHIP WITH SPATIO-TEMPORAL DATA ASSIMILATION METHODS

The framework of VAE-Var is compatible with traditional spatio-temporal data assimilation methods, such as 4DVar. In 4DVar, the observation term can be expressed as

$$\mathcal{L}_o\left(\mathbf{x}, \{\mathbf{y}_i\}_{i=0}^{n-1}\right) = \frac{1}{2} \sum_{i=0}^{N-1} \left(\mathbf{y}_i - \mathcal{M}_{0 \to i}\left(\mathbf{x}\right)\right)^\mathrm{T} \mathbf{R}^{-1} \left(\mathbf{y}_i - \mathcal{M}_{0 \to i}\left(\mathbf{x}\right)\right), \tag{15}$$

where $\{\mathbf{y}_i\}_{i=0}^{N-1}$ are observations within an assimilation window and $\mathcal{M}_{0 \to i}$ are forecasting models for representing the flow dependency. As long as the forecasting models are auto-differentiable (e.g., implemented by neural networks), VAE-Var can be easily extended to VAE-Var by substituting the original $\mathcal{L}_o$ in Algorithm 2 with the new expression above. Experiments have not been conducted due to the limitation of computational resources and detailed analysis on VAE-Var's ability of doing spatio-temporal assimilation will be conducted in our future work.

### A.8 ABLATION STUDY

**Different Initial States** To investigate how the change of initial state $\mathbf{x}_b^{(0)}$ would affect the performance of VAE-Var, we construct seven initial fields to conduct the ablation study:

1. Start from ERA5 24 hours earlier and use four 6-hour FengWu forecasts as the initial field.
2. Start from ERA5 48 hours earlier and use eight 6-hour FengWu forecasts as the initial field.
3. Start from ERA5 192 hours earlier and use thirty-two 6-hour FengWu forecasts as the initial field.
4. Directly use ERA5 fields from 24 hours earlier as the initial field.
5. Directly use ERA5 fields from 48 hours earlier as the initial field.
6. Directly use ERA5 fields from 192 hours earlier as the initial field.
7. Use ERA5 fields from half a year earlier as the initial field.

These seven settings are denoted as "init-1", "init-2", ..., "init-7", respectively. All the other experimental settings are consistent with the unfixed ERA5-simulated observation experiments in the main paper. The cyclic forecasting and assimilation is performed for 25 days and the results of RMSE are demonstrated in Figure 8. It can be found that if the initial field is not derived from FengWu forecast (init-4, init-5, init-6, init-7), the assimilation process can still continuously reduce errors over cycles, eventually achieving comparable performance. Even when the initial field is significantly different from the ground truth, e.g., with a half-year phase (init-7), z500 still converges after about 20 days of assimilation. This ablation study proves that VAE-Var is robust to initial conditions.

**Different Observation Quantities** To investigate the impact of how observation quantity affects the performance of VAE-Var, we added experiments with 100, 200, 400, 800, 10,000, 20,000, and 40,000 observation columns (the largest density corresponding to approximately one observation per grid point on a global 1.25-degree latitude-longitude grid).

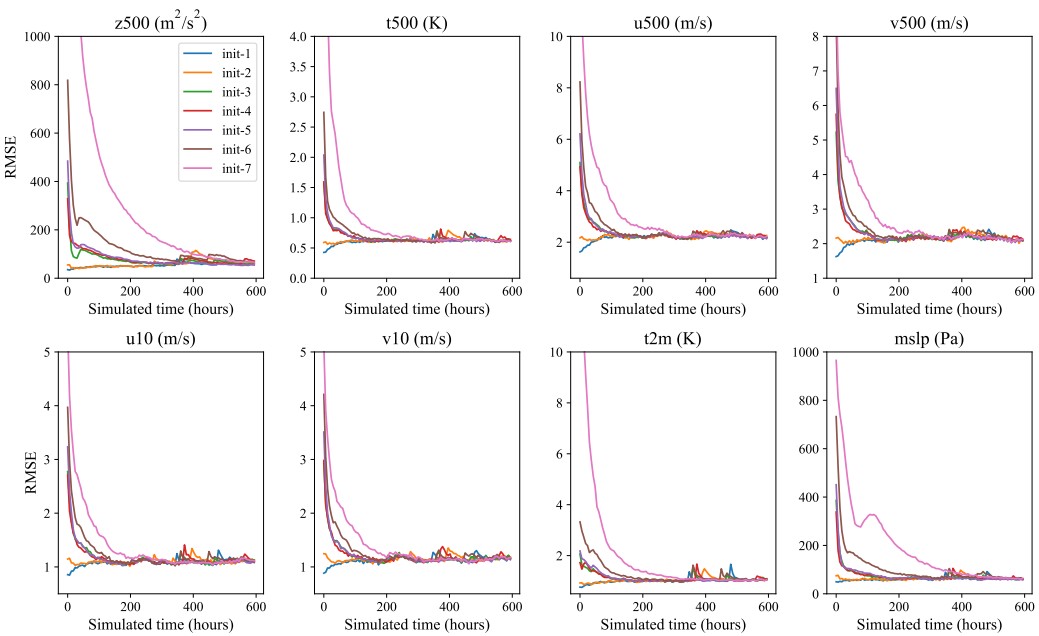

Figure 8: **Results for different initial states.** The observations are unfixed in different time steps. The system is simulated for 25 days, starting from January 1, 2022.

The experiments use a 15-day assimilation cycle from January 1 to January 15, 2022, with different observation locations at each time step but a consistent number of observations. The results for observation quantities from 100 to 8000 are demonstrated in Figure 9 and those from 8000 to 40000 are demonstrated in Figure 10. It can be found that when the number of observations is low (below 8000), VAE-Var shows a clear improvement in analysis accuracy as the observation quantity increases. However, when the observation density exceeds 8000, the improvement in accuracy becomes less pronounced. In contrast, 3DVar demonstrates more consistent improvement as the observation count increases. When the number of observations is below 10,000, VAE-Var generally outperforms 3DVar, with the advantage becoming more pronounced as the observations are sparser. However, when the number of observations exceeds 10,000, the performance of VAE-Var slightly lags behind 3DVar.

Regarding why the improvement in analysis accuracy with higher observation density is less pronounced for VAE-Var, we think this is due to the fact that the increments in the analysis field are constrained within the latent space defined by the decoder output, $\mathcal{D}(\mathbf{z})$. During training, VAE may not fully generalize to the complete analysis increment space, meaning that certain finer details of the analysis may not be captured effectively. When observation density is high, achieving further improvement requires fine-grained adjustments to the analysis increments, which may not be well-represented in the latent space. As a result, the analysis accuracy may plateau once the background field error is already small. Currently, the RMSE for the z500 field has a lower bound of approximately 26 $\text{m}^2/\text{s}^2$, which represents a good level of accuracy, but further improvements may be limited due to these modeling constraints.

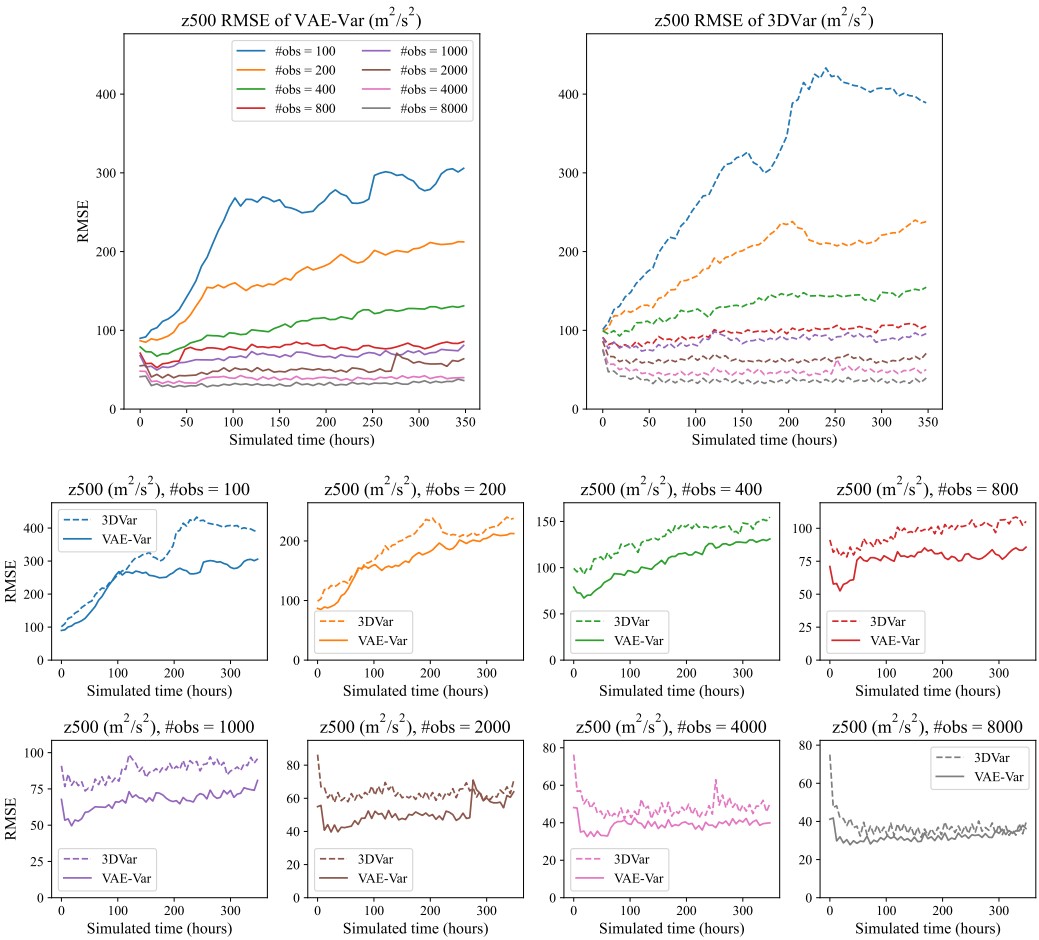

Figure 9: **Results for different observation quantities ($100 \leq$ #obs $\leq 8000$).** The observations are unfixed in different time steps. The system is simulated for 15 days, starting from January 1, 2022.

## A.9 EXPERIMENTS EVALUATED BASED ON GDAS OBSERVATIONS

Another important evaluation method in the field of data assimilation is to treat the observation sites as ground truth for assessment. In this method, all observations are first divided into two parts: one part is used for assimilation, and the other part is reserved as ground truth to evaluate the quality of the assimilated analysis field. In this section, we also adopt this method to assess the assimilation performance of VAE-Var. Specifically, we select the observation sites within the region bounded by 50°E to 150°E longitude and 15°N to 40°N latitude (with the main area covering East Asia). During the assimilation process, we do not assimilate these observations, but use them as truth for evaluation.

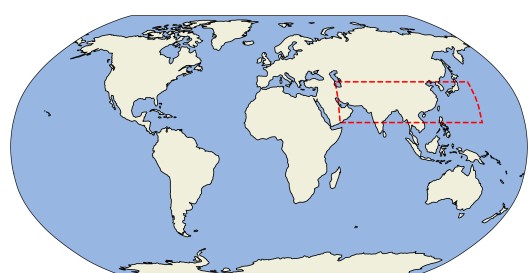

Figure 11: **Division of observations.** The region marked by the red dashed box represents the observation area reserved for evaluation.

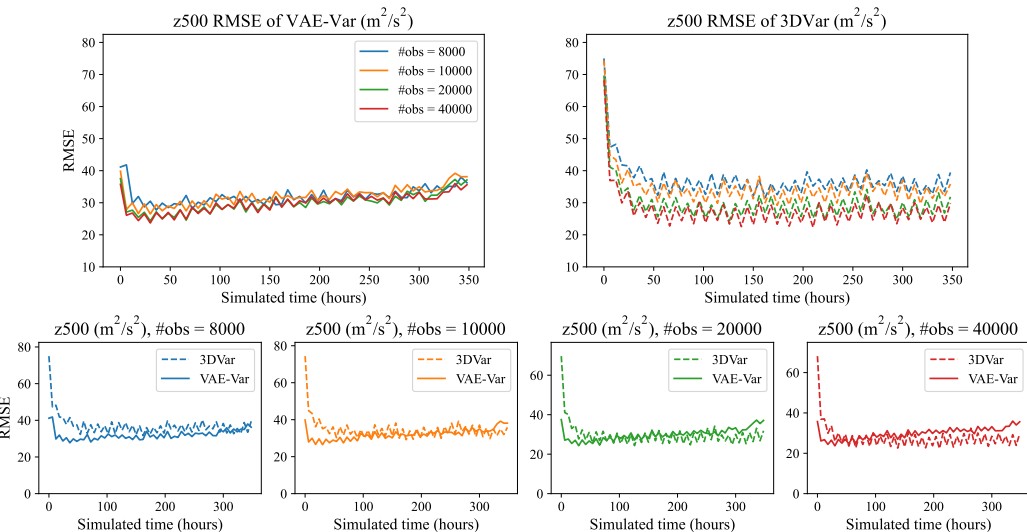

Figure 10: **Results for different observation quantities (8000 ≤ #obs ≤ 40000).** The observations are unfixed in different time steps. The system is simulated for 15 days, starting from January 1, 2022.

Starting from January 1, 2017, the cyclic forecasting and assimilation system is simulated for one year and the results are shown in Figure 12, 13, 14. It is shown that under the evaluation of GDAS observations, VAE-Var still consistently outperforms 3DVar on most variables.

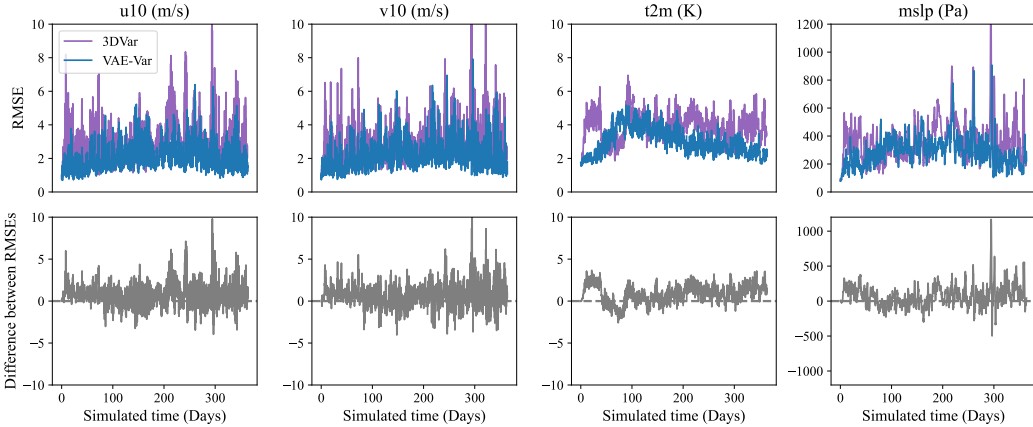

Figure 12: **Evaluation results based on GDAS observations. Surface variables (u10, v10, t2m, mslp) are reported.** The system is simulated in an auto-regressive manner for one year, starting from January 1, 2017. The upper row demonstrates the RMSEs between the analysis fields and the GDAS observations. The RMSEs are averaged among all the observations available at a given layer. For example, to calculate the RMSE of z501 at 00:00, January 1, 2017, we first interpolate the analysis field to the height of 501hPa; then, calculate the squared error between the value an observation at the height 501hPa value and the value of the analysis field at that point; by averaging all these observations and taking the square root, we can obtain the RMSE. The lower row shows the value of the RMSE of 3DVar minus that of VAE-Var. A value larger than zero indicates that VAE-Var is better than 3DVar.

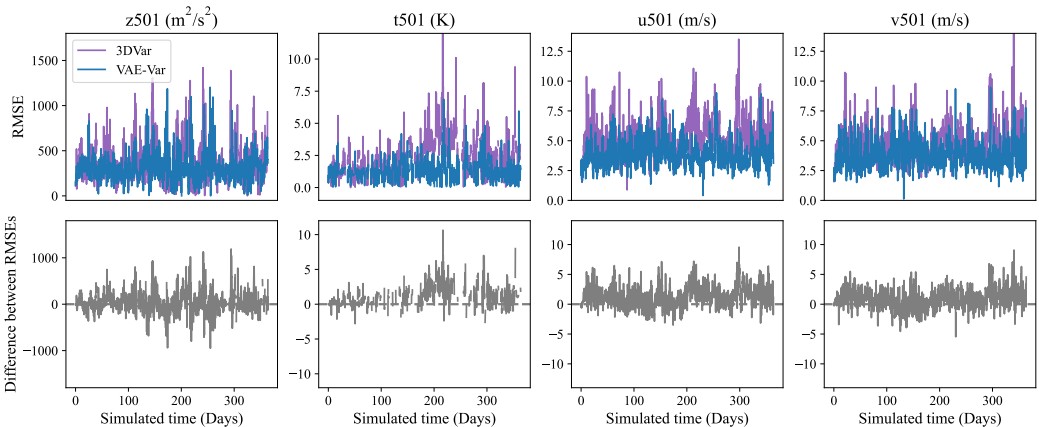

Figure 13: **Evaluation results based on GDAS observations. z501, t501, u501 and v501 are reported.** Please refer to the caption of Figure 12 for details.

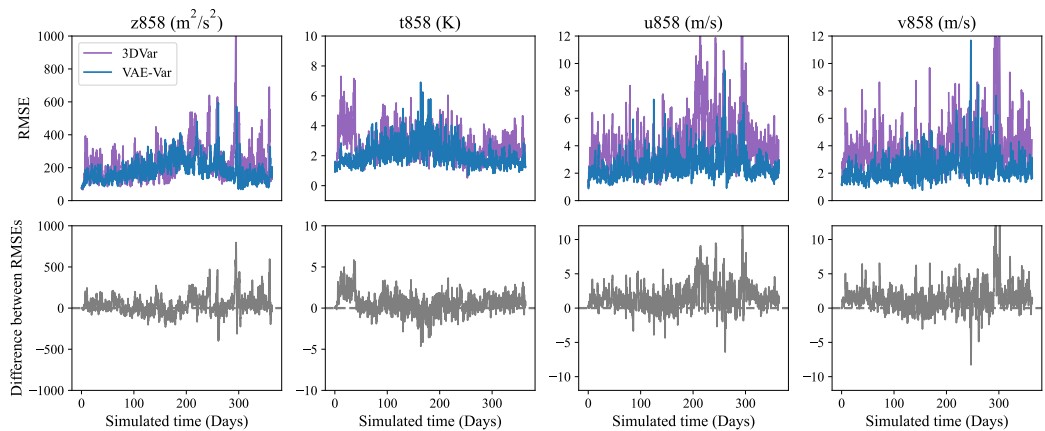

Figure 14: **Evaluation results based on GDAS observations. z858, t858, u858 and v858 are reported.** Please refer to the caption of Figure 12 for details.

### A.10 ADDITIONAL RESULTS FOR EXPERIMENTS IN THE MAIN PAPER

Results for other important variables in the fixed-position ERA5-simulated observational settings are demonstrated in Figure 15-17 and those in the unfixed-position are demonstrated in Figure 18-20. The fields of other important variables for the GDAS prepbufr observations are visualized in Figure 21-27.

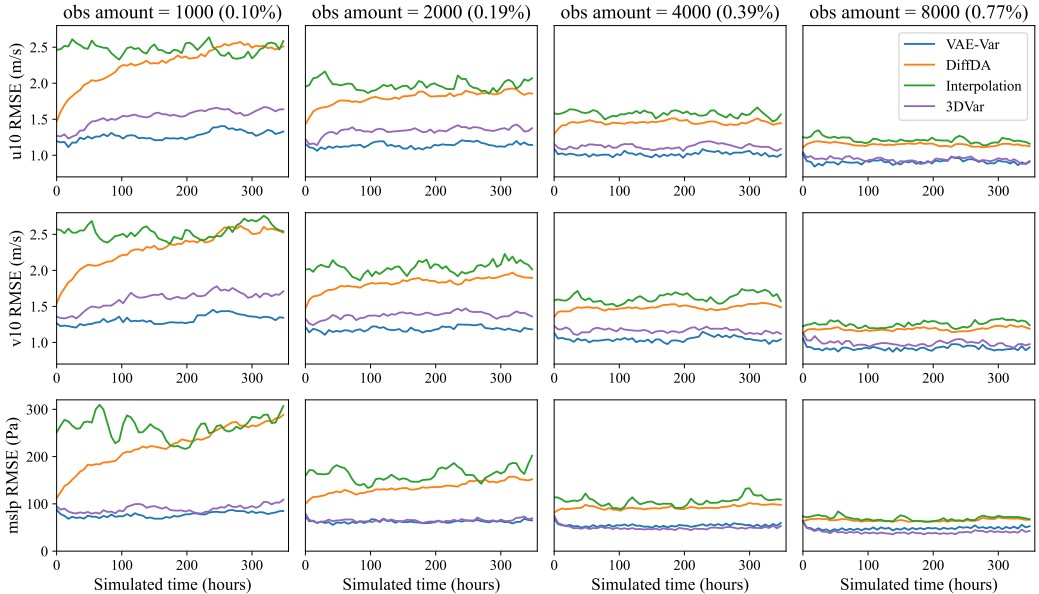

Figure 15: **Results for fixed observation positions (15 days).** The system is simulated in an auto-regressive manner for 15 days, starting from January 1, 2022. The RMSEs of the analysis field for three variables (u10, v10, mslp) are demonstrated in three different rows.

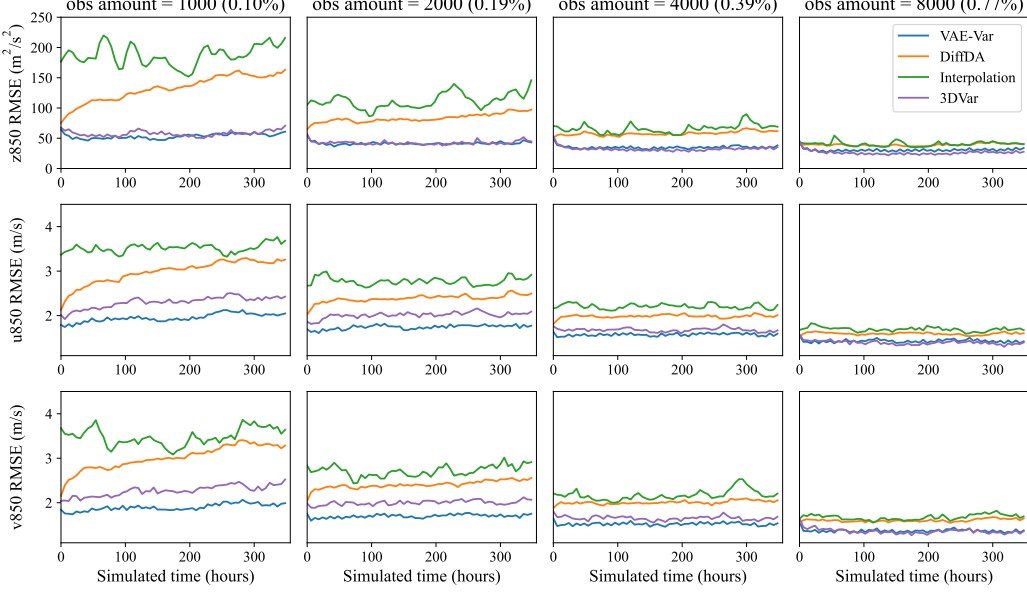

Figure 16: **Results for fixed observation positions (15 days).** The system is simulated in an auto-regressive manner for 15 days, starting from January 1, 2022. The RMSEs of the analysis field for three variables (z850, u850, v850) are demonstrated in three different rows.

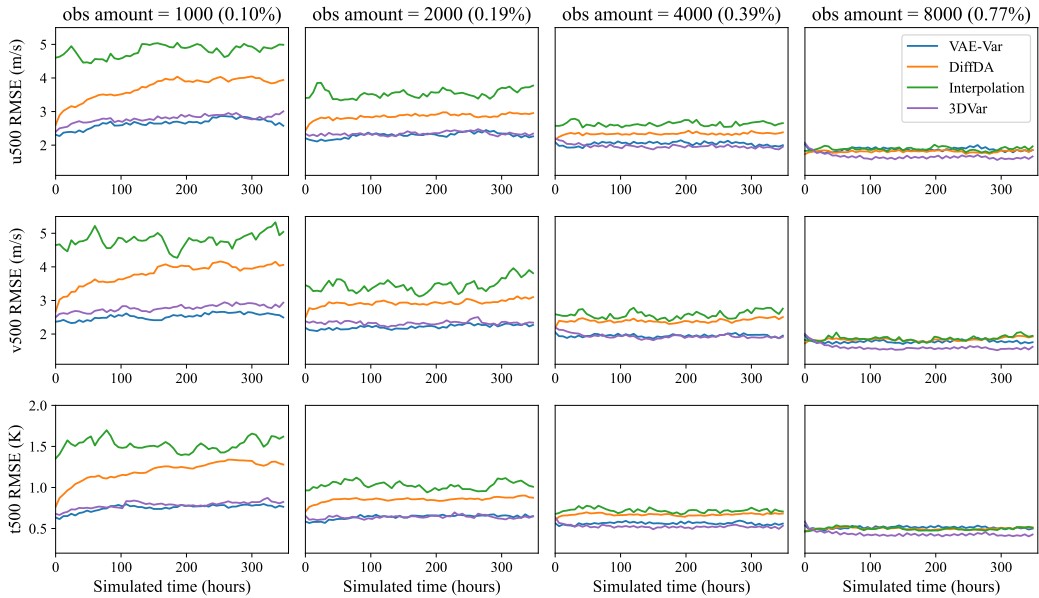

Figure 17: **Results for fixed observation positions (15 days).** The system is simulated in an auto-regressive manner for 15 days, starting from January 1, 2022. The RMSEs of the analysis field for three variables (u500, v500, t500) are demonstrated in three different rows.

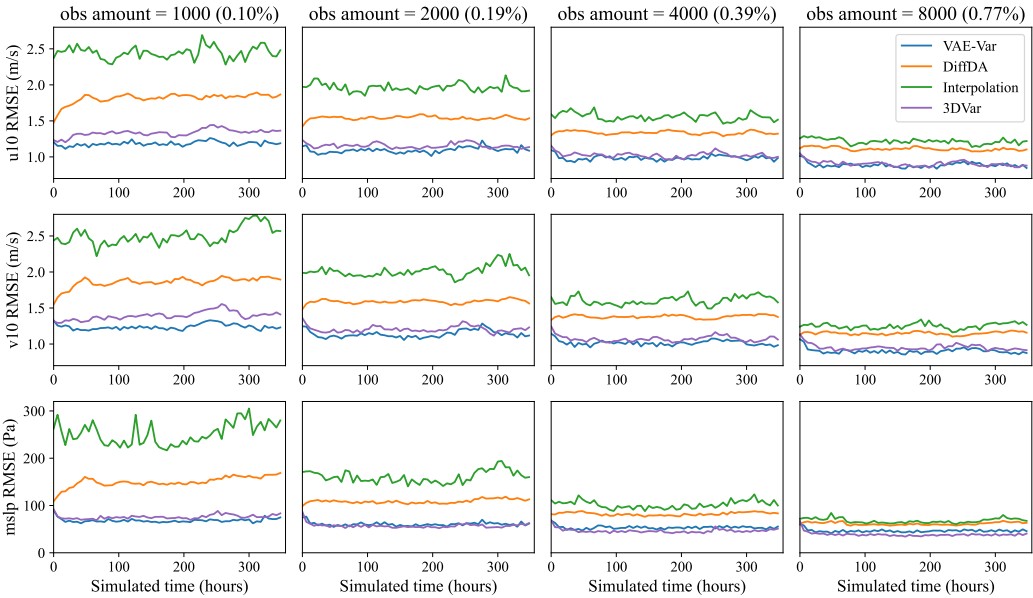

Figure 18: **Results for unfixed observation positions (15 days).** The system is simulated in an auto-regressive manner for 15 days, starting from January 1, 2022. The RMSEs of the analysis field for three variables (u10, v10, mslp) are demonstrated in three different rows.

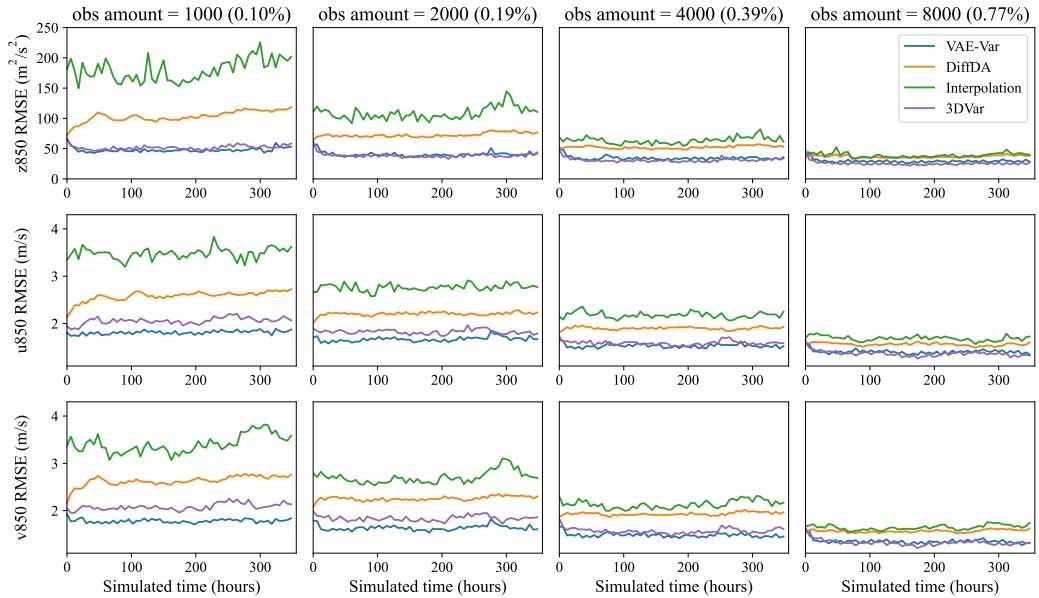

Figure 19: **Results for unfixed observation positions (15 days).** The system is simulated in an auto-regressive manner for 15 days, starting from January 1, 2022. The RMSEs of the analysis field for three variables (z850, u850, v850) are demonstrated in three different rows.

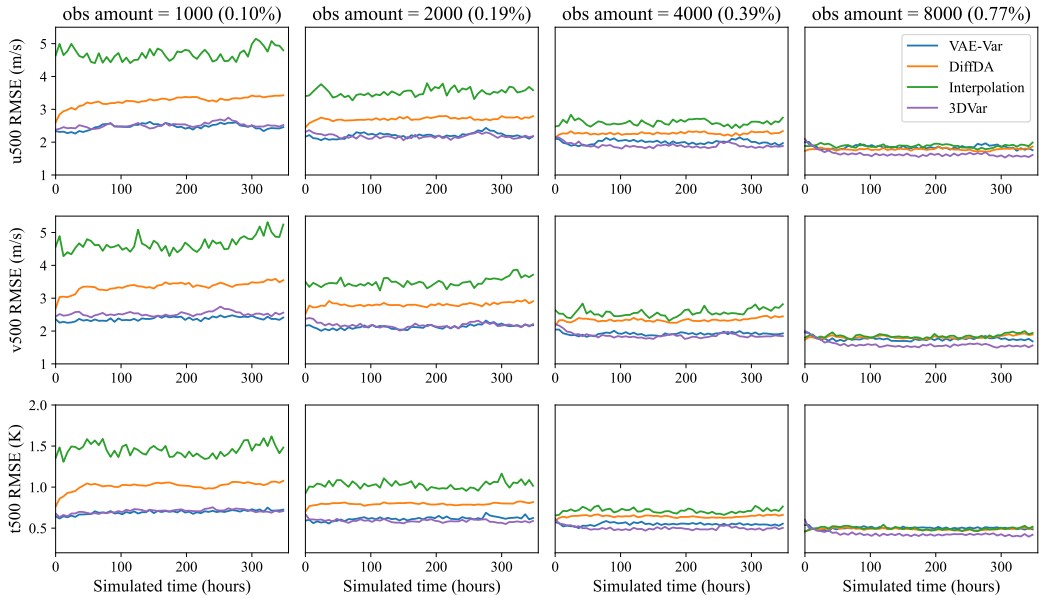

Figure 20: **Results for unfixed observation positions (15 days).** The system is simulated in an auto-regressive manner for 15 days, starting from January 1, 2022. The RMSEs of the analysis field for three variables (u500, v500, t500) are demonstrated in three different rows.

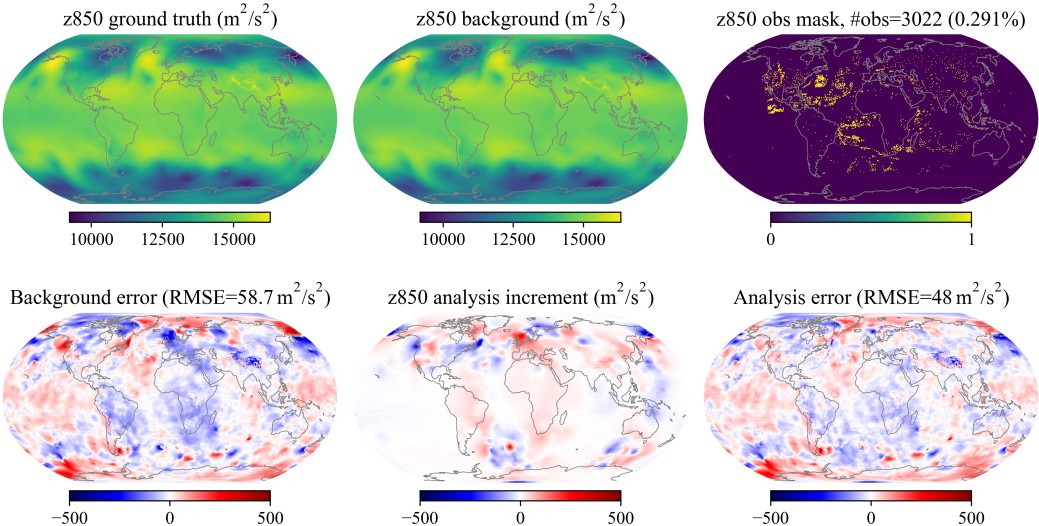

Figure 21: **Visualization of z850 assimilation results at time 2017-01-01 00:00:00.** The figure titled "z850 obs mask" correspond to observations at 858hPa because the observations use a different height axis.

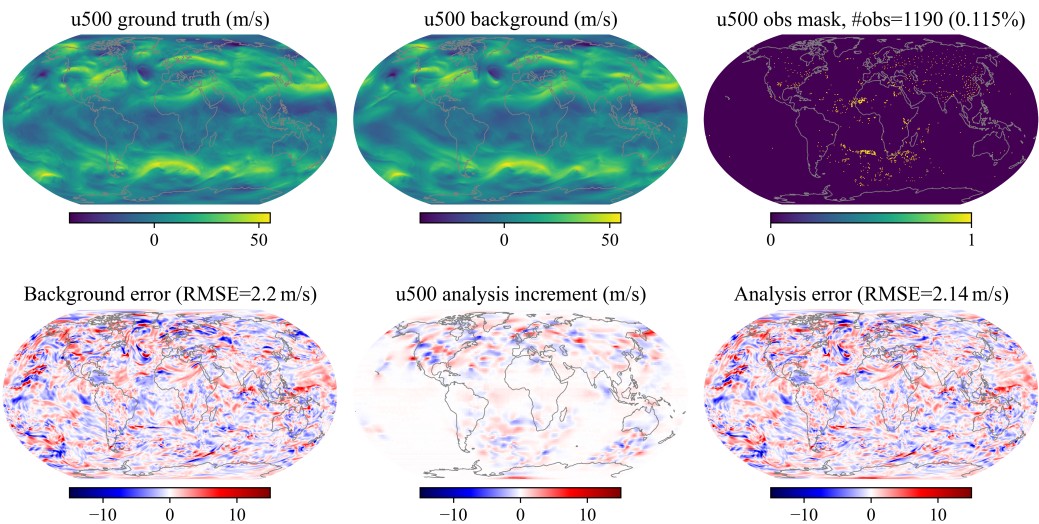

Figure 22: **Visualization of u500 assimilation results at time 2017-01-01 00:00:00.** The figure titled "u500 obs mask" correspond to observations at 501hPa because the observations use a different height axis.

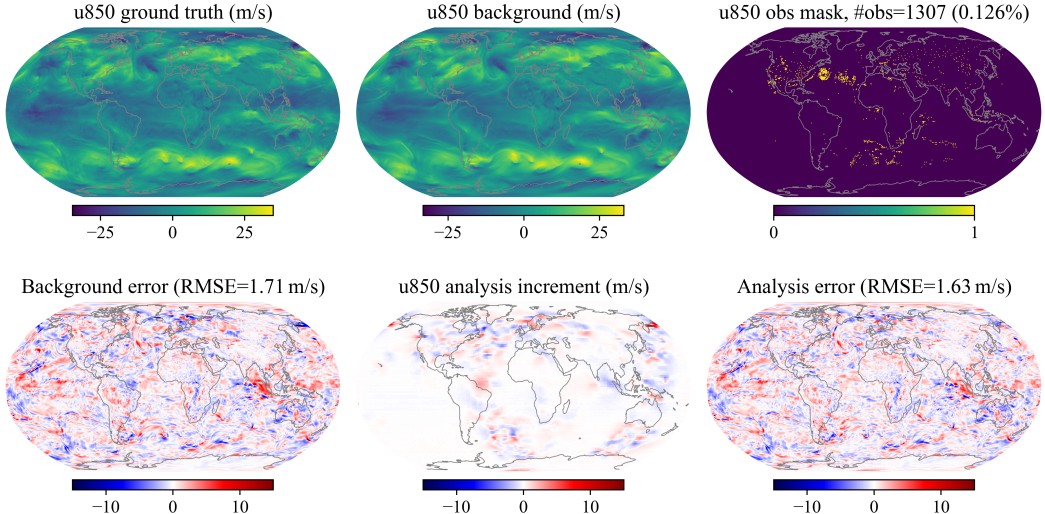

Figure 23: **Visualization of u850 assimilation results at time 2017-01-01 00:00:00.** The figure titled "u850 obs mask" correspond to observations at 858hPa because the observations use a different height axis.

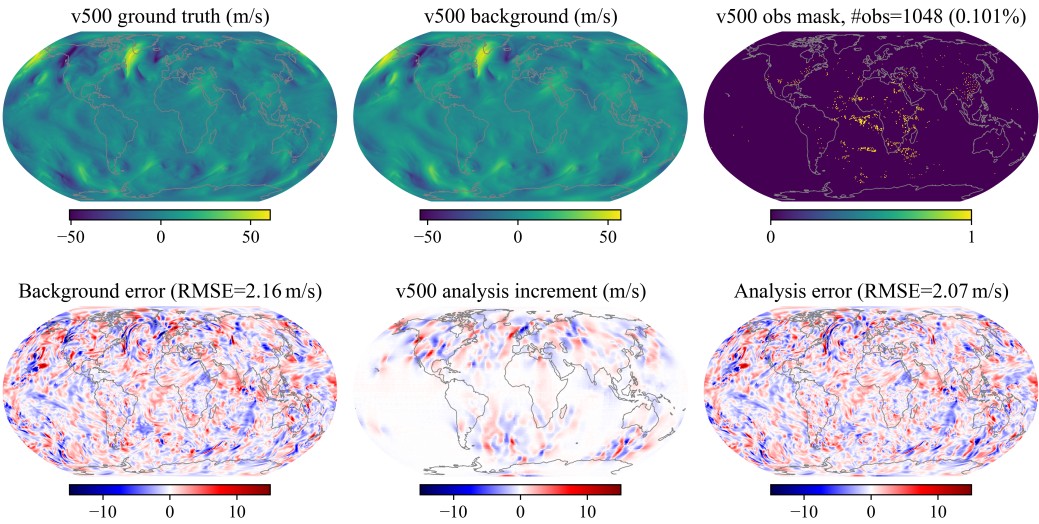

Figure 24: **Visualization of v500 assimilation results at time 2017-01-01 00:00:00.** The figure titled "v500 obs mask" correspond to observations at 501hPa because the observations use a different height axis.

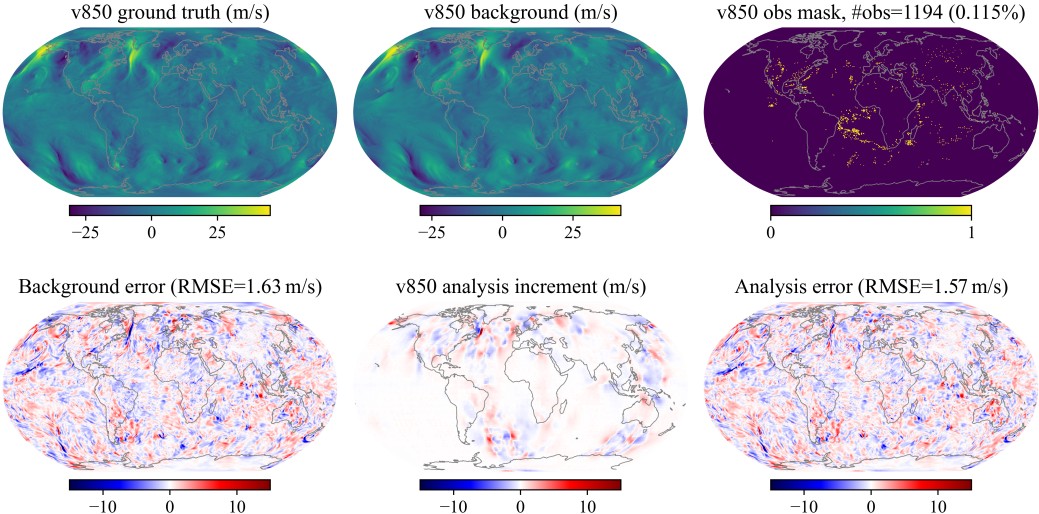

Figure 25: **Visualization of v850 assimilation results at time 2017-01-01 00:00:00.** The figure titled "v850 obs mask" correspond to observations at 858hPa because the observations use a different height axis.

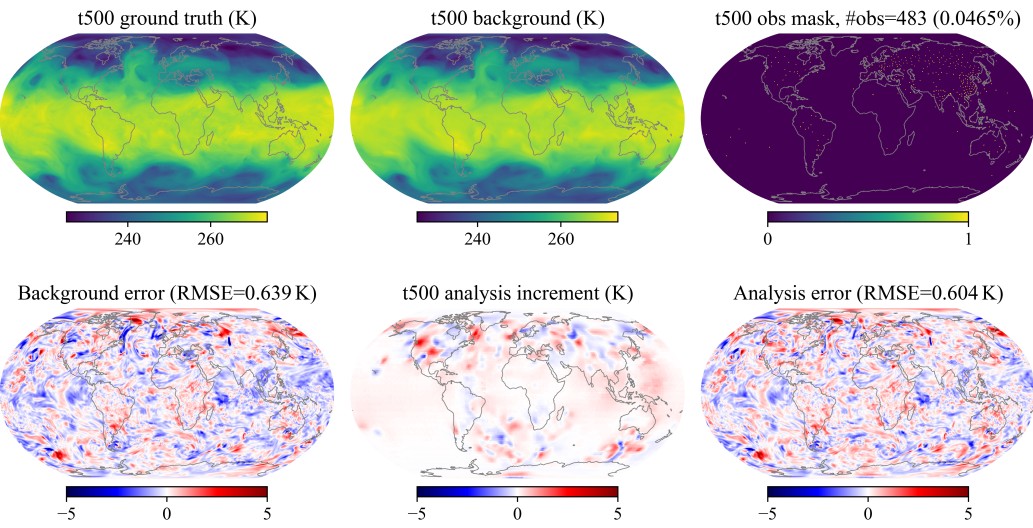

Figure 26: **Visualization of t500 assimilation results at time 2017-01-01 00:00:00.** The figure titled "t500 obs mask" correspond to observations at 501hPa because the observations use a different height axis.

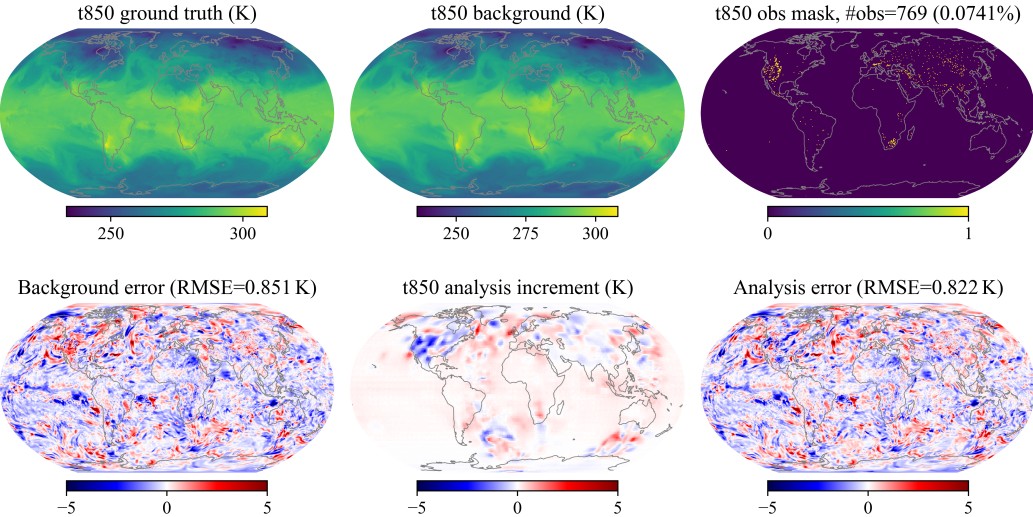

Figure 27: **Visualization of t850 assimilation results at time 2017-01-01 00:00:00.** The figure titled "t850 obs mask" correspond to observations at 858hPa because the observations use a different height axis.

