# OpenReview forum: "VAE-Var: Variational Autoencoder-Enhanced Variational Methods for Data Assimilation in Meteorology"
_ICLR.cc/2025/Conference — ICLR 2025 Poster_

### Official Review · Reviewer_Uc8k · 2024-10-25

**Soundness:** 2
**Presentation:** 3
**Contribution:** 2
**Rating:** 5
**Confidence:** 3

**Summary:**

This paper studies the data assimilation problem in the context of numerical weather forecast, and proposes a deep learning model for this problem based on a variational auto-encoder architecture. This approach captures the non-Gaussian background error distribution relative to the forecasting model within a variational data assimilation framework, by training a variational auto-encoder to learn the prior distribution. This deep learning model is evaluated on high-dimensional operational weather data and shows competitive performances compared to a recent diffusion based data assimilation algorithm.

**Strengths:**

The presented methodology is sound, and the proposed algorithm is accurately described. The paper is clearly written overall.

The proposed approach is evaluated on real-world data, on which it is shown to outperform the baselines. This suggests that considerable work has been invested to ensure the scalability of this method.

The VAE-Var algorithm is able to assimilate data from more general observation operators than DiffDA or other learning-based algorithms.

The authors propose a cyclic forecasting and assimilation setup where data assimilation and forecasting are coupled, with the forecast model, FengWu, being itself a large deep learning model. This method, already proposed in [1], is appealing for the future of numerical weather forecast as it does not rely on computational intensive weather models anymore once the neural networks are trained.

Although not emphasized in the paper, a key potential strength of this approach is that the variational problem is formulated in latent space, rather than in state space. This may be a considerable advantage if the neural network efficiently compresses the state variable. Indeed, the data assimilation problem is typically of very high dimension, which is a bottleneck in practice, and reducing the dimension of the computations involved is key.

**Weaknesses:**

One key issue in the paper is the assumption that the state-of-the-art is limited to spatial-only data assimilation, specifically 3DVar-like algorithms, where the observations are assimilated at each new time step, as in Algorithm 3. There is no mention of the 4DVar algorithm, which is the state of the art for operational data assimilation, and which differs significantly from the description proposed in section 2. Instead of modeling error by a Gaussian distribution at a given time step relative to the prediction at the previous time, 4DVar models the error distribution at a multitude of observation times, within a so-called assimilation window, thanks to a physical model that couples the different instants. By enforcing temporal dependencies in the observations via physical laws, the reconstructed state becomes more accurate, and 4DVar is known to outperform 3DVar. As opposed to the framework described in Section 2, the error distribution in 4DVar is not Gaussian, as there is a strongly nonlinear physical model in the equation. See [2] and [3] for more details.

To properly study the impact of the proposed method, I believe that it should be compared, at least on a toy system, to 4DVar, and potentially to learning-based counterparts of 4DVar that model the temporal dimension of the signal to reconstruct it, including diffusion models [4] and end-to-end reconstruction approaches [5, 6].
Furthermore, the upsides of using the presented variational auto-encoder framework rather than a diffusion model as in DiffDA is not so clear: the authors mention that their algorithm is more versatile in terms of observation operators, but this difference is not thoroughly explained in Section 3.


Additionally, I found that the paper lacks clarity in some parts. For example, (too) much space is dedicated to the Bayesian framework of variational data assimilation, whereas the training loss function of the proposed method, which relies on the VAE framework, is not explicitly state. In the experiment section, I think that enumerating the baselines in a common paragraph would improve the clarity.

Finally, the paper does not mention the computational cost of data assimilation and does not compare the computational times involved. This aspect is of significant importance, and computational savings are one of the main reasons underlying the deep learning approach for weather forecast problems. Therefore, I believe that it would be beneficial for the paper to discuss this topic and to compare the computational times between the different approaches, traditional and learning-based.

**Questions:**

In Figure 3, how would the authors explain that 3DVar outperforms DiffDA? Should the latter not be better than the former, as it can capture non-Gaussian statistics?

In line 261, the part about rescaling the background term is not very clear (although it may be explained in more details in the Appendix).

Using the term "interpolation" as a name for a nearest-neighbor method might be confusing, as "interpolation" already has a precise meaning in the data assimilation literature [3].

A paramount aspect in data assimilation is uncertainty quantification and measuring the various possible reconstruction states, under chaotic dynamics. This is briefly mentioned in the related work section, but it would be very relevant to discuss how the probabilistic framework underlying variational auto-encoders allows the learning model to sample different predictions for a given observation.

### References

[1] Huang, L., Gianinazzi, L., Yu, Y., Dueben, P. D., & Hoefler, T. (2024). Diffda: a diffusion model for weather-scale data assimilation. arXiv preprint arXiv:2401.05932.

[2] Lorenc, A. C., & Rawlins, F. (2005). Why does 4D‐Var beat 3D‐Var?. Quarterly Journal of the Royal Meteorological Society: A journal of the atmospheric sciences, applied meteorology and physical oceanography, 131(613), 3247-3257.

[3] Asch, M., Bocquet, M., & Nodet, M. (2016). Data assimilation: methods, algorithms, and applications. Society for Industrial and Applied Mathematics.

[4] Fablet, R., Chapron, B., Drumetz, L., Mémin, E., Pannekoucke, O., & Rousseau, F. (2021). Learning variational data assimilation models and solvers. Journal of Advances in Modeling Earth Systems, 13(10), e2021MS002572.

[5] Blanke, M., Fablet, R., & Lelarge, M. (2024). Neural Incremental Data Assimilation. arXiv preprint arXiv:2406.15076.

---

> ### Author Response · Authors · 2024-11-28
>
> We sincerely appreciate your careful and thoughtful comments and time. We try to explain your concerns point by point.
>
> **Q1: Relationship with 4DVar.**
>
> **Q1.1: "One key issue in the paper is the assumption that the state-of-the-art is limited to spatial-only data assimilation, specifically 3DVar-like algorithms, where the observations are assimilated at each new time step, as in Algorithm 3."**
>
> **A1.1:** One of the key advantages of VAE-Var is its ability to improve the modeling of background errors without altering the underlying framework of variational data assimilation. This allows for a straightforward extension of the framework to 4DVar. Specifically, let the current time be $t\_0$, and the observation time window be $[t\_0, t\_{N-1}]$, with observations at times $\\{ t_i \\}\_{i=0}^{N-1} $and corresponding observation values $\\{ \mathbf{y}\_i \\}\_{i=0}^{N-1}$. The observation term can be modified as follows:
> $$
> \mathcal{L}\_o(\mathbf{x}\_0, \\{\mathbf{y}\_i\\}\_{i=0}^{N-1}) = \frac12 \sum\_{i=0}^{N-1}\left(\mathbf{y}\_i - \mathcal{H}(\mathcal{M}\_{t\_0 \to t\_i}(\mathbf{x}\_0))\right)^\mathrm{T} \mathbf{R}^{-1} \left(\mathbf{y}\_i - \mathcal{H}(\mathcal{M}\_{t_0 \to t_i}(\mathbf{x}\_0))\right),
> $$
> where $\mathcal{M}\_{t\_0 \to t\_i}$ represents the flow-dependent operator. For the VAE framework, we express the new observation term as:
> $$
> \tilde{\mathcal{L}}\_o(\mathbf{z}) = \mathcal{L}\_o(\mathcal{D}(\mathbf{z}) + \mathbf{x}\_b, \\{\mathbf{y}\_i\\}\_{i=0}^{N-1}).
> $$
> As long as the flow-dependent operator $\mathcal{M}\_{t\_0 \to t\_i}$ is differentiable (which can be achieved using the method from Xiao et al. (2024)), this allows the VAE-4DVar formulation to be implemented.
>
> **Q1.2: "There is no mention of the 4DVar algorithm, which is the state of the art for operational data assimilation, and which differs significantly from the description proposed in section 2."**
>
> **A1.2:** The reason that the 4DVar algorithm is not implemented in this paper can be attributed to two main factors:
>
> 1. **Lack of 4DVar baseline results**: Most AI-based assimilation works have not considered time windows in their experiments. To our knowledge, only FengWu-4DVar (Xiao et al., 2024) and Fuxi-En4DVar (Li et al., 2024) have addressed time windows in their formulations. However, FengWu-4DVar was tested at low resolution (1.4°), and Fuxi-En4DVar, which was just recently published, did not perform cycling assimilation experiments.
> 2. **Unclear interaction between background error flow dependence and non-Gaussian modeling**: The mathematical impact of combining the flow-dependent background error model in 4DVar with the non-Gaussian background error modeling in VAE-Var is still unclear. Even if experimental results show that VAE-4DVar outperforms 4DVar, the underlying reasons for this improvement are not fully understood. As this is a research-oriented paper, we chose to use 3DVar in our experiments as a controlled comparison, which allows for a more direct evaluation of how the non-Gaussian modeling of the background field in VAE impacts assimilation performance.
>
> **Q1.3: "As opposed to the framework described in Section 2, the error distribution in 4DVar is not Gaussian, as there is a strongly nonlinear physical model in the equation. See [2] and [3] for more details."**
>
> **A1.3:** We respectfully disagree with the statement that the error distribution in 4DVar is non-Gaussian. In 4DVar, the flow-dependent term in the variational objective function is used to transform the stationary background error covariance in 3DVar into a background error covariance that implicitly evolves over time within the time window (Usui et al., 2015). However, even though this evolution occurs over time, it still represents a change in the covariance matrix of a Gaussian distribution, without departing from the Gaussian framework. Moreover, at the initial time step within the time window, the error modeling in 4DVar for the background field is identical to that in 3DVar (Rabier & Liu, 2003).

---

> ### Author Response · Authors · 2024-11-28
>
> **Q1.4: "To properly study the impact of the proposed method, I believe that it should be compared, at least on a toy system, to 4DVar, and potentially to learning-based counterparts of 4DVar that model the temporal dimension of the signal to reconstruct it, including diffusion models [4] and end-to-end reconstruction approaches [5, 6]."**
>
> **A1.4:** Thank you for your suggestion to compare the proposed method to 4DVar and learning-based counterparts such as diffusion models and end-to-end reconstruction approaches. Regarding the three works you mentioned:
>
> 1. **DiffDA (Huang et al., 2024)**: This approach does not address the assimilation of observations within a time window and can only assimilate observations at a single time point, so it does not directly align with the requirements of 4DVar.
> 2. **Fablet et al. (2021)**: While this work considers observations within a time window, it completely ignores the background field information (as noted in the second section of the paper). Therefore, it does not implement the full 4DVar framework.
> 3. **Blanke et al. (2024)**: This work considers the 4DVar framework but is lacking in experimental completeness and reproducibility. For example, the code repository linked in the paper is empty, and the experimental details provided are insufficient for full replication. Due to these issues, we believe it is not suitable as a baseline for comparison at this time.
>
> As supplementary experiments, we present results of traditional 4DVar and VAE-4DVar on the Lorenz63 system and low-resolution 1.4-degree FengWu model.
>
> **Lorenz63 system:**
>
> The Lorenz63 system is a three-variable system defined by the following ordinary differential equations:
> $$
> \frac{dX}{dt} = \sigma (Y-X) \\, \\, \\\\
> \frac{dY}{dt} = X(\rho - Z) - Y \\, \\, \\\\
> \frac{dZ}{dt} = XY - \beta Z
> $$
> Starting from a random system state, we use two sets of parameters, $\sigma = 10, \rho = 28, \beta = \frac{8}{3}$ and $\sigma = 11, \rho = 28, \beta = \frac{8}{3}$, to integrate the system for 10 steps with a step size of 0.01. These serve as the truth and background fields.
>
> For the VAE design, both the encoder and decoder were implemented using fully connected networks with two hidden layers, each with 8 neurons, and the latent space dimension was set to 3 (matching the physical space dimension). A total of 4000 samples were generated for training, with a KL-divergence and reconstruction loss weight of 0.3.
>
> Once trained, we designed observations for assimilation. Specifically, for 4DVar and VAE-4DVar, the time window is two time steps. The observation term is as follows:
> $$
> \mathcal{L}\_o(\mathbf{x}\_0, \mathbf{y}\_0, \mathbf{y}\_1) = \frac12 \left(\mathbf{y}\_0 - \mathcal{H}(\mathbf{x}\_0)\right)^\mathrm{T} \mathbf{R}^{-1} \left(\mathbf{y}\_0 - \mathcal{H}(\mathbf{x}\_0)\right)+\frac12 \left(\mathbf{y}\_1 - \mathcal{H}(\mathcal{M}\_{0\to 0.02}(\mathbf{x}\_0))\right)^\mathrm{T} \mathbf{R}^{-1} \left(\mathbf{y}\_1 - \mathcal{H}(\mathcal{M}\_{0\to 0.02}(\mathbf{x}\_0))\right),
> $$
> For 3DVar, the background field is only updated using the observation at the initial time step, as 3DVar cannot handle observations within a time window:
> $$
> \mathcal{L}_o(\mathbf{x}_0, \mathbf{y}_0) = \frac12 \left(\mathbf{y}_0 - \mathcal{H}(\mathbf{x}_0)\right)^\mathrm{T} \mathbf{R}^{-1} \left(\mathbf{y}_0 - \mathcal{H}(\mathbf{x}_0)\right),
> $$
> Observations are simulated by adding Gaussian noise to the ground truth. We evaluated the assimilation performance by computing the RMSE between the analysis and truth fields after a single assimilation step, testing under different Gaussian noise levels. All experiments were repeated 10 times, and the average results are shown in the table below. As shown, under most noise settings, VAE-3DVar outperforms 3DVar, but due to its inability to utilize the full time window, its performance is slightly worse than 4DVar. However, when VAE-Var is extended to 4DVar, the assimilation performance of VAE-4DVar exceeds that of 4DVar. This demonstrates that, at least under the current experimental parameters in the Lorenz63 system, VAE-Var can effectively be extended to four-dimensional variational assimilation.
>
> | Method \ Noise Std | 0.10  | 0.15  | 0.20  | 0.25  | 0.30  | 0.35  | 0.40  | 0.45  | 0.50  |
> | ------------------ | ----- | ----- | ----- | ----- | ----- | ----- | ----- | ----- | ----- |
> | 3DVar              | 0.168 | 0.204 | 0.233 | 0.261 | 0.271 | 0.288 | 0.298 | 0.312 | 0.317 |
> | 4DVar              | 0.149 | 0.168 | 0.179 | 0.193 | 0.205 | 0.212 | 0.219 | 0.223 | 0.237 |
> | VAE-3DVar          | 0.153 | 0.181 | 0.206 | 0.232 | 0.256 | 0.266 | 0.292 | 0.309 | 0.319 |
> | VAE-4DVar          | 0.151 | 0.159 | 0.163 | 0.174 | 0.185 | 0.189 | 0.196 | 0.200 | 0.212 |

---

> ### Author Response · Authors · 2024-11-28
>
> **A1.4 (cont'd):**
>
> **1.4-degree Low-resolution FengWu Model:**
>
> For the 1.4-degree low-resolution FengWu model, we extend VAE-Var to four-dimensional variational assimilation. The time window is set to 6 hours, and the observation term is:
> $$
> \mathcal{L}\_o(\mathbf{x}\_0, \\{\mathbf{y}\_i\\}\_{i=0}^{5}) = \frac12 \sum\_{i=0}^{5}\left(\mathbf{y}\_i - \mathcal{H}(\mathcal{M}\_{t_0 \to t\_0+i\\,\text{hours}}(\mathbf{x}\_0))\right)^\mathrm{T} \mathbf{R}^{-1} \left(\mathbf{y}\_i - \mathcal{H}(\mathcal{M}\_{t\_0 \to t\_0+i\\,\text{hours}}(\mathbf{x}\_0))\right),
> $$
> Here, the flow dependency is implemented using the FengWu forecast model with a 1-hour time step. Given the computational cost of tuning 4DVar for real-world scenarios, we currently report the results for 15-day cycling assimilation, starting from 2018-01-01 00:00, at a 5% observation ratio. The results for the RMSE of the Z500 analysis field at 00:00 every two days are shown below. VAE-4DVar outperforms VAE-3DVar significantly, demonstrating that our framework can be extended to 4DVar. Furthermore, when compared to the results from Xiao et al. (2024) in Figure 12, it can be found that the performance of VAE-4DVar is comparable with that of FengWu-4DVar under the same experimental settings.
>
> | Method \ Time | Day1  | Day3  | Day5  | Day7  | Day9  | Day11 | Day13 | Day15 |
> | ------------- | ----- | ----- | ----- | ----- | ----- | ----- | ----- | ----- |
> | VAE-3DVar     | 45.36 | 45.11 | 45.44 | 50.96 | 51.79 | 50.57 | 53.38 | 46.51 |
> | VAE-4DVar     | 50.29 | 38.88 | 37.20 | 43.21 | 42.27 | 39.81 | 40.86 | 40.15 |
>
> **Q2: "The upsides of using the presented variational auto-encoder framework rather than a diffusion model as in DiffDA is not so clear: the authors mention that their algorithm is more versatile in terms of observation operators, but this difference is not thoroughly explained in Section 3."**
>
> **A2:** Thank you for raising this point. The advantage of using the presented VAE-Var framework over a diffusion model like DiffDA lies primarily in its versatility regarding observation operators. Specifically, VAE-Var inherits the framework of variational data assimilation, which allows it to handle observations in a manner consistent with traditional algorithms. For observations from different sources, the traditional methods for handling observation operators can be directly applied to VAE-Var. In theory, this means that VAE-Var can accommodate the same types of observation operators as those used in traditional assimilation systems. Since different observation types are associated with different definitions of the observation operator $\mathcal{H}$, this aspect was not elaborated in the Methods section, but we emphasized the general applicability of our framework there.
>
> Regarding the specific experiments in this paper, the observation operators validated include the mask operator (where observation sites are located on grid points, consistent with DiffDA’s experimental setup) and the linear interpolation operator (where observation sites are not on the grid points but the observed variable matches the physical field variable, a scenario that DiffDA cannot currently handle). We have provided details on these observation operators in the experimental section.
>
> From a theoretical standpoint, our framework is also capable of handling more complex observations, such as satellite observations (where the observed variable is brightness temperature and the observation operator $\mathcal{H}$ would need to be constructed using radiative transfer equations). However, due to dataset limitations, we plan to conduct experiments involving these types of observations in future work.

---

> ### Author Response · Authors · 2024-11-28
>
> **Q3: "(Too) much space is dedicated to the Bayesian framework of variational data assimilation, whereas the training loss function of the proposed method, which relies on the VAE framework, is not explicitly state. In the experiment section, I think that enumerating the baselines in a common paragraph would improve the clarity."**
>
> **A3:** Thank you for your feedback. The extended discussion of the Bayesian framework in the paper is intended to provide background for the ICLR audience, many of whom may be less familiar with variational data assimilation. Since our method builds upon traditional variational assimilation and incorporates AI-based improvements, we believe this background helps position our work within the broader context of data assimilation for the AI community. In the revised version of the paper, we will slightly reduce the length of this discussion to maintain a more concise focus on the key aspects relevant to the ICLR audience.
>
> After the VAE training is completed, VAE-Var utilizes the pre-trained decoder $\mathcal{D}$ to perform zero-shot data assimilation for observations, without requiring any further training. The loss function used during the VAE training is explicitly described in Section 3 of the paper.
>
> As for your suggestion to present the baselines in a single paragraph, we agree that this would improve clarity, and we will make this adjustment in the revised version.
>
> **Q4: "The paper does not mention the computational cost of data assimilation and does not compare the computational times involved. "**
>
> **A4:** The main goal of VAE-Var is to improve the assimilation accuracy of 3DVar. In terms of algorithm design, VAE-Var replaces the linear transformation $\mathbf{U}$ in 3DVar with a nonlinear neural network $\mathcal{D}$. The number of floating-point operations in $\mathcal{D}$ is no samller than in $\mathbf{U}$, so from a floating-point computation perspective, the computational complexity of VAE-Var is not lower than that of 3DVar. However, VAE-Var's main advantage is its easy implementation on GPUs, which significantly improves overall assimilation efficiency. For example, on a single A100 GPU, one cycle of assimilation takes approximately 18 seconds. In contrast, 3DVar is typically implemented on CPUs, where one cycle of assimilation can take several minutes (Smith, et al., 2014). In our baseline experiments, running the 3DVar algorithm on a single CPU node takes about 4 minutes per cycle. Therefore, in practice, VAE-Var can easily leverage GPU capabilities to achieve faster assimilation than traditional 3DVar.
>
> As for comparisons with sampling-based AI methods such as DiffDA, direct comparisons are challenging because these studies do not provide detailed timing information. However, considering that diffusion-based models generally have slower inference speeds, we infer that methods like DiffDA may not have a computational efficiency advantage over VAE-Var.
>
> We hope this explanation provides clarity. Please let us know if further details are needed.
>
> **Q5: In Figure 3, how would the authors explain that 3DVar outperforms DiffDA? Should the latter not be better than the former, as it can capture non-Gaussian statistics?**
>
> **A5:** Thank you for your comment. In principle, DiffDA has the potential to outperform 3DVar, as it is designed to capture non-Gaussian statistics, which should provide more accurate results in certain contexts. However, there are a couple of reasons why, in the current experiments, DiffDA performs worse than 3DVar:
>
> 1. **Limited Training Data**: The dataset available for training is relatively small, with global historical reanalysis data spanning only 37 years, providing at most a few tens of thousands of valid samples. This is quite small compared to the millions and even billions of samples typically used in fields like computer vision. Diffusion models, in particular, require large datasets to perform well; for example, the best-performing models rely on billion-level image datasets (Rombach, et al., 2022). Although the idea behind DiffDA is promising, in the context of limited data, the data-hungry nature of diffusion model will result in suboptimal performance.
> 2. **Model Design Limitations**: The algorithm design of DiffDA may limit its ability to effectively propagate observation information over long distances. DiffDA's sampling approach for the posterior probability $p(\mathbf{x}|\mathbf{x}_b, \mathbf{y})$ relies on a soft-bleed mechanism to incorporate observations. This method tends to allow observations to primarily influence nearby grid points while neglecting their impact on distant regions of the physical field. In contrast, 3DVar utilizes recursive filtering to account for long-range correlations, which helps to incorporate the effect of observations over larger spatial distances.
>
> Thus, while DiffDA has great potential, its performance may be constrained by data limitations and its sampling strategy in the current setup.

---

> ### Author Response · Authors · 2024-11-28
>
> **Q6: "In line 261, the part about rescaling the background term is not very clear (although it may be explained in more details in the Appendix)."**
>
> **A6:** Thank you for your valuable feedback. We apologize for the lack of clarity regarding the rescaling of the background term. As you mentioned, this point may be explained in more detail in the Appendix. In the revised version, we will provide a clearer explanation of this process in the main text to ensure better understanding.
>
> Here, we provide a more detailed explanation on the choice of the coefficient $\lambda$. To provide some intuition, let us explain the role of $\lambda$ with the following analogy: Suppose the physical space has a dimensionality of 100, while the latent space has a dimensionality of 10. Since the observation deviation term $\mathcal{L}_o\left(\mathcal{D}(\mathbf{z}) + \mathbf{x}_b, \mathbf{y}\right)$ is calculated in the physical space, and the background error term $\mathcal{L}_b(\mathbf{z}) = \frac{1}{2} \mathbf{z}^\mathrm{T} \mathbf{z}$ is calculated in the latent space, the former would intuitively be 100 times larger than the latter. To balance this dimensionality difference, we introduce the coefficient $\lambda$ to scale the background error term in the latent space, such that $\mathcal{L}_b(\mathbf{z}) = \frac{1}{2} \lambda \mathbf{z}^\mathrm{T} \mathbf{z}$. The parameter $\lambda$ is thus approximately equal to $\left(\frac{d_x}{d_z}\right)^2$, where $d_x$ and $d_z$ are the dimensions of the physical space and the latent space, respectively. In our experimental setup, the physical space has dimensions $d_x = 69 \times 721 \times 1440$ and the latent space has dimensions $d_z = 32 \times 721 \times 1440$. This gives $\lambda \approx 4.64$, and we round this value to 4 for simplicity.
>
> **Q7: "Using the term "interpolation" as a name for a nearest-neighbor method might be confusing, as "interpolation" already has a precise meaning in the data assimilation literature [3]."**
>
> **A7:** Thank you for your comment. In our paper, we use the term "interpolation" in two distinct contexts, and We would like to clarify both.
>
> First, in the baseline assimilation method, we refer to the interpolation algorithm used for the assimilation process. Specifically, this interpolation is based on bilinear interpolation in two dimensions, which we believe is appropriately categorized as an interpolation method in the context of data assimilation. This approach is consistent with the work of Huang et al. (2024), where interpolation is similarly used for baseline assimilation.
>
> Second, in the settings of real observations, we apply the nearest-neighbor method to map the observations to the nearest grid points. The use of the term "interpolation" here is indeed not precise, and it has been corrected in the new version. Thank you for pointing it out.

---

> ### Author Response · Authors · 2024-11-28
>
> **Q8: "It would be very relevant to discuss how the probabilistic framework underlying variational auto-encoders allows the learning model to sample different predictions for a given observation."**
>
> **A8:** We fully acknowledge that uncertainty quantification of the analysis field (reconstructed field) is a crucial aspect in data assimilation, particularly in the context of chaotic systems. However, leveraging stochastic neural networks to sample different analysis fields (reconstruction fields) based on deterministic observations remains relatively underexplored. The main challenge in this area lies in the fact that once the background field conditional distribution $p(\mathbf{x}|\mathbf{x}_b)$ is constructed, incorporating the observation information $\mathbf{y}$ into the sampling process becomes quite difficult.
>
> To the best of our knowledge, Diffusion models, particularly through works such as DiffDA (Huang, et al., 2024) and SDA (Rozet & Louppe, 2023), are the only successful attempts to sample from the posterior probability $p(\mathbf{x}|\mathbf{x}_b, \mathbf{y})$, as already referenced in the paper. In contrast, when it comes to VAE, there has yet to be a successful application of their stochastic nature for sampling analysis field uncertainty. This is mainly due to the challenge of developing a sampling strategy for $p(\mathbf{x}|\mathbf{x}_b, \mathbf{y})$ within the VAE framework.
>
> Currently, the use of VAE in data assimilation has been limited to either dimensionality reduction for latent space assimilation (Melinc & Zaplotnik, 2024) or, as in our case, modeling the uncertainty of the background field. In our framework, the assimilation still follows a deterministic variational approach and calculates the maximum likelihood $\arg\max_\mathbf{x} p(\mathbf{x}|\mathbf{x}_b, \mathbf{y})$ for estimating the analysis field. Thus, we think it is premature to explore the specific question of how to leverage the stochastic nature of VAE for sampling different analysis field states from a given observation. This topic is still in its infancy and will require further exploration in future work.
>
> **References:**
>
> [1] Xiao, Y., Bai, L., Xue, W., Chen, H., Chen, K., Han, T., & Ouyang, W. (2024). Towards a Self-contained Data-driven Global Weather Forecasting Framework. In *Forty-first International Conference on Machine Learning*.
>
> [2] Li, Y., Han, W., Li, H., Duan, W., Chen, L., Zhong, X., ... & Sun, X. (2024). Fuxi‐en4dvar: An assimilation system based on machine learning weather forecasting model ensuring physical constraints. *Geophysical Research Letters*, *51*(22), e2024GL111136.
>
> [3] Usui, N., Fujii, Y., Sakamoto, K., & Kamachi, M. (2015). Development of a four-dimensional variational assimilation system for coastal data assimilation around Japan. *Monthly Weather Review*, *143*(10), 3874-3892.
>
> [4] Rabier, F., & Liu, Z. (2003, September). Variational data assimilation: theory and overview. In *Proc. ECMWF Seminar on Recent Developments in Data Assimilation for Atmosphere and Ocean, Reading, UK, September 8–12* (pp. 29-43).
>
> [5] Huang, L., Gianinazzi, L., Yu, Y., Dueben, P. D., & Hoefler, T. (2024). Diffda: a diffusion model for weather-scale data assimilation. *arXiv preprint arXiv:2401.05932*.
>
> [6] Fablet, R., Chapron, B., Drumetz, L., Mémin, E., Pannekoucke, O., & Rousseau, F. (2021). Learning variational data assimilation models and solvers. Journal of Advances in Modeling Earth Systems, 13(10), e2021MS002572.
>
> [7] Blanke, M., Fablet, R., & Lelarge, M. (2024). Neural Incremental Data Assimilation. arXiv preprint arXiv:2406.15076.
>
> [8] Smith, T. M., Gao, J., Calhoun, K. M., Stensrud, D. J., Manross, K. L., Ortega, K. L., ... & Riedel, C. (2014). Examination of a real-time 3DVAR analysis system in the hazardous weather testbed. *Weather and forecasting*, *29*(1), 63-77.
>
> [9] Rombach, R., Blattmann, A., Lorenz, D., Esser, P., & Ommer, B. (2022). High-resolution image synthesis with latent diffusion models. In *Proceedings of the IEEE/CVF conference on computer vision and pattern recognition* (pp. 10684-10695).
>
> [10] Rozet, F., & Louppe, G. (2023). Score-based data assimilation. *Advances in Neural Information Processing Systems*, *36*, 40521-40541.
>
> [11] Melinc, B., & Zaplotnik, Ž. (2024). 3D‐Var data assimilation using a variational autoencoder. *Quarterly Journal of the Royal Meteorological Society*, *150*(761), 2273-2295.

---

> > ### Comment · Reviewer_Uc8k · 2024-12-01
> > **Answer to the authors**
> >
> > I appreciate the authors' thorough and thoughtful response. The additional explanations improve the clarity of the paper, and I acknowledge their effort to address my concerns.
> >
> > Regarding spatio-temporal data assimilation, I understand that the proposed work primarily focuses on improving the background component, making it theoretically compatible with any variational data assimilation technique. However, I remain convinced that the omission of 4DVar from the paper's introduction and problem statement is not fully justified. Even in the updated version, there is no explicit mention of spatio-temporal data assimilation or 4DVar as the state-of-the-art method for this problem. In their response (A.1.2), the authors argue that such omissions are consistent with the AI literature, which often does not reference spatio-temporal assimilation. While this may be true, a paper aiming to advance data assimilation techniques should reference and contextualize its contributions relative to 4DVar, as it remains the standard in this field. A discussion of 4DVar would enhance the paper's rigor and bridge the gap between traditional and AI-based methodologies.
> >
> > Additionally, I believe that the comparison between the Gaussian nature of the background error in 3DVar and the complex, non-Gaussian distribution modeled by diffusion models such as SDA is somewhat misleading: diffusion architectures in these works model the prior distribution of trajectories within a spatio-temporal Bayesian inverse problem formulation. This is conceptually distinct from the background error distribution in 3DVar. The proper "traditional" counterpart to SDA’s prior trajectory distribution is the trajectory distribution in 4DVar, which is inherently non-Gaussian due to the nonlinear dynamics of the physical model. Clarifying this distinction in the paper would help readers better understand the advantages and limitations of the proposed approach.
> >
> > The authors’ response partially addresses my concerns, and I appreciate their clarifications and additional context. However, I still find that the improvements of the proposed method over traditional spatio-temporal data assimilation techniques remain unclear. Given these considerations, I increase my score from 3 to 5.

---

> > > ### Author Response · Authors · 2024-12-02
> > >
> > > Thank you for your thoughtful and detailed response, as well as for increasing your score. We appreciate your valuable feedback and would like to address your concerns.
> > >
> > > Firstly, regarding the comparison with 4DVar, we acknowledge your point, and we will certainly include this in the revised version of the paper. Since uploading new manuscript is not allowed now, we briefly outline the changes we are making:
> > >
> > > 1. We will change the second-to-last section from “Related Work” to “Discussion.” In this section, we will keep the existing content about latent data assimilation, and we will add a discussion on 4DVar, including the extension of VAE-Var to VAR-4DVar and its relation to other spatio-temporal methods.
> > > 2. In Appendix A.2, we will provide a comparison of VAE-4DVar with 3DVar, 4DVar, and VAE-3DVar, using the Lorenz63 system as an example. This comparison will also include experimental results to clarify the advantages of the proposed method.
> > >
> > > Secondly, regarding your comment about the comparison between the Gaussian characteristics of the background error in 3DVar and the complex, non-Gaussian distribution modeled by diffusion models such as SDA, we believe there may have been some misunderstanding. This could be due to the fact that your review did not reference SDA, which is methodologically distinct from the other works you mentioned.
> > >
> > > To clarify:
> > >
> > > - **For a general 4DVar problem**, we consider a time window $[t\_0, t\_{N-1}]$, where the assimilation algorithm takes as input: (1) the observations $\\{\mathbf{y}\_i\\}\_{i=0}^{N-1}$, and (2) the initial background state estimate $\mathbf{x}\_b$ for the time window. From a Bayesian perspective, the general 4DVar problem is modeled as the probability distribution $p\left(\\{\mathbf{x}\_i\\}\_{i=0}^{N-1}\middle| \mathbf{x}\_b, \\{\mathbf{y}\_i\\}\_{i=0}^{N-1}\right)$.
> > > - **Traditional algorithms**: The 4DVar algorithm seeks the maximum likelihood estimate, i.e., $\arg\max\_{\\{\mathbf{x}\_i\\}\_{i=0}^{N-1}} p\left(\\{\mathbf{x}\_i\\}\_{i=0}^{N-1}\middle| \mathbf{x}\_b, \\{\mathbf{y}_i\\}\_{i=0}^{N-1}\right)$. As outlined in Fisher et al. (2021), weak-constraint 4DVar can be formulated to minimize this objective function, providing an estimate of the time series $\\{\mathbf{x}\_i\\}\_{i=0}^{N-1}$. In strongly-constraint 4DVar, the physical model is assumed to deterministically evolve the states over time, which means that the entire sequence of states $\\{\mathbf{x}\_i\\}\_{i=0}^{N-1}$ is determined once the initial state $\mathbf{x}\_0$ is known. This is the standard formulation of 4DVar that we refer to in our additional experiments.
> > > - **AI methods**: While many AI approaches attempt to solve the spatio-temporal reconstruction problem, many of them still operate within traditional frameworks. For example, the work by Fablet et al. (2021) is based on the weak-constraint 4DVar framework, but it replaces the model evolution and variational function optimization with a neural network. The work you mentioned, SDA (Rozet & Louppe, 2023), is indeed more innovative. It uses diffusion models to learn the prior distribution of the temporal physical field states $p\left(\\{\mathbf{x}\_i\\}\_{i=0}^{N-1}\right)$, and then applies Diffusion Posterior Sampling to sample the posterior $p\left(\\{\mathbf{x}\_i\\}\_{i=0}^{N-1}\middle| \\{\mathbf{y}\_i\\}\_{i=0}^{N-1}\right)$. We recognize the ingenuity in this approach, especially in handling time-varying observations. We also acknowledge that SDA still has limitations, such as neglecting the estimation of the initial physical state $\mathbf{x}_b$ at the start of the time window. This estimate is critical for operational forecasting, and its omission presents challenges for the algorithm. This is why a fair comparison between VAE-Var and SDA is difficult: VAE-Var is designed to handle the distribution of $\mathbf{x}_b$, which SDA does not address. We believe that combining the strengths of both methods could lead to an even more robust assimilation algorithm. We will include this discussion in the revised version of the paper under the "Discussion" section.
> > >
> > > Thank you again for your helpful comments. We look forward to presenting these revisions in the updated version of the paper.
> > >
> > > **References:**
> > >
> > > [1] Fisher, M., Trémolet, Y., Auvinen, H., Tan, D., & Poli, P. (2011). Weak-constraint and long-window 4D-Var. *ECMWF Technical Memoranda*, *655*, 47.
> > >
> > > [2] Fablet, R., Chapron, B., Drumetz, L., Mémin, E., Pannekoucke, O., & Rousseau, F. (2021). Learning variational data assimilation models and solvers. Journal of Advances in Modeling Earth Systems, 13(10), e2021MS002572.
> > >
> > > [3] Rozet, F., & Louppe, G. (2023). Score-based data assimilation. *Advances in Neural Information Processing Systems*, *36*, 40521-40541.

---

> ### Author Response · Authors · 2024-12-01
>
> Dear Reviewer Uc8k,
>
> We hope this message finds you well. We apologize for any inconvenience caused by reaching out over the weekend. As the rebuttal discussion period is coming to a close, we would greatly appreciate your feedback on whether our rebuttal has effectively addressed your concerns. Please let us know if you have any further questions, and we would be more than happy to assist you.
>
> We sincerely appreciate your valuable time and effort.
>
> Best Regards,
>
> The Authors

---

### Official Review · Reviewer_bsmw · 2024-10-29

**Soundness:** 3
**Presentation:** 3
**Contribution:** 3
**Rating:** 6
**Confidence:** 4

**Summary:**

The authors offer a way of conducting sparse data assimilation using a minimization problem by estimating the error term of a forecasting model as compared to the state directly, and combining it with variational data assimilation. They achieve convincing results on weather forecasting.

**Strengths:**

The paper offers a method of data assimilation for sparse meteorological problems which surpass current baselines like DiffDA, and give very promising results. The method is clear, and the problem is particularly applicable to real-time weather forecasting.

**Weaknesses:**

Currently, the model seems method-specific, limiting its generalizability as a data assimilation approach since any new improvements or retraining of the original forecasting model would require retraining of this model. Is there a way to perhaps make this more robust? However, in the grand scheme of things, due to its superior performance, this isn't that big of an issue in my opinion.

**Questions:**

When generating, why is it integrated 0->2t and t->2t instead of just 0->t vs the ground truth at t?

How is the speed in terms of assimilation versus previous data assimilation approaches?

Since you're modeling the error; suppose that the initial state is very ill-estimated. Is the data assimilation approach still able to resolve the problem, or does it rely on the initial state estimate being within some stability region?

---

> ### Author Response · Authors · 2024-11-28
>
> We sincerely appreciate your careful and thoughtful comments and time, especially your approval for our proposed framework, which really means a lot to us. We try to explain your concerns point by point.
>
> **Q1: "Currently, the model seems method-specific, limiting its generalizability as a data assimilation approach since any new improvements or retraining of the original forecasting model would require retraining of this model. Is there a way to perhaps make this more robust?"**
>
> **A1:** Thank you for raising this important point. Currently, our framework has been tested only with the FengWu forecasting model, and as such, its generalizability is indeed limited. Theoretically, if we wish to generalize our approach to other forecasting models with the same forecasting variables, it would be possible to fine-tune the original VAE model. However, most other forecasting models, such as GraphCast, use different forecasting variables (i.e., different input-output structures compared to FengWu), which means that the model would require retraining in these cases.
>
> In future work, we plan to explore this issue and investigate ways to improve the robustness of our framework, including adapting it to different forecasting models. A possible solution would be to add an adapter in the VAE design to handle different types of input-output structures. In this case, only the adapter module would need to be fine-tuned for different forecasting models.
>
> **Q2: "When generating, why is it integrated 0->2t and t->2t instead of just 0->t vs the ground truth at t?"**
>
> **A2:** Thank you for your insightful question. The reason we integrate from 0 to 2t and from t to 2t, rather than just from 0 to t versus the ground truth at t, is primarily to follow the approach used in the NMC method. Our algorithm currently does not make significant modifications to the variational algorithm framework, so we adopted a sample generation strategy similar to that of the NMC method.
>
> It is also worth noting that the original NMC paper (Parrish & Derber, 1992) does not provide a detailed explanation for this choice and we deduce that it is largely based on empirical considerations. In our experiments, we found that using two integrals (0 to 2t and t to 2t) generates samples that are smoother and easier to learn compared to using a single integration.
>
> We hope this clarifies the reasoning behind our approach. Thank you again for your valuable feedback.
>
> **Q3: "How is the speed in terms of assimilation versus previous data assimilation approaches?"**
>
> **A3:** The main goal of VAE-Var is to improve the assimilation accuracy of 3DVar. In terms of algorithm design, VAE-Var replaces the linear transformation $\mathbf{U}$ in 3DVar with a nonlinear neural network $\mathcal{D}$. The number of floating-point operations in $\mathcal{D}$ is no samller than in $\mathbf{U}$, so from a floating-point computation perspective, the computational complexity of VAE-Var is not lower than that of 3DVar. However, VAE-Var's main advantage is its easy implementation on GPUs, which significantly improves overall assimilation efficiency. For example, on a single A100 GPU, one cycle of assimilation takes approximately 18 seconds. In contrast, 3DVar is typically implemented on CPUs, where one cycle of assimilation can take several minutes (Smith, et al., 2014). In our baseline experiments, running the 3DVar algorithm on a single CPU node takes about 4 minutes per cycle. Therefore, in practice, VAE-Var can easily leverage GPU capabilities to achieve faster assimilation than traditional 3DVar.
>
> As for comparisons with sampling-based AI methods such as DiffDA, direct comparisons are challenging because these studies do not provide detailed timing information. However, considering that diffusion-based models generally have slower inference speeds, we infer that methods like DiffDA may not have a computational efficiency advantage over VAE-Var.
>
> We hope this explanation provides clarity. Please let us know if further details are needed.

---

> ### Author Response · Authors · 2024-11-28
>
> **Q4: "Since you're modeling the error; suppose that the initial state is very ill-estimated. Is the data assimilation approach still able to resolve the problem, or does it rely on the initial state estimate being within some stability region?"**
>
> **A4:** Thank you for raising this insightful question. Theoretically, VAE-Var achieves error reduction effectively when the initial state is the forecast field from FengWu, thereby enabling better performance in cyclic forecasting. However, during our experiments, we observed that even when the initial state is not derived from FengWu’s forecasts, the assimilation process can still continuously reduce errors over cycles, eventually achieving comparable performance.
>
> To investigate this further, we constructed initial fields using the following five approaches:
>
> 1. Start from ERA5 192 hours earlier and use thirty-two 6-hour FengWu forecasts as the initial field.
> 2. Directly use ERA5 fields from 24 hours earlier as the initial field.
> 3. Directly use ERA5 fields from 48 hours earlier as the initial field.
> 4. Directly use ERA5 fields from 192 hours earlier as the initial field.
> 5. Use ERA5 fields from half a year earlier as the initial field.
>
> The table below shows the results of z500 RMSE ($m^2/s^2$) for different initializations, displaying error statistics at every 72-hour interval during a 25-day assimilation process (96 assimilation steps). Detailed plots are included in the revised paper appendix.
>
> | Init Method \ Time | Day1    | Day4   | Day7   | Day10  | Day13  | Day16  | Day19 | Day22 | Day25 |
> | ------------------ | ------- | ------ | ------ | ------ | ------ | ------ | ----- | ----- | ----- |
> | 1                  | 393.62  | 104.61 | 79.30  | 63.31  | 61.41  | 82.18  | 67.09 | 67.95 | 59.08 |
> | 2                  | 328.50  | 115.12 | 85.37  | 64.95  | 64.12  | 62.83  | 72.43 | 62.29 | 75.63 |
> | 3                  | 484.13  | 124.82 | 86.88  | 70.48  | 65.97  | 61.50  | 62.86 | 58.12 | 53.03 |
> | 4                  | 818.27  | 222.12 | 139.85 | 96.52  | 81.24  | 94.59  | 77.21 | 87.15 | 68.30 |
> | 5                  | 2019.68 | 685.26 | 358.80 | 228.72 | 158.47 | 122.28 | 93.37 | 77.58 | 67.31 |
>
> From the results, we observe that all five initialization methods lead to eventual convergence of the assimilation system. The primary distinction lies in the number of steps required for convergence, with larger initial errors necessitating more steps. This demonstrates the robustness of VAE-Var to the initial state.
>
> Here we provide a potential explanation: during the cyclic assimilation and forecasting process, only in the first cycle does the error distribution of the background field significantly deviate from the VAE training set. In subsequent cycles, the background field is provided by the FengWu forecast, so VAE-Var can handle it relatively well. As this cycle continues, analysis field errors are progressively reduced until they stabilize at a low level.
>
> **References:**
>
> [1] Parrish, D. F., & Derber, J. C. (1992). The National Meteorological Center's spectral statistical-interpolation analysis system. *Monthly Weather Review*, *120*(8), 1747-1763.
>
> [2] Smith, T. M., Gao, J., Calhoun, K. M., Stensrud, D. J., Manross, K. L., Ortega, K. L., ... & Riedel, C. (2014). Examination of a real-time 3DVAR analysis system in the hazardous weather testbed. *Weather and forecasting*, *29*(1), 63-77.

---

> > ### Comment · Reviewer_bsmw · 2024-12-01
> >
> > Thanks to the authors for their comprehensive rebuttal. They have answered most of my questions, but the robustness of the model to further updates of the model and generalizability to other realizations still seems somewhat limited.
> >
> > Though some of the other reviewers still seem to have some remaining concerns regarding rigor and 4D-Var comparisons, I still believe this is a good addition to the data assimilation community. As a result, I will maintain my current score.

---

> > > ### Author Response · Authors · 2024-12-02
> > >
> > > Thank you for your recognition of our work.

---

### Official Review · Reviewer_jauN · 2024-11-01

**Soundness:** 2
**Presentation:** 3
**Contribution:** 2
**Rating:** 6
**Confidence:** 5

**Summary:**

The paper introduces VAE-Var, a data assimilation method that combines variational autoencoders (VAE) with a traditional variational data assimilation technique to enhance weather forecasting accuracy. By using a VAE, VAE-Var relaxes the traditional Gaussian background error assumption, enabling it to better model complex, non-Gaussian background error distributions. Through experiments with both synthesized observations and real-world observations, VAE-Var demonstrates improved or comparable accuracy over established methods, including 3DVar and DiffDA. The approach’s adaptability to diverse data types and its demonstrated year-long forecasting stability showcases VAE-Var’s promise for real-world weather forecasting applications.

**Strengths:**

The paper communicates its concepts clearly. The use of VAEs to model non-Gaussian background error distributions is an original contribution, addressing a major assumption in background error modeling, with promising results. The experiment on real-world observations moves this work beyond a proof-of-concept. The topic is highly relevant: data assimilation is essential for numerical weather prediction, and developing efficient DA algorithms for AI-based weather prediction is necessary to improve their capability. Additionally, the integration of machine learning, specifically generative modeling, into DA represents a forward-looking approach that could influence further research in hybrid models for weather and climate prediction.

**Weaknesses:**

### 1. Regarding ERA5 as the 'ground truth'.
ERA5, as a reanalysis product, can serve as a useful reference, but it should not be treated as "ground truth," especially for evaluating data assimilation algorithms. ERA5 is known to have a lot of issues that can be attributed to terrain effects, poor coverage of assimilated observations, and the numerical model used. For example, ERA5 exhibits underestimated temperature in some regions of Europe[1], an overly strong equatorial mesospheric jet, mismatches in the analyzed near-surface wind/temperature/humidity[2], significant precipitation/temperature biases in some regions of North America[3], etc.

Therefore, merely evaluating data assimilation algorithms on ERA5 data overlooks the actual 'ground truth' beyond ERA5's inherent biases and errors. The authors could benefit from reviewing some non-ML DA papers that work on real observations, to see how their DA approaches are evaluated. For example, to assess whether a data assimilation algorithm truly works, it is important to compare its performance directly with real observations. A potential test could be holding out actual observations at the same positions at each time step and using these to evaluate the analyses from all implemented DA algorithms. This experiment would require no retraining and would allow for direct comparison and diagnostics of the DA algorithm developed. Paper [4], which uses the root-mean-square departures from actual observation to validate their approach, serves as a reference, though there are likely many more (for example, [5]).

Regarding ERA5 as ground truth and only evaluating algorithms on ERA5 are common issues in AI for weather research. Recognizing that ERA5 is not a definitive ground truth would provide the community with a clearer perspective and encourage more comprehensive benchmarking with different data sources, thus advancing the field.
[1] https://www.mdpi.com/2073-4441/14/4/543
[2] https://confluence.ecmwf.int/display/CKB/ERA5%3A+data+documentation#ERA5:datadocumentation-Knownissues
[3] https://hess.copernicus.org/articles/24/2527/2020/
[4] https://journals.ametsoc.org/view/journals/mwre/151/7/MWR-D-22-0260.1.xml
[5] https://rmets.onlinelibrary.wiley.com/doi/full/10.1002/qj.3917

### 2. Lack of reproducibility and missing details weaken the results and claims.
- As the code is not provided, additional details on the configuration of algorithms and baselines are necessary. However, they are not provided.
- There are no details on how the observation operator is constructed, particularly for real observations.
- 3DVar details are missing, such as how $\text{U}_p$, $S$, $\text{U}_v$, and $\text{U}_h$ are determined?
- Configuration details for DiffDA are also absent, leaving readers uncertain about how this baseline was set up and tuned for the experiments.
- Providing the above details would enhance the reliability and transparency of the reported results, allowing readers to better assess and understand the performance claims of VAE-Var relative to these baselines.


### 3. Simplification
- As the authors mention, the simplification in the VAE-Var objective function is not mathematically rigorous. More importantly, it also weaken the claim that VAE-Var could better leverage the non-Gaussian statistics. By omitting the exact term in the background error component that accounts for the nonlinear transformation from a Gaussian to a non-Gaussian distribution, the model loses a critical factor needed to rigorously capture the intended non-Gaussian effects.  Unfortunately, addressing this limitation may be challenging, as the most straightforward solution would involve making the original, more complex optimization problem tractable.
- A minor issue related to appendix A.2: the reference for equation (7), *Giaquinta & Modica (2012): Mathematical analysis: functions of one variable. Springer Science & Business Media*, does not have Chapter 2.5.3 and equation (7).

**Questions:**

1. One major question that need to be addressed: VAE-Var improves minimally or does not improve with an increasing amount of available observations, whereas other approaches, both ML-based and traditional, show noticeable gains under similar conditions (Figures 3, 4, 8, 9, 10, 11, 12, 13). This behavior is unusual for data assimilation algorithms, as more observations typically enhance analysis accuracy. A thorough discussion or ablation study may be needed to investigate this.

Some suggestions from the review feedback system : )
- Conduct an ablation study to investigate why VAE-Var's performance doesn't improve significantly with more observations
- Discuss potential reasons for this behavior, such as possible saturation effects or limitations in the model's capacity to incorporate additional information
- Analyze the implications of this behavior for the algorithm's practical use in different observational settings
- If possible, propose modifications to the algorithm that might address this limitation

2. Formulating  $p(y∣x)$ as a Gaussian distribution is indeed an approximation of the observation errors. Although it is widely used in operational numerical weather prediction (NWP), it is an approximation instead of a well-defined characteristics. This assumption does not always hold. For instance, quality-controlled observations of brightness temperature from the High Resolution Infrared Sounder exhibit distributions significantly different from Gaussian. Incorrectly assuming Gaussian statistics in such cases can lead to substantial biases in the estimated state, as shown in studies like[6], which diagnose and assess the impacts of non-Gaussianity in innovations on data assimilation accuracy. For rigor, the paper could benefit from framing the Gaussian distribution of $p(y∣x)$ as a well-accepted but approximate assumption, rather than misleadingly stating that Gaussian error is a well-defined characteristic of observation errors.

[6]: Pires, Carlos A., Olivier Talagrand, and Marc Bocquet. "Diagnosis and impacts of non-Gaussianity of innovations in data assimilation." *Physica D: Nonlinear Phenomena* 239.17 (2010): 1701-1717.

3. Minor issues:
- Line 202-2-3 and Algorithm 1 and Figure 1. Is $\tau$  the number of time steps or the time gap between two snapshots? It is a bit confusing here.
- Inconsistency between the lower half panel of Figure 1 and Algorithm 2: How is the hidden state initialized? 0 or a random field?
- Algorithm 3: the model time step T is not explicitly defined in the algorithm.
- The last term in Equation (8) should be $\frac{1}{2}\log \det \left( \frac{\partial \mathcal{D}(\textbf{z})}{\partial \textbf{z}} \right)^T \left( \frac{\partial \mathcal{D}(\textbf{z})}{\partial \textbf{z}} \right)$

---

> ### Author Response · Authors · 2024-11-28
>
> We sincerely appreciate your careful and thoughtful comments and time. We try to explain your concerns point by point.
>
> **Q1: Regarding ERA5 as the 'ground truth'.**
>
> **A1:** Thank you for your insightful comments. We fully acknowledge the issues associated with using ERA5 as a definitive "ground truth," as it is well-known to have biases and errors stemming from terrain effects, poor coverage of assimilated observations, and the numerical model used. We agree that simply evaluating data assimilation algorithms on ERA5 data does not address the underlying concerns regarding ERA5’s inherent biases.
>
> Our decision to use ERA5 as a reference in our study is based on two main considerations:
>
> 1. **Challenges in obtaining high-quality observational datasets**: Currently, obtaining a globally comprehensive, publicly available observational dataset remains difficult. The GDAS dataset, while large, contains multi-source observational data that are mixed from different instruments and vary in quality. For example, measurements such as z500 can exhibit significant deviations from ERA5 by several hundred $m^2/s^2$, and the dataset has not undergone rigorous quality control. As a result, we do not consider such observational data as reliable "ground truth." In contrast, ERA5, though not perfect, is a standardized reanalysis product that provides a more consistent reference for our work.
> 2. **The logical issue of using observational data as ground truth**: The ultimate goal of data assimilation is to generate analysis fields that balance forecast models and observational data, both of which are inherently biased. Using these biased observations to evaluate the performance of DA algorithms introduces a logical flaw, as the assumptions about bias are part of the problem. Additionally, observational data are sparse, and the performance at specific sites cannot fully represent the accuracy of the entire forecast field. Thus, we believe that there is still considerable room for development in global weather system assimilation evaluations, but this topic is not the main focus of our paper.
>
> For these reasons, most of the current work in the field of AI for data assimilation, including ours, uses ERA5 as a reference. ERA5’s standardized dataset allows AI researchers to focus on the algorithmic aspects rather than the challenges of obtaining, processing, and evaluating observational data. This helps to accelerate the iterative process of improving AI-based data assimilation algorithms. That said, we also look forward to the emergence of high-quality, standardized observational datasets like ERA5, which will enable the community to conduct more robust evaluations using real observational data in the future.
>
> Here, we also add an experiment in which the observations are used for ground truth for evaluation. Specifically, we select the observation sites within the region bounded by 50°E to 150°E longitude and 15°N to 40°N latitude (with the main area covering East Asia). During the assimilation process, we do not assimilate these observations, but use them as truth for evaluation. Starting from January 1, 2017, the cyclic forecasting and assimilation system is simulated for one year, and we calculate the RMSE between the analysis field and the GDAS observations within that region and compare the results of VAE-Var and 3DVar. In the table below, we demonstrate the RMSE results for z501 ($m^2/s^2$) and u501 ($m/s$)  averaged among each month and please refer to the Appendix of the manuscript for more details. It can be found that for z501, VAE-Var outperforms 3DVar on most months (10 out of 12), while for u501, VAE-Var outperforms 3DVar on all months, which further supports the effectiveness of our approach.
>
> RMSE of z501:
>
> | Method \ Time | Jan.   | Feb.   | Mar.   | Apr.   | May.   | Jun.   | Jul.   | Aug.   | Sep.   | Oct.   | Nov.   | Dec.   |
> | ------------- | ------ | ------ | ------ | ------ | ------ | ------ | ------ | ------ | ------ | ------ | ------ | ------ |
> | 3DVar         | 341.68 | 346.87 | 360.39 | 341.92 | 369.91 | 275.26 | 376.18 | 387.01 | 331.14 | 334.73 | 319.33 | 422.23 |
> | VAE-Var       | 310.89 | 316.54 | 261.39 | 278.80 | 334.37 | 347.66 | 307.46 | 290.11 | 399.75 | 265.60 | 277.20 | 364.65 |
>
> RMSE of u501:
>
> | Method \ Time | Jan. | Feb. | Mar. | Apr. | May. | Jun. | Jul. | Aug. | Sep. | Oct. | Nov. | Dec. |
> | ------------- | ---- | ---- | ---- | ---- | ---- | ---- | ---- | ---- | ---- | ---- | ---- | ---- |
> | 3DVar         | 5.04 | 4.77 | 4.86 | 5.00 | 4.98 | 4.87 | 5.46 | 6.28 | 5.23 | 5.35 | 5.41 | 5.81 |
> | VAE-Var       | 3.42 | 3.50 | 4.00 | 4.05 | 4.38 | 4.34 | 4.01 | 3.70 | 4.17 | 3.69 | 3.52 | 4.01 |

---

> ### Author Response · Authors · 2024-11-28
>
> **Q2: Lack of reproducibility and missing details weaken the results and claims.**
>
> **Q2.1: "As the code is not provided, additional details on the configuration of algorithms and baselines are necessary. However, they are not provided."**
>
> **A2.1**: Thank you for your comments. We appreciate the importance of reproducibility and transparency in research. We would like to emphasize that the code will be made publicly available once the paper is accepted for publication. In the meantime, we will provide as many details as possible regarding the implementation of the algorithms and baselines to address your concerns.
>
> **Q2.2: "There are no details on how the observation operator is constructed, particularly for real observations."**
>
> **A2.2**: Thank you for raising this important point. The observation operator construction is indeed a key part of the assimilation process.
>
> For simulated observations, the observation operator $\mathcal{H}$ is simply implemented as a mask, where we apply it at the locations corresponding to the available observations.
>
> For real observations, we construct a new, finer grid for the observations and map the observations to the nearest grid points. Theoretically, this grid can be fine enough to minimize the error introduced by nearest-neighbor approximation. As a trade-off between accuracy and computational cost, the grid is designed to have the same latitude and longitude resolution as the FengWu physical grid, but with 40 vertical layers, which is much denser than FengWu. After data processing, all observations are placed on a grid with a finer resolution than FengWu. The observation operator $\mathcal{H}$ is then defined as the interpolation from the FengWu grid to the observation grid, which can be implemented using a differentiable 'Linear' layer in PyTorch.
>
> Since the geopotential height changes approximately exponentially with physical height, logarithmic linear interpolation is required when performing vertical layer interpolation. The computation of the interpolation matrix $P\in\mathbb{R}^{40\times 13}$ is shown below, where $X$ represents the scale of the geopotential height at the physical grid points (in this work is [50, 100, 150, 200, 250, 300, 400, 500, 600, 700, 850, 925, 1000]hPa) and $Y$ represents that at the observational grid points (in this work is [50, 54, 58, 63, 68, 73, 79, 86, 92, 100, 108, 116, 126, 136, 147, 158, 171, 185, 199, 215, 232, 251, 271, 293, 316, 341, 368, 398, 430, 464, 501, 541, 584, 631, 681, 735, 794, 858, 926, 1000]hPa).
>
> **Define the logarithmic interpolation matrix $P$:**
>
> $P = \text{zeros}( \text{len}(Y), \text{len}(X) )$
>
> ```python
> for i = 0 to len(Y):
>     for j = 0 to len(X):
>         if Y[i] == X[j]:
>             P[i, j] = 1
>         elif Y[i] > X[j] and Y[i] < X[j+1]:
>             P[i, j] = (log(X[j+1]) - log(Y[i])) / (log(X[j+1]) - log(X[j]))
>             P[i, j+1] = (log(Y[i]) - log(X[j])) / (log(X[j+1]) - log(X[j]))
> return P
> ```
>
> After calculating the interpolation matrix, the PyTorch construction of the observation operator is shown below.
>
> **For the physical layer $x \in \mathbb{R}^{69 \times 721 \times 1440}$, the observation operator $\mathcal{H}(x)$ is implemented in PyTorch as:**
>
> ```python
> x0 = x.unsqueeze(0)
> y = []
> y.append(x0[:, :4])
>
> for i = 0 to 5:
>     mat = x0[:, 4+i*13:4+(i+1)*13]
>     mat = torch.nn.functional.linear(mat.transpose(1, 3), P).transpose(1, 3)
>     y.append(mat)
>
> y = torch.cat(y, 1)
> y = y.squeeze(0)
> return y
> ```
>
> We hope this explanation clarifies how the observation operator is constructed, particularly for real observations. We have added this part to the revised manuscript.

---

> ### Author Response · Authors · 2024-11-28
>
> **Q2.3: "3DVar details are missing, such as how $\mathbf{U}_p$, $\mathbf{S}$, $\mathbf{U}_v$, and $\mathbf{U}_h$ are determined?"**
>
> **A2.3** Thank you for your comments regarding the details of the 3DVar implementation. The implementation follows the GEN_BE method from Descombes et al. (2015) and we reproduce it with Python. Generally speaking, GEN\_BE consists of three steps. First, 1000 ERA5 fields are sampled between 1979 and 2015; then the NMC method is applied to calculate the resulting background error samples; last, calculate the statistics of $\mathbf{U}$. In GEN\_BE, $\mathbf{U}$ is assumed to be decomposed into four operators, $\mathbf{U} = \mathbf{U}_p\mathbf{S}\mathbf{U}_v\mathbf{U}_h$, where $\mathbf{U}_p$ handles the transformation of physical variables, $\mathbf{S}$ represents the standard deviation, $\mathbf{U}_v$ corresponds to vertical transformation, and $\mathbf{U}_h$ handles horizontal transformation.
>
> The details for constructing the four operators are as follows:
>
> 1. **Implementation of $\mathbf{U}_p$**
>
> $\mathbf{U}_p$ is the only transformation that does not require parameters derived from the NMC samples. It is used to convert the stream function and potential function ($\phi$, $\psi$) into wind components ($u$, $v$) in the latitude-longitude directions, while keeping other variables unchanged. Specifically, it is computed using the following equations:
> $$
> u=\frac{\partial \psi}{\partial y} +\frac{\partial \phi}{\partial x},v=-\frac{\partial \psi}{\partial x} + \frac{\partial \phi}{\partial y}
> $$
>
> 2. **Implementation of $\mathbf{S}$**
>
> $\mathbf{S}$ represents the error variance for each variable layer. To calculate it, we first apply the inverse transformation of $\mathbf{U}_p$ to all error samples (i.e., converting winds from the latitude-longitude direction back to stream functions and potential functions). This is done using spectral methods, which are supported by the `torch-harmonic` library in PyTorch for solving the differential equations. After transforming the fields, we compute the sample variance at each pixel. The resulting variances are averaged in the latitude-longitude direction, yielding 69 values corresponding to the error variance at each vertical layer.
>
> 3. **Implementation of $\mathbf{U}_v$**
>
> $\mathbf{U}_v$ is computed using eigenvalue decomposition. Specifically, for each upper-air variable, we first average the variable in the latitude-longitude direction to obtain 1000 69-length vectors. The covariance matrix $\mathbf{A}$ of these 1000 vectors is then computed. We perform eigen-decomposition on $\mathbf{A}$ such that $\mathbf{A} = \mathbf{P} \mathbf{\Lambda} \mathbf{P}^{-1}$, and $\mathbf{U}_v$ is then defined as $\mathbf{U}_v = \mathbf{P} \mathbf{\Lambda}^{1/2}$, where $\mathbf{P}$ is the eigenvector matrix and $\mathbf{\Lambda}$ is the diagonal matrix of eigenvalues.
>
> 4. **Implementation of $\mathbf{U}_h$**
>
> $\mathbf{U}_h$ is essentially a horizontal recursive filter applied to each of the 69 layers. The length scale $L$ for the filter is given by:
> $$
> L=\left(\frac{8\cdot \mathrm{Variance(\phi)}}{\mathrm{Variance}(\nabla^2\phi)}\right)^{1/4}
> $$
> This filter is applied to each layer individually to perform horizontal smoothing.
>
> To summarize, $\mathbf{U}_p$ is a fixed, parameter-free variable transformation operator that converts stream and potential functions into wind components; $\mathbf{S}$ represents error variance with 69 parameters corresponding to each vertical layer; $\mathbf{U}_v$ corresponds to eigenvalue decomposition applied to the 1000 samples, yielding 69*69 + 69 parameters; $\mathbf{U}_h$ is a horizontal recursive filter applied to each of the 69 layers, with 69 parameters.
>
> **Q2.4: "Configuration details for DiffDA are also absent, leaving readers uncertain about how this baseline was set up and tuned for the experiments."**
>
> **A2.4:** The experimental results for DiffDA are based on the optimal results provided in the original paper. We have ensured that the observation settings and initial fields used for VAE-Var are identical to those used for DiffDA. Therefore, the comparison between VAE-Var and DiffDA is conducted under fair and consistent conditions.
>
> **Q2.5: "Providing the above details would enhance the reliability and transparency of the reported results, allowing readers to better assess and understand the performance claims of VAE-Var relative to these baselines."**
>
> **A2.5**: Thank you for your valuable suggestion. We agree that providing these details will enhance the reliability and transparency of the reported results. These additional details will be included in the revised manuscript, and we will also add a "Reproducibility Statement" section to ensure that readers can more effectively assess and understand the performance claims of VAE-Var relative to the baselines.

---

> ### Author Response · Authors · 2024-11-28
>
> **Q3: Simplification**
>
> **Q3.1: "As the authors mention, the simplification in the VAE-Var objective function is not mathematically rigorous. More importantly, it also weaken the claim that VAE-Var could better leverage the non-Gaussian statistics. By omitting the exact term in the background error component that accounts for the nonlinear transformation from a Gaussian to a non-Gaussian distribution, the model loses a critical factor needed to rigorously capture the intended non-Gaussian effects. Unfortunately, addressing this limitation may be challenging, as the most straightforward solution would involve making the original, more complex optimization problem tractable."**
>
> **A3.1:** Thank you for pointing out the issue related to the omission of the Jacobian determinant term. We acknowledge that removing this term weakens the mathematical rigor of the model, particularly in capturing the non-Gaussian statistics. We agree that this term would be crucial for rigorously modeling the transformation from a Gaussian to a non-Gaussian distribution. However, we believe that the replacement of the Jacobian term with an amplification factor $\lambda$ in the background error term is reasonable for an AI work, for the following reasons:
>
> 1. **Intuitive Reasoning for the Simplification**: From an intuitive perspective, this simplification still maintains the core function of the variational objective. When including the Jacobian determinant term, the full variational objective function can be expressed as:
>
>    $\mathcal{L}(\mathbf{z}) = \frac{1}{2} \mathbf{z}^\mathrm{T} \mathbf{z} + \frac{1}{2} \log \det \left( \frac{\partial \mathcal{D}(\mathbf{z})}{\partial \mathbf{z}} \right)^\mathrm{T} \left( \frac{\partial \mathcal{D}(\mathbf{z})}{\partial \mathbf{z}} \right) + \mathcal{L}_o \left( \mathcal{D}(\mathbf{z}) + \mathbf{x}_b, \mathbf{y} \right)$
>
>    The core of the VAE-Var framework is to learn the non-Gaussian distribution of the background error. By constructing the decoder $\mathcal{D}$, we restrict the analysis field to the region defined by $\mathcal{D}(\mathbf{z}) + \mathbf{x}_b$. Therefore, the observation error term $\mathcal{L}_o \left( \mathcal{D}(\mathbf{z}) + \mathbf{x}_b, \mathbf{y} \right)$ is central to the algorithm, ensuring that the analysis field not only fits the observations but also retains the structure of the background error distribution.
>
>    The first and second terms can be viewed as a regularization to avoid overfitting the observations. From this perspective, the specific form of these two terms is not so important. Borrowing from the ideas in machine learning, we unify them into a parameterized term $\frac12\lambda \mathbf{z}^\mathrm{T} \mathbf{z}$. The parameter $\lambda$ balances the latent space and physical space, and should therefore be chosen as the square of the ratio of the dimensions of the physical space to the latent space. In our experiments, the latent space has dimensions of $32 \times 721 \times 1440$, while the physical space has dimensions of $69 \times 721 \times 1440$, yielding a squared ratio of approximately 4.64, which we approximate as 4. This value of $\lambda$ is used in our experiments.
>
> 2. **Empirical Support for the Simplification**: The experimental results support our approach. Both the simulated observation and the GDAS observation experiments show that the assimilation of the analysis field using VAE-Var outperforms the baselines 3DVar and DiffDA, demonstrating the effectiveness of our algorithm.
>
> 3. **Perspective from AI Research**: In the context of AI-based data assimilation, to the best of our knowledge, almost no large-scale AI-based assimilation work has strictly followed the theoretical framework without any approximations. For instance, latent space assimilation algorithms, such as those in Melinc and Zaplotnik (2024), have not rigorously established a relationship between the results of latent space assimilation and the original space, instead adopting an empirical approach to algorithm design. Similarly, diffusion-based assimilation works, such as those by Huang et al. (2024), have not established a strict Bayesian model linking inpainting-based posterior sampling algorithms to the original assimilation problem. In Rozet & Louppe (2023), some approximations are made in the posterior sampling algorithm for $p(y | x(t))$. In comparison, we believe that proposing a well-founded theoretical framework, followed by empirical approximations and experimental validation, is a reasonable progression in AI algorithm development.

---

> ### Author Response · Authors · 2024-11-28
>
> **Q3.2: "A minor issue related to appendix A.2: the reference for equation (7), *Giaquinta & Modica (2012): Mathematical analysis: functions of one variable. Springer Science & Business Media*, does not have Chapter 2.5.3 and equation (7)."**
>
> **A3.2:** Thank you for pointing this out. You are correct that the section 2.5.3 does not exist in the 2012 edition of Giaquinta & Modica. The reference in the manuscript was intended to cite section 2.5.2 in Giaquinta & Modica (2010), where the change of variables formula is discussed. We have also included a more detailed derivation in the revised manuscript. We appreciate your careful attention to detail and apologize for the confusion caused by this error.
>
> **Q4: "VAE-Var improves minimally or does not improve with an increasing amount of available observations, whereas other approaches, both ML-based and traditional, show noticeable gains under similar conditions (Figures 3, 4, 8, 9, 10, 11, 12, 13). This behavior is unusual for data assimilation algorithms, as more observations typically enhance analysis accuracy. "**
>
> **Q4.1: "Conduct an ablation study to investigate why VAE-Var's performance doesn't improve significantly with more observations."**
>
> **A4.1:**  Thank you for your suggestion. In response, we have conducted additional experiments with varying numbers of observation points. In the original version of the paper, we used observation densities of 1,000, 2,000, 4,000, and 8,000 observation columns. To investigate the impact of observation quantity further, we have now added experiments with 100, 200, 400, 800, 10,000, 20,000, and 40,000 observation columns (the largest density corresponding to approximately one observation per grid point on a global 1.25-degree latitude-longitude grid).
>
> We conducted the experiments using a 15-day assimilation cycle from January 1 to January 15, 2022, with different observation locations at each time step but a consistent number of observations. Due to space limitations, we present the RMSE of the z500 variable of the analysis fields at midnight every two days in the table below, comparing the performance of 3DVar and VAE-Var. Further details can be found in the Appendix of the revised manuscript.
>
> Results of 3DVar assimilation:
>
> | #Obs \ Time | Day1   | Day3   | Day5   | Day7   | Day9   | Day11  | Day13  | Day15  |
> | ----------- | ------ | ------ | ------ | ------ | ------ | ------ | ------ | ------ |
> | 100         | 101.04 | 174.45 | 251.28 | 319.26 | 331.13 | 433.29 | 402.50 | 397.14 |
> | 200         | 98.75  | 131.97 | 166.23 | 199.15 | 236.19 | 210.90 | 211.27 | 240.09 |
> | 400         | 99.28  | 111.71 | 124.77 | 131.81 | 146.62 | 142.99 | 139.23 | 152.17 |
> | 800         | 91.22  | 85.13  | 91.30  | 97.51  | 101.60 | 101.90 | 101.42 | 107.27 |
> | 1000        | 90.94  | 75.59  | 82.82  | 89.96  | 92.64  | 93.95  | 93.35  | 96.83  |
> | 2000        | 86.33  | 60.34  | 67.36  | 63.80  | 63.99  | 65.86  | 64.15  | 66.96  |
> | 4000        | 76.33  | 46.26  | 46.84  | 47.14  | 48.36  | 49.14  | 50.80  | 52.00  |
> | 8000        | 75.01  | 37.54  | 37.06  | 37.27  | 37.00  | 37.74  | 39.24  | 38.82  |
> | 10000       | 74.38  | 36.13  | 34.64  | 37.08  | 35.24  | 36.27  | 37.04  | 34.97  |
> | 20000       | 69.68  | 31.04  | 30.66  | 30.68  | 31.12  | 31.32  | 29.66  | 33.12  |
> | 40000       | 68.17  | 29.48  | 30.13  | 28.75  | 27.90  | 28.45  | 29.71  | 29.25  |
>
> Results of VAE-Var assimilation:
>
> | #Obs \ Time | Day1  | Day3   | Day5   | Day7   | Day9   | Day11  | Day13  | Day15  |
> | ----------- | ----- | ------ | ------ | ------ | ------ | ------ | ------ | ------ |
> | 100         | 89.84 | 138.32 | 256.51 | 263.31 | 259.13 | 262.76 | 292.72 | 305.22 |
> | 200         | 86.78 | 111.45 | 157.95 | 157.87 | 179.05 | 187.31 | 199.23 | 210.11 |
> | 400         | 78.94 | 79.23  | 96.86  | 102.26 | 115.18 | 126.07 | 127.70 | 130.34 |
> | 800         | 70.99 | 75.83  | 79.16  | 79.33  | 80.69  | 80.24  | 83.77  | 83.46  |
> | 1000        | 67.76 | 59.26  | 66.18  | 68.48  | 67.03  | 70.05  | 73.95  | 74.53  |
> | 2000        | 54.95 | 42.28  | 47.83  | 46.98  | 51.14  | 50.13  | 63.13  | 61.71  |
> | 4000        | 48.10 | 33.25  | 39.51  | 40.52  | 39.98  | 38.61  | 39.39  | 39.52  |
> | 8000        | 41.14 | 28.48  | 30.35  | 29.50  | 32.54  | 32.91  | 31.56  | 34.98  |
> | 10000       | 39.82 | 28.98  | 32.68  | 31.26  | 33.87  | 33.38  | 35.61  | 39.20  |
> | 20000       | 37.51 | 26.70  | 29.42  | 29.74  | 31.49  | 32.09  | 33.60  | 37.28  |
> | 40000       | 35.73 | 26.99  | 29.97  | 29.48  | 31.83  | 32.65  | 32.90  | 35.98  |

---

> ### Author Response · Authors · 2024-11-28
>
> **A4.1 (cont'd):** As shown in the table, when the number of observations is low (below 8,000), VAE-Var shows a clear improvement in analysis accuracy as the observation quantity increases. However, when the observation density exceeds 8,000, the improvement in accuracy becomes less pronounced. In contrast, 3DVar demonstrates more consistent improvement as the observation count increases. When the number of observations is below 10,000, VAE-Var generally outperforms 3DVar, with the advantage becoming more pronounced as the observations are sparser. However, when the number of observations exceeds 10,000, the performance of VAE-Var slightly lags behind 3DVar.
>
> **Q4.2: "Discuss potential reasons for this behavior, such as possible saturation effects or limitations in the model's capacity to incorporate additional information."**
>
> **A4.2:** First, we explain why VAE-Var shows a greater advantage over 3DVar with sparser observations. The primary improvement of VAE-Var lies in better characterizing background field errors, which enhances the overall accuracy of the assimilation system. Data assimilation works by balancing the background field and the observations. When the observation density is low, the accurate representation of the background field becomes crucial, and this is where VAE-Var offers significant advantages. However, as the observation density increases, the relative importance of background field modeling decreases, since the high-density observations themselves can provide sufficient information for assimilation. From VAE-Var to 3DVar to interpolation, their reliance on background information decreases and reliance on observational information increases, and that's why when the observation is dense, the advantage of VAE-Var is very little.
>
> Second, regarding why the improvement in analysis accuracy with higher observation density is less pronounced for VAE-Var, we think this is due to the fact that the increments in the analysis field are constrained within the latent space defined by the decoder output, $\mathcal{D}(\mathbf{z})$. During training, VAE may not fully generalize to the complete analysis increment space, meaning that certain finer details of the analysis may not be captured effectively. When observation density is high, achieving further improvement requires fine-grained adjustments to the analysis increments, which may not be well-represented in the latent space. As a result, the analysis accuracy may plateau once the background field error is already small. Currently, the RMSE for the z500 field has a lower bound of approximately $26 m^2/s^2$, which represents a good level of accuracy, but further improvements may be limited due to these modeling constraints.
>
> **Q4.3: "Analyze the implications of this behavior for the algorithm's practical use in different observational settings."**
>
> **A4.3:** Based on the above experimental results, VAE-Var demonstrates its strength in assimilating sparse observations, where observation sites account for less than 1% of the grid points. Assimilation of sparse observations and the reconstruction of physical fields are crucial challenges in both data assimilation and dynamical systems (Brajard et al., 2020; Rodgers, 2020; Geer, 2021). Therefore, our work has broad applicability in addressing these important issues.
>
> **Q4.4: "If possible, propose modifications to the algorithm that might address this limitation."**
>
> **A4.4:** There are two potential solutions to address this limitation. The first is to train a larger VAE neural network with more parameters and increase the number of training iterations. However, this would require more computational resources and might still face accuracy bottlenecks. The second approach is to combine the strengths of VAE-Var and 3DVar by creating a hybrid model. In this hybrid model, VAE-Var would be used to construct the analysis increments for sparse observations, while 3DVar would fine-tune the analysis increments for denser observations. We plan to explore both of these methods in future work.
>
> **Q5: “Formulating $p(y|x)$ as a Gaussian distribution is indeed an approximation of the observation errors.”**
>
> **A5:** Thank you for highlighting this important point. You are absolutely right. The reason we frame $p(y|x)$ as a Gaussian distribution in the paper is that our focus is primarily on site-based observations, where the relationship between observations and the physical field is often more interpolative in nature. As such, we referenced studies like Trémolet (2006) and DelSole & Yang (2010), which assume Gaussian statistics for $p(y|x)$. We acknowledge that satellite observations, such as those from the High Resolution Infrared Sounder, can indeed introduce non-Gaussianity, and that the Gaussian assumption is an approximation. This was an oversight on our part, and we will revise the manuscript to clarify this point.

---

> ### Author Response · Authors · 2024-11-28
>
> **Q6: Minor issues.**
>
> **Q6.1: "Line 202-2-3 and Algorithm 1 and Figure 1. Is τ the number of time steps or the time gap between two snapshots? It is a bit confusing here."**
>
> **A6.1:** Thank you for pointing this out. $\tau$ corresponds to the time gap between two snapshots in Line 202-2-3 and Algorithm 1 and Figure 1. We will make it clear in the revised version.
>
> **Q6.2: "Inconsistency between the lower half panel of Figure 1 and Algorithm 2: How is the hidden state initialized? 0 or a random field?"**
>
> **A6.2:** Thank you for pointing this out. The hidden state is initialized to zeros. There is an error in Figure 1, and we will correct it in the revised version.
>
> **Q6.3: "The model time step $T$ is not explicitly defined."**
>
> **A6.3:** Thank you for pointing this out. The time step $T$ represents the assimilation cycle. We apologize for not explicitly defining it in the manuscript, and will ensure this is clarified in the revised version.
>
> **Q6.4: "The last term in Equation (8) should be $\frac12\log\det\left(\frac{\partial \mathcal{D}(\mathbf{z})}{\partial \mathbf{z}}\right)^\mathrm{T}\left(\frac{\partial \mathcal{D}(\mathbf{z})}{\partial \mathbf{z}}\right)$."**
>
> **A6.4:** Thank you for pointing this out. We will revise it in the updated manuscript.
>
> **References:**
>
> [1] Descombes, G., Auligné, T., Vandenberghe, F., Barker, D. M., & Barre, J. (2015). Generalized background error covariance matrix model (GEN_BE v2. 0). *Geoscientific Model Development*, *8*(3), 669-696.
>
> [2] Melinc, B., & Zaplotnik, Ž. (2024). 3D‐Var data assimilation using a variational autoencoder. *Quarterly Journal of the Royal Meteorological Society*, *150*(761), 2273-2295.
>
> [3] Huang, L., Gianinazzi, L., Yu, Y., Dueben, P. D., & Hoefler, T. (2024). Diffda: a diffusion model for weather-scale data assimilation. *arXiv preprint arXiv:2401.05932*.
>
> [4] Rozet, F., & Louppe, G. (2023). Score-based data assimilation. *Advances in Neural Information Processing Systems*, *36*, 40521-40541.
>
> [5] Giaquinta, M., & Modica, G. (2010). *Mathematical analysis: An introduction to functions of several variables*. Springer Science & Business Media.
>
> [6] Trémolet, Y. (2006). Accounting for an imperfect model in 4D‐Var. *Quarterly Journal of the Royal Meteorological Society: A journal of the atmospheric sciences, applied meteorology and physical oceanography*, *132*(621), 2483-2504.
>
> [7] DelSole, T., & Yang, X. (2010). State and parameter estimation in stochastic dynamical models. *Physica D: Nonlinear Phenomena*, *239*(18), 1781-1788.
>
> [8] Brajard, J., Carrassi, A., Bocquet, M., & Bertino, L. (2020). Combining data assimilation and machine learning to emulate a dynamical model from sparse and noisy observations: A case study with the Lorenz 96 model. *Journal of computational science*, *44*, 101171.
>
> [9] Rodgers, C. D. (2000). *Inverse methods for atmospheric sounding: theory and practice* (Vol. 2). World scientific.
>
> [10] Geer, A. J. (2021). Learning earth system models from observations: machine learning or data assimilation?. *Philosophical Transactions of the Royal Society A*, *379*(2194), 20200089.

---

> ### Comment · Reviewer_jauN · 2024-11-30
> **Thanks for the authors' response**
>
> Thank you for the thorough and thoughtful rebuttal. I’m very impressed with the clarity and depth of your responses, and I feel that most of the minor points have been addressed in a satisfactory manner. And I appreciate the effort in explaining the practical considerations behind using ERA5 and the rationale for the mathematical simplifications.
>
> That said, the reliance on ERA5 as ground truth and the compromises in mathematical rigor, while understandable, are still notable. The rigorous methods to validate data assimilation algorithms already exist and have been used by the community for quite some time. The use of AI in this field should not serve as an excuse to sidestep established standards of rigor. Instead, it should be an opportunity to enhance established standards, ensuring both innovation and scientific credibility.
>
> Additionally, I don’t believe addressing the oversimplification issues by pointing to the limitations of previously published works is the best approach. While acknowledging prior challenges is helpful for context, I think it’s more impactful to focus on overcoming those limitations and setting a higher standard moving forward.
>
> I fully recognize the complexities involved and the progress your work represents. However, I feel that striving for greater rigor and innovation will further solidify AI’s role in transforming data assimilation.
>
> Thank you again for engaging in this discussion, and I look forward to seeing how your work continues to evolve. And of course I will increase the score : ).

---

> > ### Author Response · Authors · 2024-12-01
> >
> > Thank you for your thoughtful feedback and for increasing your score. We appreciate your recognition of our efforts to address the practical considerations and challenges in our work.
> >
> > We agree that maintaining rigor in data assimilation, especially with AI, is crucial, and we are committed to balancing innovation with established standards. Your comments will guide us as we continue to explore and refine our methods in this field.
> >
> > Thank you again for your valuable input, and we look forward to further advancing this research.

---

### Official Review · Reviewer_bhb7 · 2024-11-04

**Soundness:** 3
**Presentation:** 2
**Contribution:** 3
**Rating:** 6
**Confidence:** 4

**Summary:**

The study introduces VAE-Var, a neural network-based data assimilation method that combines VAE with traditional variational DA techniques. The authors claim that VAE-Var is designed to enhance DA by (i) Capturing non-Gaussian structures in background error distributions. (ii) Allowing effective assimilation of real-world, often irregularly distributed observations. Tests conducted using the FengWu weather model demonstrate that VAE-Var outperforms DiffDA and 3D-Var. Although the research topic is of interest, from the reviewer’s point of view, the novelty of the proposed approach is limited since (varitional-) autoencoder based data assimilation methods are already well established.

**Strengths:**

The VAE-Var approach can integrate partial and irregular observation points, giving it certain advantages over purely supervised machine learning surrogates for data assimilation. Numerical results show that the VAE-Var approach consistently outperforms both DiffDA and the conventional 3D-Var approach in terms of accuracy and efficiency.

**Weaknesses:**

The combination of neural network based AE (including VAE) and variational data assimilation is a well known approach. There are more than 10 publications regarding this idea in dynamical systems, for ex,
- Peyron, M., Fillion, A., Gürol, S., Marchais, V., Gratton, S., Boudier, P. and Goret, G., 2021. Latent space data assimilation by using deep learning. Quarterly Journal of the Royal Meteorological Society, 147(740), pp.3759-3777.
-Melinc, B. and Zaplotnik, Ž., 2024. 3D‐Var data assimilation using a variational autoencoder. Quarterly Journal of the Royal Meteorological Society, 150(761), pp.2273-2295.
- Cheng, S., Chen, J., Anastasiou, C., Angeli, P., Matar, O.K., Guo, Y.K., Pain, C.C. and Arcucci, R., 2023. Generalised latent assimilation in heterogeneous reduced spaces with machine learning surrogate models. Journal of Scientific Computing, 94(1), p.11.
- Mohd Razak, S., Jahandideh, A., Djuraev, U. and Jafarpour, B., 2022. Deep learning for latent space data assimilation in subsurface flow systems. SPE Journal, 27(05), pp.2820-2840.
- Zhong, C., Cheng, S., Kasoar, M. and Arcucci, R., 2023. Reduced-order digital twin and latent data assimilation for global wildfire prediction. Natural hazards and earth system sciences, 23(5), pp.1755-1768.
…..

The proposed approach uses ground truth samples for background error specification. This might be a valid approach in twin experiments. However, ground truth is out of reach in most applications of DA methods, such as weather prediction. In other words, we never know the historical ‘true’ weather, as observations also contain uncertainties. I would suggest focusing on the innovation quantity (i.e., y-H(xb) or y-H(xa)) for a propre error covariance specification/analysis, see
-Tandeo, P., Ailliot, P., Bocquet, M., Carrassi, A., Miyoshi, T., Pulido, M. and Zhen, Y., 2020. A review of innovation-based methods to jointly estimate model and observation error covariance matrices in ensemble data assimilation. Monthly Weather Review, 148(10), pp.3973-3994.

**Questions:**

The authors should also compare their approach to ensemble-based data assimilation algorithms, such as EnKF, which are known to yield superior results for chaotic dynamical systems.

The role of the parameter lambda needs further explanation and why lambda = 4 is chosen in this paper?

---

> ### Author Response · Authors · 2024-11-28
>
> We sincerely appreciate your careful and thoughtful comments and time. We try to explain your concerns point by point.
>
> **Q1: "Although the research topic is of interest, from the reviewer’s point of view, the novelty of the proposed approach is limited since (varitional-) autoencoder based data assimilation methods are already well established. There are more than 10 publications regarding this idea in dynamical systems."**
>
> **A1:** Thank you for your valuable feedback. We appreciate your insights into the related work, and we would like to clarify some important distinctions between our approach and the existing literature, particularly in regard to the use of variational autoencoders (VAEs) in data assimilation.
>
> First, it is essential to emphasize that our proposed framework fundamentally differs from the latent space assimilation methods you mentioned. The works you referenced follow the latent space assimilation paradigm, where the goal is to reduce the dimensionality of the physical space by training a neural network (such as a VAE) to map high-dimensional physical fields to a lower-dimensional latent space. In this approach, traditional data assimilation methods (e.g., 3DVar, ensemble Kalman filter) are then applied in the latent space, and the solution is mapped back to the physical space. The primary motivation behind these approaches is to reduce computational cost, rather than improving the accuracy of the assimilation process itself.
>
> We believe there are three main issues with latent space-based data assimilation algorithms. First, latent space-based data assimilation algorithms mostly focus on improving efficiency in their modeling process, and as a result, it is challenging for them to bring any gain in the accuracy of the assimilation algorithm. Second, with the exception of a few works like Mack et al. (2020), most of the existing literature lacks a clear theoretical explanation, such as how the results of assimilation in the latent space relate to those obtained from direct assimilation in the physical space. For example, in the work of Melinc and Zaplotnik (2024), they train a VAE to reduce the dimensionality of the physical field and perform data assimilation in the latent space. A clear flaw in this approach is the assumption that the background error covariance matrix $\mathbf{B}_z$ in the latent space can be diagonal. The VAE only learns dimensionality reduction and orthogonalization for the physical field, but it does not learn the corresponding reduction and orthogonalization of the physical field errors. Consequently, when the forecast model changes, the VAE that originally orthogonalized the physical field errors may no longer be able to orthogonalize the new errors. Thus, this work fails to address the theoretical challenges of latent space assimilation. Third, to our knowledge, none of these works have been tested on real-world weather systems to evaluate the effectiveness of the algorithms. Even in the case of Melinc and Zaplotnik (2024), while ERA5 is used as observations, the forecast model employs a trivial identity model where the input equals the output.
>
> **In contrast, the motivation of our work is not dimensionality reduction but rather improving the accuracy of traditional data assimilation algorithms.** Specifically, we observe that in conventional assimilation methods, the background error distribution is modeled as Gaussian, using a covariance matrix $\mathbf{B}$. However, in real-world systems, this distribution is often non-Gaussian. Our goal is to train a VAE to learn this non-Gaussian distribution, thereby improving the representation of the background errors. This distinction is crucial: in our approach, the VAE is used to learn the distribution of the background errors, whereas other works (such as Melinc and Zaplotnik, 2024) use the VAE for dimensionality reduction and latent space modeling. We strictly follows the Bayesian modeling and derive the variational assimilation function in the original physical space under the assumption of VAE-represented background error distributions (as shown in the Appendix). We also identify the presence of a Jacobian determinant term in the assimilation function. Although we do not solve the variational function with the Jacobian determinant term explicitly in this paper, the mathematical foundation of our approach is rigorous. Moreover, our experiments show that even with the approximation of replacing the Jacobian determinant term with a lambda term, the assimilation results remain ideal and surpass the performance of traditional methods. Therefore, both theoretically and experimentally, our work stands out as pioneering.

---

> ### Author Response · Authors · 2024-11-28
>
> **A1 (cont'd):** Summing up, our innovations are significant both theoretically and experimentally. **From a theoretical standpoint, we are the first to establish the variational assimilation function under the VAE representation of the background error distribution. Furthermore, experimentally, we have demonstrated the validity of our approximations through real-world experiments assimilating GDAS observations at a 0.25-degree resolution.** Therefore, our work represents a highly innovative contribution to the field.
>
> We hope this clarification better highlights the novelty and distinction of our approach. We look forward to any further questions or suggestions you may have.
>
> **Q2: "Ground truth is out of reach in most applications of DA methods, such as weather prediction. In other words, we never know the historical ‘true’ weather, as observations also contain uncertainties. I would suggest focusing on the innovation quantity (i.e., y-H(xb) or y-H(xa)) for a propre error covariance specification/analysis"**
>
> **A2:** Thank you for your insightful suggestion. We agree that using innovation quantities (i.e., y - H(x_b) or y - H(x_a)) to specify and analyze error covariance is a theoretically sound and more precise approach, particularly from a Bayesian modeling perspective. This method allows for the joint estimation of both model error covariance ($\mathbf{Q}$) and observation error covariance ($\mathbf{R}$). However, we would like to highlight a few important challenges when applying this approach to high-dimensional systems.
>
> One significant limitation of innovation-based methods is that they are difficult to scale to high-dimensional systems, as mentioned in the review (Tandeo et al., 2020). For instance, Dee (1995) pointed out that for such algorithms to be effective, the number of observations must exceed the number of adjustable parameters in the covariance matrix by at least two to three orders of magnitude. In practical scenarios, the number of observations typically ranges on the order of $10^5$, which means that, if observations must exceed the number of adjustable parameters by two orders of magnitude, the number of adjustable parameters can only be in the thousands. For a neural network, training on a system with a resolution of $69 \times 1440 \times 721$ with only a few thousand parameters would be practically impossible.
>
> Additionally, to the best of our knowledge, most global-scale high-resolution meteorological AI data assimilation studies, including works by Melinc & Zaplotnik (2024), Huang et al. (2024), Xiao et al. (2023), and Li et al. (2024), have relied on ERA5 as a reference for background field construction. This highlights the inherent challenges in fully decoupling from ERA5 in the early stages of AI-based data assimilation research. We believe that leveraging ERA5 for background field construction at this stage is both reasonable and justifiable.
>
> Also, from the perspective of our algorithm framework, our algorithm has strong potential to break free from the reliance on ERA5. In VAE-Var, the NMC method essentially utilizes the ERA5 data to generate uncertainty/error samples for FengWu forecasts. In the field of variational data assimilation, there are many alternative approaches that do not require analysis fields. For example, ensemble forecast (Pereira & Berre, 2006) can be used instead of the NMC method to generate these samples. Although this approach might be more complex to implement, it is theoretically feasible. In our future work, we will conduct further detailed explorations in this direction.

---

> ### Author Response · Authors · 2024-11-28
>
> **Q3: "The authors should also compare their approach to ensemble-based data assimilation algorithms, such as EnKF, which are known to yield superior results for chaotic dynamical systems."**
>
> **A3:** Thank you for your suggestions. To our knowledge, both variational (e.g., 3DVar, 4DVar) and ensemble-based methods (e.g., EnKF, LETKF) are widely used in the field, and there is no definitive consensus on which approach is superior for chaotic dynamical systems. Different meteorological centers have adopted different algorithms; for example, the UK Met Office uses 3DVar and 4DVar in its operational systems (Hu, et al., 2023).
>
> Furthermore, the performance of assimilation algorithms is highly dependent on the choice of parameters. As of now, there are few open-source variational and ensemble-based assimilation algorithms corresponding to the physical quantities of ERA5. Therefore, in our paper, we use the best-tuned results from our own implementation of 3DVar as the baseline of traditional algorithms for comparison.
>
> As a supplementary experiment, we conducted an additional experiment using the local ensemble transform kalman filter (LETKF) algorithm (Hunt, et al., 2007), following the approach outlined in DIDA (Li et al., 2023). In the LETKF algorithm we implemented, a total of 32 ensemble members were used. The initial ensemble was obtained from the publicly available ensemble members provided by the ECMWF website. The horizontal localization radius was set to 200 km, and the vertical localization coefficient was set to 0.3. The assimilation system ran for a total of 15 days. The results of this experiment are presented in the table below. The table below shows the RMSE ($m^2/s^2$) of the Z500 analysis field every two days at midnight for both LETKF, 3DVar, VAE-Var across three different observational settings. It can be found that the performance of LETKF is slightly worse than 3DVar, and VAE-Var outperforms both LETKF and 3DVar.
>
> #Obs = 1000
>
> | Method \ Time | Day1   | Day3   | Day5   | Day7   | Day9   | Day11 | Day13 | Day15 |
> | ------------- | ------ | ------ | ------ | ------ | ------ | ----- | ----- | ----- |
> | 3DVar         | 90.93  | 75.58  | 82.81  | 89.95  | 92.63  | 93.94 | 93.34 | 96.83 |
> | LETKF         | 169.88 | 131.99 | 113.61 | 116.97 | 110.56 | 97.25 | 94.89 | 79.00 |
> | VAE-Var       | 67.76  | 59.25  | 66.17  | 68.47  | 67.03  | 70.05 | 73.94 | 74.53 |
>
> #Obs = 2000
>
> | Method \ Time | Day1   | Day3   | Day5   | Day7   | Day9   | Day11 | Day13 | Day15 |
> | ------------- | ------ | ------ | ------ | ------ | ------ | ----- | ----- | ----- |
> | 3DVar         | 86.32  | 60.34  | 67.35  | 63.80  | 63.98  | 65.85 | 64.14 | 66.95 |
> | LETKF         | 174.21 | 102.80 | 110.78 | 114.38 | 107.41 | 99.56 | 86.16 | 92.02 |
> | VAE-Var       | 54.95  | 42.28  | 47.82  | 46.97  | 51.13  | 50.12 | 63.12 | 61.70 |
>
> #Obs = 8000
>
> | Method \ Time | Day1   | Day3   | Day5  | Day7  | Day9  | Day11 | Day13 | Day15 |
> | ------------- | ------ | ------ | ----- | ----- | ----- | ----- | ----- | ----- |
> | 3DVar         | 75.01  | 37.53  | 37.06 | 37.27 | 36.99 | 37.74 | 39.24 | 38.81 |
> | LETKF         | 172.73 | 113.33 | 74.86 | 45.14 | 40.58 | 57.36 | 48.02 | 39.17 |
> | VAE-Var       | 41.14  | 28.47  | 30.34 | 29.49 | 32.54 | 32.90 | 31.56 | 34.98 |

---

> ### Author Response · Authors · 2024-11-28
>
> **Q4: "The role of the parameter lambda needs further explanation and why lambda = 4 is chosen in this paper?"**
>
> **A4:** Thank you for your thoughtful comment. As mentioned in the Appendix, the parameter $\lambda$ arises from the derivation based on Bayesian theory and measure theory. Specifically, we found that the variational function based on VAE-Var introduces a Jacobian determinant term, which represents the scaling factor of volume elements when transforming from the latent space to the physical space. However, due to the high dimensionality of the Jacobian matrix of the neural network, it is impractical to compute this term directly. Therefore, we opted to omit this term and, as compensation, introduced a scaling factor $\lambda$ in the background error term.
>
> To provide some intuition, let us explain the role of $\lambda$ with the following analogy: Suppose the physical space has a dimensionality of 100, while the latent space has a dimensionality of 10. Since the observation deviation term $\mathcal{L}_o\left(\mathcal{D}(\mathbf{z}) + \mathbf{x}_b, \mathbf{y}\right)$ is calculated in the physical space, and the background error term $\mathcal{L}_b(\mathbf{z}) = \frac{1}{2} \mathbf{z}^\mathrm{T} \mathbf{z}$ is calculated in the latent space, the former would intuitively be 100 times larger than the latter. To balance this dimensionality difference, we introduce the coefficient $\lambda$ to scale the background error term in the latent space, such that $\mathcal{L}_b(\mathbf{z}) = \frac{1}{2} \lambda \mathbf{z}^\mathrm{T} \mathbf{z}$.
>
> The parameter $\lambda$ is thus approximately equal to $\left(\frac{d_x}{d_z}\right)^2$, where $d_x$ and $d_z$ are the dimensions of the physical space and the latent space, respectively. In our experimental setup, the physical space has dimensions $d_x = 69 \times 721 \times 1440$ and the latent space has dimensions $d_z = 32 \times 721 \times 1440$. This gives $\lambda \approx 4.64$, and we round this value to 4 for simplicity.
>
> Additionally, we provide a comparison of assimilation results with different values of $\lambda$. The table below shows the RMSE ($m^2/s^2$) of the Z500 analysis field every two days at midnight. As can be seen, the assimilation results are optimal when $\lambda = 4$, which further supports our choice of this value.
>
> | $\lambda$ \ Time | Day1  | Day3   | Day5   | Day7   | Day9   | Day11  | Day13  | Day15  |
> | ---------------- | ----- | ------ | ------ | ------ | ------ | ------ | ------ | ------ |
> | 1.0              | 51.84 | 105.71 | 165.08 | 161.35 | 162.36 | 164.67 | 242.43 | 211.77 |
> | 4.0              | 54.95 | 42.28  | 47.82  | 46.97  | 51.13  | 50.12  | 63.12  | 61.70  |
> | 9.0              | 61.77 | 49.87  | 60.76  | 61.96  | 65.68  | 64.56  | 67.74  | 76.55  |
> | 16.0             | 66.86 | 55.85  | 77.37  | 79.53  | 84.44  | 86.30  | 86.53  | 95.61  |
> | 25.0             | 70.06 | 66.50  | 92.50  | 100.10 | 106.70 | 108.66 | 109.07 | 122.64 |

---

> ### Author Response · Authors · 2024-11-28
>
> **References:**
>
> [1] Mack, J., Arcucci, R., Molina-Solana, M., & Guo, Y. K. (2020). Attention-based convolutional autoencoders for 3d-variational data assimilation. *Computer Methods in Applied Mechanics and Engineering*, *372*, 113291
>
> [2] Melinc, B., & Zaplotnik, Ž. (2024). 3D‐Var data assimilation using a variational autoencoder. *Quarterly Journal of the Royal Meteorological Society*, *150*(761), 2273-2295.
>
> [3] Dee, D. P. (1995). On-line estimation of error covariance parameters for atmospheric data assimilation. *Monthly weather review*, *123*(4), 1128-1145.
>
> [4] Tandeo, P., Ailliot, P., Bocquet, M., Carrassi, A., Miyoshi, T., Pulido, M., & Zhen, Y. (2020). A review of innovation-based methods to jointly estimate model and observation error covariance matrices in ensemble data assimilation. *Monthly Weather Review*, *148*(10), 3973-3994.
>
> [5] Huang, L., Gianinazzi, L., Yu, Y., Dueben, P. D., & Hoefler, T. (2024). Diffda: a diffusion model for weather-scale data assimilation. *arXiv preprint arXiv:2401.05932*.
>
> [6] Xiao, Y., Bai, L., Xue, W., Chen, K., Han, T., & Ouyang, W. (2023). Fengwu-4dvar: Coupling the data-driven weather forecasting model with 4d variational assimilation. *arXiv preprint arXiv:2312.12455*.
>
> [7] Li, Y., Han, W., Li, H., Duan, W., Chen, L., Zhong, X., ... & Sun, X. (2024). Fuxi‐en4dvar: An assimilation system based on machine learning weather forecasting model ensuring physical constraints. *Geophysical Research Letters*, *51*(22), e2024GL111136.
>
> [8] Pereira, M. B., & Berre, L. (2006). The use of an ensemble approach to study the background error covariances in a global NWP model. *Monthly weather review*, *134*(9), 2466-2489.
>
> [9] Hu, G., Dance, S. L., Bannister, R. N., Chipilski, H. G., Guillet, O., Macpherson, B., ... & Yussouf, N. (2023). Progress, challenges, and future steps in data assimilation for convection‐permitting numerical weather prediction: Report on the virtual meeting held on 10 and 12 November 2021. *Atmospheric Science Letters*, *24*(1), e1130.
>
> [10] Hunt, B. R., Kostelich, E. J., & Szunyogh, I. (2007). Efficient data assimilation for spatiotemporal chaos: A local ensemble transform Kalman filter. *Physica D: Nonlinear Phenomena*, *230*(1-2), 112-126.
>
> [11] Li, Y., Ju, X., Xiao, Y., Jia, Q., Zhou, Y., Qian, S., ... & Xue, W. (2023, November). Rapid simulations of atmospheric data assimilation of hourly-scale phenomena with modern neural networks. In *Proceedings of the International Conference for High Performance Computing, Networking, Storage and Analysis* (pp. 1-13).

---

> > ### Comment · Reviewer_bhb7 · 2024-11-29
> >
> > I thank the authors for addressing my comments and performing additional experiments. They have adequately addressed my concerns about benchmarking against KF-type methods and conducting ablation tests for the lambda parameter. However, I am still not fully convinced by the authors' claim that they have addressed the challenge of non-gaussian prior background error. I would be happy to raise my score if they could provide more details regarding this point.
> >
> > If I understand correctly (pelase correct me inc case of any misunderstanding), the key point is to use VAE to transfer non-guassian prior error distribution in the full physics space (as shown in the appendix A.2) to a guassian distribution in the latent space when performing DA. However, I think this is not the ‘ same guassianity’ required in typical DA problems.  In a ‘twin experiment’ set up (as the authors performed with ERA5), supposing the true state $x_{true}$ at a given time step and the prior error distribution is known (either guassian or not), we can sample an ensemble of background states  $x_b^i$ with i = 1,…,n. What VAE does is to encode each single background state (or background error as the authors performed) $x_b^i$ to a guassian distribution in the latent space $N(\mu^i,\sigma^i)$. A sampled vector $z^i$ (with fixed i) could be generated with this distribution. However, there is no guarantee that the distribution of all  $ \mu^i $ or $ \z^i $ (i=1,…,n) is Guassian (Maybe some numerical experiments on the ERA5 dataset could be helpful for verification). In other words, VAE generate a Guassian mixture instead of a single gaussian distribution in the latent space from the ensemble $x_b^i$ (i=1,…,n) while in typical DA, the assumption is $x_{true}-x_b$ follows a guassian distribution (i.e, all x_b^i could be considered as samples generated from the same gaussian distribution, which, from the reviewer's point of view will not the be the case in the latent space of VAE-Var).
> >
> > One solution to enhance the Gaussian distribution in the latent space might be simply add a gaussian constraint for the whole training batch in the autoencoder but this may require a very large batch size to train and it could also be done with a deterministic autoencoder.

---

> ### Author Response · Authors · 2024-11-30
>
> We would like to thank the reviewer for their thoughtful feedback and for acknowledging the additional experiments we conducted. Regarding your concern about the non-Gaussian prior background error, we would like to offer a more detailed explanation. We believe there may be some confusion regarding the purpose of the VAE training process and certain statistical approximations used during training.
>
> **Brief Overview of VAE’s Principle**
>
> First, we would like to briefly restate the core principle of the Variational Autoencoder (VAE). The primary goal of VAE is to learn a deep latent variable model, which is designed to capture complex distributions that are difficult to express analytically (e.g. expressed as Gaussian, Gamma, etc.). While the true data distribution is challenging to model directly, the latent variable model introduces an auxiliary variable, denoted as $\mathbf{z}$, whose distribution is pre-specified (typically a standard normal distribution). The goal is then to learn the transformation $p(\mathbf{x}|\mathbf{z})$ that maps the latent variable $\mathbf{z}$ to the observed data $\mathbf{x}$, such that:
> $$
> p\_\mathbf{x}(\mathbf{x}) = \int p(\mathbf{x}|\mathbf{z})p\_\mathbf{z}(\mathbf{z})\mathrm{d}\mathbf{z}
> $$
>
> In the context of data assimilation, $p\_\mathbf{x}(\mathbf{x})$ on the left hand side represents the background error distribution, which we aim to estimate. The distribution of the latent variable $\mathbf{z}$ is assumed to be a standard normal distribution, $\mathcal{N}(\mathbf{0}, \mathbf{I})$, and the only part that we need to learn is the transformation $p(\mathbf{x}|\mathbf{z})$, which is realized by the decoder $\mathcal{D}$. The current problem setup involves a set of background error samples $\mathbf{x}\_1, \mathbf{x}\_2, \dots, \mathbf{x}\_N$ generated using an NMC method. These samples represent realizations of $p_\mathbf{x}(\mathbf{x})$. Our task is to learn the transformation $p(\mathbf{x}|\mathbf{z})$, i.e., the decoder $\mathcal{D}$. Once this is learned, the decoder can be directly used in the VAE-Var framework, where the distribution of the latent variable $\mathbf{z}$ remains a standard Gaussian distribution.
>
> VAE is an **approximation algorithm** designed to solve this deep latent variable problem. Intuitively, for each background error sample $\mathbf{x}\_i$, it is challenging to know exactly which latent sample $\mathbf{z}$ it corresponds to. Rather than explicitly computing $p_\mathbf{x}(\mathbf{x})$, which would be computationally intractable, VAE cleverly assumes that, given $\mathbf{x}\_i$, the distribution $p(\mathbf{z}|\mathbf{x}\_i)$ can be approximated by a Gaussian distribution, $q(\mathbf{z}|\mathbf{x}\_i) = \mathcal{N}(\mu(\mathbf{x}\_i), \Sigma(\mathbf{x}\_i))$, and uses a neural network to learn the mean and covariance of this Gaussian. The VAE’s training objective then optimizes the Evidence Lower Bound (ELBO), significantly reducing the computational burden. In this context, the encoder is essentially an approximation step for learning the decoder.
>
> For more details on the deep latent variable part, we refer to Kingma and Welling (2019), Section 1, and for the statistical approximations made in VAE, we refer to Section 2 of the same paper.
>
> **Response to Your Concerns**
>
> From the above, we can clarify the following points:
>
> 1. **VAE’s goal is to learn the decoder $\mathcal{D}$,** and the latent variable $\mathbf{z}$ is assumed to follow a standard Gaussian distribution, which is consistent with the VAE-Var framework.
> 2. **The idea that VAE’s latent space is a mixture of Gaussians** is not entirely accurate. The latent variable $\mathbf{z}$ is conditionally distributed as $p(\mathbf{z}|\mathbf{x}\_i)$ given the sample $\mathbf{x}\_i$, and in VAE this is approximated by a (non-standard) Gaussian distribution $q(\mathbf{z}|\mathbf{x}\_i) = \mathcal{N}(\mu(\mathbf{x}\_i), \Sigma(\mathbf{x}\_i))$. However, the **full distribution** of $\mathbf{z}$, i.e., $p\_\mathbf{z}(\mathbf{z})$, remains a standard normal distribution (at least theoretically assumed to be). It is true that if we iterate through all the dataset samples $\\{\mathbf{x}_i\\}\_{i=1}^n$，$\sum\_{i=1}^n \frac1N q(\mathbf{z}|\mathbf{x}\_i)$ is mixture of Gaussian, but it is indeed different from $p\_\mathbf{z}(\mathbf{z})=\int p(\mathbf{z}|\mathbf{x}) p\_\mathbf{x}(\mathbf{x}) \mathrm{d} \mathbf{x}.$
> 3. **Regarding your suggestion to add a Gaussian constraint to the entire training batch**, we agree that this could be a possible solution. However, it would be computationally expensive. The VAE architecture already addresses this by including a KL-divergence term, $KL[p(\mathbf{z}|\mathbf{x}) | p\_\mathbf{z}(\mathbf{z})]$, which ensures that the distribution $p(\mathbf{z}|\mathbf{x})$ is as close as possible to a standard Gaussian. This avoids the need for sampling from all training data and is one of the key advantages of the VAE architecture.

---

> > ### Author Response · Authors · 2024-11-30
> >
> > **Experimental Verification on FengWu-ERA5 Background Error**
> >
> > Following the discussion above, we would like to provide empirical evidence for the standard Gaussianity of the latent space. After training the VAE on ERA5 background error samples $\mathbf{x}_i$, we focus on the mean part of the latent variables $\mu(\mathbf{x}_i)$, which corresponds to the expected value of the latent variable for each background error sample.
> >
> > We selected 1000 ERA5 samples, generated at 6-hour intervals starting from January 1, 2014. These samples were obtained using the NMC method, with a 1.4-degree resolution and a horizontal discretization of $128 \times 256$. To examine the structure of the latent space, we computed the covariance matrix of the latent variables. Due to the high dimensionality of the latent space ($32 \times 128 \times 256$), directly calculating the full covariance matrix is computationally impractical. Instead, we focused on computing the covariance along one dimension (either 32, 128, or 256) while averaging over the other two dimensions. The resulting covariance matrices for each dimension are shown in anonymous links (https://imgur.com/A0SnWZw), (https://imgur.com/YPl6Sem), and (https://imgur.com/WA1KD6Z).
> >
> > The visualizations reveal that the three covariance matrices are nearly identical, with the diagonal entries predominantly close to 1 (with some values around 0.5, which are still significantly larger than the non-diagonal entries), and the non-diagonal entries approximating 0. This suggests that the encoder has effectively mapped the background error samples into a nearly orthogonal space, providing strong evidence that the encoder is performing as expected.
> >
> > **Reference:**
> >
> > Kingma, D. P., & Welling, M. (2019). An introduction to variational autoencoders. *Foundations and Trends® in Machine Learning*, *12*(4), 307-392

---

> > > ### Comment · Reviewer_bhb7 · 2024-11-30
> > >
> > > I thank the authors for the thorough explanation and extra numerical experiments. I have increased my score accordingly.
> > >
> > > I also notice that the error samples during the training phase are generated across a different range of predictions, as the authors stated in algorithm 1. Therefore, I believe the authors only use the 'ERA5 ground truth' to evaluate the DA performance, rather than employing it to construct the background matrix. This is indeed a novelty in latent DA algorithms, combining an operational error quantification technique. However, the first part of fig 1 could be quite misleading. I would suggest write explicitly ‘NMC’ and use ‘prediction difference’ instead of ‘Error samples’. The ‘ground truth samples’ are only used to initialize the predictions in NMC? If so, maybe use, for ex, ‘current analysis state’ ?
> > >
> > > Regarding the latent space geometry, the extra experiments indeed numerically show the independence of latent features. However, no conclusion could be made regarding the Gaussianity of \mu(x). I do understand that processing the ERA5 dataset could be computationally impractical. The authors could consider adding a latent space analysis by applying VAE-VAR to the Lorenz system (appendix A.2 where they show background statistics) as this is an important aspect of their contribution.

---

> > > > ### Author Response · Authors · 2024-12-01
> > > >
> > > > Thank you for your recognition of our work.
> > > >
> > > > Regarding the first point, you are correct. In our approach, the NMC method is used to compute the forecast differences to generate the error samples required for training, and the ERA5 dataset is only used to initialize the forecasts. We will update Figure 1 in the revised version to make this clearer.
> > > >
> > > > For the second point, thank you for the suggestion. We will include an analysis of the latent space Gaussianity by applying VAE-Var to the Lorenz63 system, as shown in Appendix A.2, and we will place this analysis in the appendix of the revised version of the paper.

---

### Official Review · Reviewer_2mtD · 2024-11-04

**Soundness:** 3
**Presentation:** 4
**Contribution:** 3
**Rating:** 8
**Confidence:** 2

**Summary:**

This paper proposes a new Variational Data Assimilation method based on Variational Autoencoders, called VAE-Var. The method is separated into two stages, the training and the assimilation stage. During the training stage, a dataset of error samples is constructed by comparing two forecasts made at different time intervals. These samples are then used to train the VAE model, which learns the error distribution of the background states. During the assimilation stage, the VAE decoder is used to map latent vectors back to the physical state, that when combined to the background field represents the analysis states after data assimilation. The proposed methodology is compared against DiffDA, 1DVAR, and interpolation for the FengWu forecasting model and show superior performance across benchmarks

**Strengths:**

The paper is well-written, and clearly communicates the goal. The authors also explain the shortcomings of the other methods and how they propose to overcome them. The methods section is clear and the results show that VAE-Var out-performs all baselines.

**Weaknesses:**

There is not an obvious weakness to the paper. However I am not an expert in meteorology so maybe I missed something.

**Questions:**

- In my understanding, Diffusion models can capture more complex distributions but require a lot of data. Could the authors provide more information on how fair the comparison between the VAE and the Diffusion model is? Meaning that if the distribution that they model is not  complex and the number of available data points is not large enough, isn't Diffusion models expected to perform worse?

---

> ### Author Response · Authors · 2024-11-28
>
> We sincerely appreciate your careful and thoughtful comments and time, especially your approval for our proposed framework, which really means a lot to us. We try to explain your concerns point by point.
>
> **Q1: Could the authors provide more information on how fair the comparison between the VAE and the Diffusion model is?**
>
> **A1:** We appreciate the reviewer’s interest in the fairness of our comparisons. We believe that the comparison between VAE-Var and DiffDA is fair, as the experimental setups and data sources are carefully aligned:
>
> 1. **Training Data:** Both VAE-Var and DiffDA were trained using the same dataset, namely ERA5 reanalysis data spanning 37 years from 1979 to 2015.
> 2. **Assimilation Configuration:** The assimilation settings for both methods are identical.
>    - **Initialization:** The starting date for assimilation is January 1, 2022, and the initial conditions are derived by conducting a 48-hour forecast based on ERA5.
>    - **Observations:** Observations are randomly sampled from simulated ERA5 data, using sets of 1,000, 2,000, 4,000, and 8,000 observation points.
>    - **Assimilation Period:** Both methods were evaluated over the same 15-day assimilation period.
> 3. **Forecasting Models:** The only difference is in the forecast models used—DiffDA is built on GraphCast, whereas VAE-Var uses the FengWu model. However, both GraphCast and FengWu are state-of-the-art forecasting models with comparable performance levels, so we expect this difference to have minimal impact on the results.
>
> Finally, our experimental results demonstrate a high level of statistical significance in favor of VAE-Var. This suggests that the observed improvements in analysis accuracy are primarily driven by advancements in the assimilation algorithm rather than differences in forecast models.
>
> **Q2: If the distribution that they model is not complex and the number of available data points is not large enough, isn't Diffusion models expected to perform worse?**
>
> **A2:** Thank you for this thoughtful question. We agree with the general assumption that as models become more complex and datasets larger, diffusion-based methods are likely to achieve better results. However, this scenario does not align with the characteristics of weather data assimilation, and our work does not artificially create a setting that favors VAEs.
>
> In the domain of weather data assimilation, the size of the dataset is inherently limited. The ERA5 dataset, for example, provides only 37 years of training data, totaling tens of thousands of samples, which is significantly smaller than datasets commonly seen in computer vision. This fundamental limitation makes it challenging for data-intensive models like diffusion to fully realize their potential without overfitting.
>
> Additionally, an important property in the data assimilation field is that the background error $\mathbf{x} - \mathbf{x}_b$ is independent of $\mathbf{x}_b$. Attempting to model the full conditional distribution $p(\mathbf{x}|\mathbf{x}_b)$, while theoretically possible, introduces unnecessary complexity and increases the risk of overfitting.
>
> The key contributions of our work lie in recognizing these two domain-specific characteristics of weather data assimilation:
>
> 1. **Simplified Target Distribution:** By leveraging the independence of  $\mathbf{x} - \mathbf{x}_b$ from $\mathbf{x}_b$, we focus on modeling the simpler distribution of $\mathbf{x} - \mathbf{x}_b$ instead of the more complex $p(\mathbf{x}|\mathbf{x}_b)$.
> 2. **Lightweight Model Selection:** Considering the limited size of weather datasets, we adopt a lightweight VAE model that is less prone to overfitting compared to more complex alternatives like diffusion models.
>
> In summary, while diffusion-based methods may perform better in other data assimilation problems or for different weather-related tasks with larger datasets, for the specific case of weather data assimilation with limited samples, we believe our framework offers a more suitable and effective solution.
>
> We hope this addresses your concerns and further clarifies the rationale behind our approach. Thank you again for this insightful comment.

---

> ### Comment · Reviewer_2mtD · 2024-12-02
>
> Thank you very much for you comment and the comprehensive response. It answered my question.

---

> > ### Author Response · Authors · 2024-12-03
> >
> > Thank you for your recognition of our work.

---

### Official Review · Reviewer_D5gS · 2024-11-07

**Soundness:** 3
**Presentation:** 4
**Contribution:** 3
**Rating:** 8
**Confidence:** 5

**Summary:**

The paper presents VAE-Var a new data assimilation algorithm based on variational assimilation in order to capture non-Gaussian characteristics of the conditional background distribution p(x|x_b) and thus enhance the accuracy of the method. The algorithm also able to incorporate real measurements coming form different systems.

The theory is clear. The architecture is well justified. The experiments too. The results of the main paper are very interesting, the one in the supplementary material bring some nuances.

The experiments rely on FengWu, an adaptation is done to construct the encoder and the decoder. ERA5 reanalysis data from 1979 to 2015 are used as training set. Two types of observations are used : simulated one from ERA5 to mimic real-world observations, and GDAS prepbufr with various system and instruments. The initial setting is derived from ERA5 dataset. For ERA5 simulations, fixed or unfixed observations are used. For fixed observations VAE-Var outperforms the other methods  in the main paper. In the supplementary material figure 8, sometimes 3DVar perform better than VAE-Var as for mslpRSME with 8000 observations. Graphically speaking, VAE-Var performs closely to 3DVar. It is worth noting that VAE-Var performs better most of the time with 1000 observations.
For GDAS, the RMSE is always smaller for VAE-Var. Bias is always better, except for z500.

All in all, this is a good paper.

**Strengths:**

In section 3.1.the use of the hypothesis of independence between the background states x-x_ b and background state x_b is well supported. The same goes for the non-Gaussian error distribution assumption.

The justification of the use VAE is clear. The generation of the training set construction is well formulated, the same goes for the loss function.

The authors are honest regarding their results, especially regarding the limitations in supplementary material A.2.

VAE-Var focuses learning the latent space of the error fields. This is an AI forecasting model effective at global scale with real-world measurements.

**Weaknesses:**

Limitations :

The performance presented in the main paper show very interesting results regarding VAE-Var. It must be nuanced with the supplementary material figure for broader considerations.

The authors mention in the supplementary material A.2. that they do not consider the determinant term in the equation 8 because it would make the assimilation of the objective function extremely hard to optimize. Thus, they introduce a scaling factor in the background term of the assimilation objective function. This is honesty from the authors. Maybe some tricks can be found later to make it rigorous.

Minor comments that do not impact the score:

Line 243-244, in Chen et al., the dimensionality of the problem is 189 * 721 * 1440. What does the 69 correspond to? Is it a confusion with what is used in the present study section 4.1?

Line 262-263, the authors claim that all the components of L are differentiable. Why H is differentiable?

There is no 2.5.3. in Giaquinta & Modica 2012, only 2.5.a-d regarding convexity.

**Questions:**

I am not completely convinced by the right part of the equation 2, could you give more details or a reference?

Line 129, why is the matrix R of the observation error is assumed to be diagonal?

Why do the authors use ERA5 reanalysis data from 1979 to 2015?

In the supplementary material A.2., could the authors give a reference where this term is rigorously considered? So, further work to make a link between the complexity and the improvements of VAE-Var approach can be supported.

---

> ### Author Response · Authors · 2024-11-28
>
> We sincerely appreciate your careful and thoughtful comments and time, especially your approval for our proposed framework, which really means a lot to us. We try to explain your concerns point by point.
>
> **Q1: “Line 243-244, in Chen et al., the dimensionality of the problem is 189 * 721 * 1440. What does the 69 correspond to? Is it a confusion with what is used in the present study section 4.1?”**
>
> **A1:** Thank you for pointing this out. You are correct that in the original Chen et al. paper, the FengWu model's resolution is indeed described as 189 * 721 * 1440, where 189 = 4 + 37 * 5. This corresponds to 4 near-surface variables and 37 high-altitude levels with 5 variables each.
>
> However, the version of the FengWu model used in this study adopts a modified configuration with only 13 high-altitude levels. As a result, the total number of channels (or dimensionality) becomes 69 = 4 + 13 * 5, which is the version employed in the experiments presented in Section 4.1.
>
> We have now revised this part of the manuscript to clarify this distinction and ensure consistency. Thank you for catching this detail, and we appreciate your careful reading.
>
> **Q2: "Line 262-263, the authors claim that all the components of L are differentiable. Why H is differentiable?"**
>
> **A2:** Thank you for raising this important question.
>
> The observation operator $\mathcal{H}$ is a mapping function that transforms the physical field variables into observation space. In our experiments, the observation quantities share the same dimensional units as the physical state variables, but the observations are sparse and may not align directly with the grid points of the physical field. Thus, $\mathcal{H}$ in our study acts as an interpolation operator, which is implemented using the `F.Linear` function from the PyTorch library. Since `F.Linear` supports automatic differentiation, $\mathcal{H}$ is also differentiable in our setup. Please refer to the Appendix in the revised manuscript for details.
>
> Furthermore, in more complex observational settings, such as satellite observations where the physical field variables (e.g., wind, temperature, pressure) and observation quantities (e.g., brightness temperature) have different dimensional units, $\mathcal{H}$ can be derived by solving the radiative transfer equation. Libraries like JAX provide methods to differentiate through the solutions of such equations by converting them into neural network-compatible operations. Hence, $\mathcal{H}$ remains theoretically differentiable even in scenarios involving satellite observations.
>
> To summarize, whether in the relatively straightforward observational settings discussed in this paper or in more complex cases involving satellite observations, it is theoretically feasible to construct a differentiable observation operator $\mathcal{H}$.
>
> **Q3: "There is no 2.5.3. in Giaquinta & Modica 2012, only 2.5.a-d regarding convexity."**
>
> **A3:** Thank you for pointing this out. You are correct that the section 2.5.3 does not exist in the 2012 edition of Giaquinta & Modica. The reference in the manuscript was intended to cite section 2.5.2 in Giaquinta & Modica (2010), where the change of variables formula is discussed. We have also included a more detailed derivation in the revised manuscript. We appreciate your careful attention to detail and apologize for the confusion caused by this error.

---

> ### Author Response · Authors · 2024-11-28
>
> **Q4: "I am not completely convinced by the right part of equation 2, could you give more details or a reference?"**
>
> **A4:** Thank you for your insightful comment. We recognize that the derivation of Equation 2 in the manuscript may appear to skip some intermediate steps. Below, we provide a more detailed explanation to clarify the logic:
>
> We aim to maximize $ p(\mathbf{x} | \mathbf{x}\_b, \mathbf{y}) $, which can be rewritten as:
>
> $$
> \arg\max\_{\mathbf{x}} p(\mathbf{x}|\mathbf{x}\_b,\mathbf{y})=\arg\max\_{\mathbf{x}}\frac{p(\mathbf{x}\_b, \mathbf{y}|\mathbf{x}) p(\mathbf{x})}{p(\mathbf{x}\_b, \mathbf{y})}=\arg\max\_{\mathbf{x}} p(\mathbf{x}\_b, \mathbf{y}|\mathbf{x}) p(\mathbf{x}).
> $$
>
> Here, $p(\mathbf{x}\_b, \mathbf{y})$ is independent of $\mathbf{x}$ and thus can be omitted from the optimization.
>
> Now, assuming that $\mathbf{x}\_b$ and $\mathbf{y}$ are independent conditional on $\mathbf{x}$, we can decompose $p(\mathbf{x}\_b, \mathbf{y}|\mathbf{x})$ as $p(\mathbf{x}\_b|\mathbf{x}) p(\mathbf{y}|\mathbf{x})$. Substituting this into the equation gives:
>
> $$
> \arg\max\_{\mathbf{x}} p(\mathbf{x}|\mathbf{x}\_b, \mathbf{y}) = \arg\max\_{\mathbf{x}} p(\mathbf{x}\_b|\mathbf{x}) p(\mathbf{y}|\mathbf{x}) p(\mathbf{x}).
> $$
>
> Next, applying Bayes’ rule to $p(\mathbf{x}\_b|\mathbf{x})$, we have $p(\mathbf{x}\_b|\mathbf{x}) = p(\mathbf{x}|\mathbf{x}\_b)p(\mathbf{x}\_b)/p(\mathbf{x})$. Substituting this back, and noting that $p(\mathbf{x}\_b)$ is constant with respect to $\mathbf{x}$, we obtain:
>
> $$
> \arg\max\_{\mathbf{x}} p(\mathbf{x}|\mathbf{x}\_b, \mathbf{y}) = \arg\max\_{\mathbf{x}} p(\mathbf{x}|\mathbf{x}\_b) p(\mathbf{y}|\mathbf{x}).
> $$
>
> This is the right-hand side of Equation 2, which follows logically under the given assumptions.
>
> We will add these steps to the appendix of the revised manuscript to ensure the derivation is clear to readers. Thank you for pointing this out.
>
> **Q5: Why R is assumed to be diagonal?**
>
> **A5:** Thank you for pointing this out. The assumption of a diagonal observation error covariance matrix $\mathbf{R}$ is based on the following considerations:
>
> 1. In *Kalnay (2003)*, Chapter 5, Remark 5.4.2, $\mathbf{R}$ is decomposed into three components:
>    - **Instrumental error** ($\mathbf{R}_{\text{instr}}$), which arises from the measurement process and is generally assumed to be uncorrelated.
>    - **Error of representativeness** ($\mathbf{R}_{\text{repr}}$), caused by interpolating model grid values to observation locations, which is also typically treated as uncorrelated.
>    - **Error in the observation operator** ($\mathbf{R}_{H}$), introduced when converting model states to observation equivalents. This component can introduce correlations, especially in satellite observations where solving the radiative transfer equation often generates coupled errors.
> 2. In our study, we focus on ground-based station observations, where the observed variables directly correspond to the physical model variables. In this case, the observation operator error ($\mathbf{R}_{H}$) is negligible, leaving only the instrumental and representativeness errors, which are typically uncorrelated. Hence, $\mathbf{R}$ can reasonably be assumed diagonal in this specific context.
> 3. We acknowledge, however, that this diagonal assumption does not universally hold for general data assimilation problems, particularly for satellite observations or other cases involving complex observation operators. In that case, approaches like matrix decomposition should be applied to approximate $\mathbf{R}$.
>
> We will revise the manuscript to clarify these points and ensure the limitations of this assumption are explicitly discussed. Thank you for bringing this to our attention.
>
> **Q6: "Why do the authors use ERA5 reanalysis data from 1979 to 2015?"**
>
> **A6:** The decision to use ERA5 reanalysis data from 1979 to 2015 was based on the following considerations:
>
> 1. The ERA5 dataset begins in 1979, which marks the earliest available reanalysis data within this collection. This provides a natural starting point for constructing a training dataset.
> 2. Data from 2016 onward is typically reserved for validation and testing in order to assess the generalizability of the models. Following this convention ensures that the training data does not overlap with data used for model evaluation.
> 3. Additionally, this time range (1979–2015) aligns with the experimental setups used in the original FuXi (Chen, et al., 2023) and FengWu (Chen, et al., 2023) papers.

---

> ### Author Response · Authors · 2024-11-28
>
> **Q7: "In the supplementary material A.2., could the authors give a reference where this term is rigorously considered? So, further work to make a link between the complexity and the improvements of VAE-Var approach can be supported."**
>
> Thank you for highlighting this important point. In Bayesian modeling, one class of approaches rigorously addressing the Jacobian determinant of the mapping $\mathcal{D}$ from a Gaussian prior to the target dataset distribution is **normalizing flow** (Rezende & Mohamed, 2015). These methods constrain the structure of $\mathcal{D}$ to ensure the Jacobian determinant is computationally tractable. Typical constraints include the use of $1\times1$ convolution layers, single-variable activation functions, or invertible transformations with simple Jacobian structures. For instance, Glow is a well-known example in this family of methods (Kingma & Dhariwal, 2018).
>
> While normalizing flow approaches retain the universal approximation property, they face significant challenges when applied to high-dimensional systems, such as weather modeling. Specifically, the structural constraints lead to substantial computational complexity, making training time-intensive and resource-demanding. To our knowledge, few success has been achieved in modelling high-dimensional systems using normalizing flow.
>
> As part of future work, we will explore incorporating the advantages of normalizing flows into VAE-Var, with a focus on addressing the scalability and efficiency challenges in high-dimensional meteorological systems. Thank you for suggesting this direction; it provides valuable guidance for extending and improving our approach. We will include the discussion above in the revised manuscript.
>
> **References:**
>
> [1] Giaquinta, M., & Modica, G. (2010). *Mathematical analysis: An introduction to functions of several variables*. Springer Science & Business Media.
>
> [2] Kalnay, E. (2003). *Atmospheric Modeling, Data Assimilation and Predictability* (Vol. 341). Cambridge University Press.
>
> [3] Chen, L., Zhong, X., Zhang, F., Cheng, Y., Xu, Y., Qi, Y., & Li, H. (2023). FuXi: A cascade machine learning forecasting system for 15-day global weather forecast. *npj Climate and Atmospheric Science*, *6*(1), 190.
>
> [4] Chen, K., Han, T., Gong, J., Bai, L., Ling, F., Luo, J. J., ... & Ouyang, W. (2023). Fengwu: Pushing the skillful global medium-range weather forecast beyond 10 days lead. *arXiv preprint arXiv:2304.02948*.
>
> [5] Rezende, D., & Mohamed, S. (2015). Variational inference with normalizing flows. In *International conference on machine learning* (pp. 1530-1538). PMLR.
>
> [6] Kingma, D. P., & Dhariwal, P. (2018). Glow: Generative flow with invertible 1x1 convolutions. *Advances in neural information processing systems*, *31*.

---

> > ### Comment · Reviewer_D5gS · 2024-12-03
> > **Answer to Official Comment by Authors, A7**
> >
> > Thank you for your answer to my question.
> >
> > For A7, your answer is well detailed and will be interesting to the larger ICLR community.

---

> ### Comment · Reviewer_D5gS · 2024-12-03
> **Answer to Official Comment by Authors, A1-3**
>
> Thank you for your answer and corrections.
>
> For A1., this answer makes sense. The reader needs to be careful to understand it.
>
> For A 2., great, I see your answer in Appendix A 6., at line 347-348
>
> For A 3., yes, this is right.

---

> ### Comment · Reviewer_D5gS · 2024-12-03
> **Answer to Official Comment by Authors, A4-6**
>
> Thank you for your answers to my questions.
>
> For A 4., thanks, this is way clearer now.
>
> For A 5., this is interesting, this should be added to the supplementary material.
>
> For A 6., it makes sense. I note that 'Following the convention of the original paper (Cheng et al. 2023)' has been added line 289-290.

---

> ### Author Response · Authors · 2024-12-03
>
> Thank you for your response and for recognizing our work. Regarding A1, we will make it clearer in the revised manuscript. Regarding A5, in the revised version of the paper, we will include an explanation of the diagonal properties of $\mathbf{R}$ in the appendix.

---

### Comment · Area_Chair_AfWA · 2024-11-26

Dear all,

The deadline for the authors-reviewers phase is approaching (December 2).

@For reviewers, please read, acknowledge and possibly further discuss the authors' responses to your comments. While decisions do not need to be made at this stage, please make sure to reevaluate your score in light of the authors' responses and of the discussion.

- You can increase your score if you feel that the authors have addressed your concerns and the paper is now stronger.
- You can decrease your score if you have new concerns that have not been addressed by the authors.
- You can keep your score if you feel that the authors have not addressed your concerns or that remaining concerns are critical.

Importantly, you are not expected to update your score. Nevertheless, to reach fair and informed decisions, you should make sure that your score reflects the quality of the paper as you see it now. Your review (either positive or negative) should be based on factual arguments rather than opinions. In particular, if the authors have successfully answered most of your initial concerns, your score should reflect this, as it otherwise means that your initial score was not entirely grounded by the arguments you provided in your review. Ponder whether the paper makes valuable scientific contributions from which the ICLR community could benefit, over subjective preferences or unreasonable expectations.

@For authors, please respond to remaining concerns and questions raised by the reviewers. Make sure to provide short and clear answers. If needed, you can also update the PDF of the paper to reflect changes in the text. Please note however that reviewers are not expected to re-review the paper, so your response should ideally be self-contained.

The AC.

---

### Meta-Review · Area_Chair_AfWA · 2024-12-20

**Metareview:**

The reviewers recommend acceptance (8-8-6-6-6-5). The paper presents a data assimilation approach based on a variational autoencoder trained to encode/decode error fields. The approach is well-motivated and the results are convincing. The author-reviewer discussion has been constructive and has led to a number of improvements to the paper, in particular regarding the presentation, the discussion of related work, and the addition of new results. The reviewers have raised some concerns about the novelty of the approach, but the authors have provided convincing arguments arguing for the relevance of their work nonetheless. The reviewers also appreciate the comparison against reasonable baselines (3DVar, DiffDA) and the honest discussion of the limitations of the approach. During the discussion with Reviewer Uc8k, mentions and comparisons against spatio-temporal data assimilation methods such as 4DVar have been strongly encouraged, as they constitute the state-of-the-art in the field. As AC, I agree with this recommendation and request the authors to make sure this is properly discussed. For these reasons, I recommend acceptance. I encourage the authors to address the remaining concerns and to implement the modifications discussed with the reviewers in the final version of the paper.

**Additional Comments On Reviewer Discussion:**

he author-reviewer discussion has been constructive and has led to a number of improvements to the paper, in particular regarding the presentation, the discussion of related work, and the addition of new results.

---

### Decision · Program_Chairs · 2025-01-22

Accept (Poster)